# The connectome of the adult Drosophila mushroom body provides insights into function

Feng Li[1]*, Jack W Lindsey[2], Elizabeth C Marin[3], Nils Otto[3,4], Marisa Dreher[1], Georgia Dempsey[3], Ildiko Stark[3], Alexander S Bates[5], Markus William Pleijzier[5], Philipp Schlegel[3,5], Aljoscha Nern[1], Shin-ya Takemura[1], Nils Eckstein[1], Tansy Yang[1], Audrey Francis[1], Amalia Braun[3], Ruchi Parekh[1], Marta Costa[3], Louis K Scheffer[1], Yoshinori Aso[1], Gregory SXE Jefferis[3,5], Larry F Abbott[2], Ashok Litwin-Kumar[2], Scott Waddell[4], Gerald M Rubin[1]*

[1]Janelia Research Campus, Howard Hughes Medical Institute, Ashburn, United States; [2]Department of Neuroscience, Columbia University, Zuckerman Institute, New York, United States; [3]Drosophila Connectomics Group, Department of Zoology, University of Cambridge, Cambridge, United Kingdom; [4]Centre for Neural Circuits & Behaviour, University of Oxford, Oxford, United Kingdom; [5]Neurobiology Division, MRC Laboratory of Molecular Biology, Cambridge, United Kingdom

**\*For correspondence:**
lif@janelia.hhmi.org (FL);
rubing@janelia.hhmi.org (GMR)

**Competing interests:** The authors declare that no competing interests exist.

**Abstract** Making inferences about the computations performed by neuronal circuits from synapse-level connectivity maps is an emerging opportunity in neuroscience. The mushroom body (MB) is well positioned for developing and testing such an approach due to its conserved neuronal architecture, recently completed dense connectome, and extensive prior experimental studies of its roles in learning, memory, and activity regulation. Here, we identify new components of the MB circuit in *Drosophila*, including extensive visual input and MB output neurons (MBONs) with direct connections to descending neurons. We find unexpected structure in sensory inputs, in the transfer of information about different sensory modalities to MBONs, and in the modulation of that transfer by dopaminergic neurons (DANs). We provide insights into the circuitry used to integrate MB outputs, connectivity between the MB and the central complex and inputs to DANs, including feedback from MBONs. Our results provide a foundation for further theoretical and experimental work.

## Introduction

Dramatic increases in the speed and quality of imaging, segmentation and reconstruction in electron microscopy now allow large-scale, dense connectomic studies of nervous systems. Such studies can reveal the chemical synapses between all neurons, generating a complete connectivity map. Connectomics is particularly useful in generating biological insights when applied to an ensemble of neurons with interesting behavioral functions that have already been extensively studied experimentally. Knowing the effects on behavior and physiology of perturbing individual cell types that can also be unambiguously identified in the connectome is of considerable value. Here, we present a connectomic analysis of one such neuronal ensemble, the mushroom body (MB) of an adult *Drosophila melanogaster*.

Understanding how memories of past events are formed and then used to influence ongoing behavior are key challenges in neuroscience. It is generally accepted that parallel changes in connection strength across multiple circuits underlie the formation of a memory and that these changes are integrated to produce net changes in behavior. Animals learn to predict the value of sensory cues

based on temporal correlations with reward or punishment (*Pavlov and Thompson, 1902*). Such associative learning entails lasting changes in connections between neurons (reviewed in *Abraham et al., 2019*; *Martin et al., 2000*). It is now clear that different parts of the brain process and store different aspects of the information learned in a single event (reviewed in *Josselyn and Frankland, 2018*). In both flies and mammals, dopaminergic neurons play a key role in conveying information about whether an event has a positive or negative valence, and there are compelling parallels between the molecular diversity of dopaminergic cell types across these evolutionarily distant animals (*Watabe-Uchida and Uchida, 2018*). However, we have limited understanding of how information about the outside world or internal brain state reaches different dopaminergic populations. Nor do we understand the nature of the information that is stored in each parallel memory system or how these parallel memories interact to guide coherent behavior. We believe such processes are governed by general and evolutionarily-conserved principles. In particular, we believe the circuit logic that allows a brain to balance the competing demands of rapid learning with long-term stability of memory are likely to be the same in flies and mammals. Developing a comprehensive

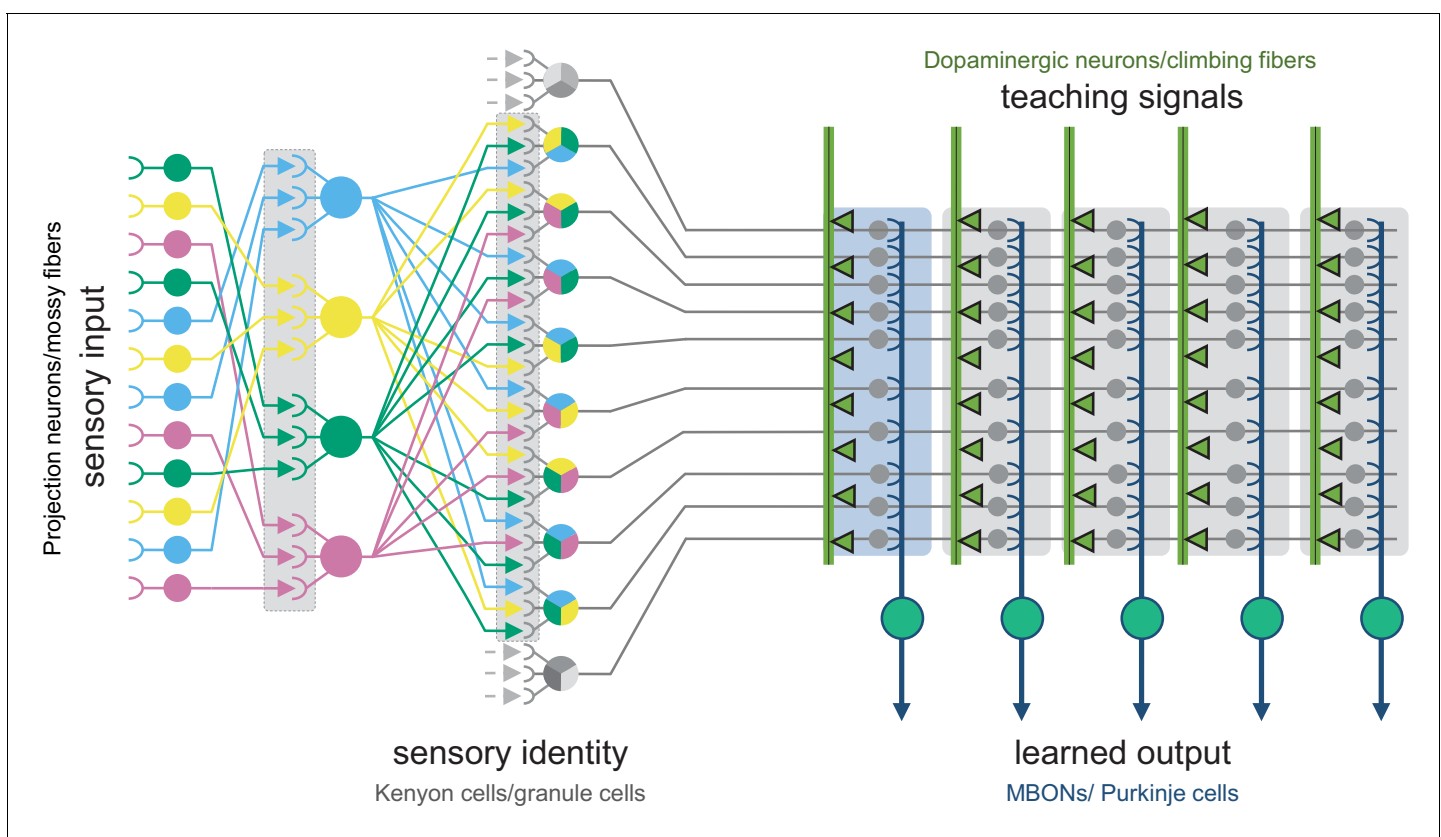

**Figure 1.** The shared circuit architecture of the mushroom body and the cerebellum. In both the insect MB and the vertebrate cerebellum sensory information is represented by sparse activity in parallel axonal fibers; Kenyon cells (KCs) in the MB and granule cells (GCs) in the cerebellum (reviewed in *Modi et al., 2020*). In general, each KC or GC has claw-like dendrites that integrate sensory input from a small number of neurons, called projection neurons in insects and mossy fibers in vertebrates. In the MB, teaching signals are provided by dopaminergic neurons (DANs) and in the cerebellum by climbing fibers. Learned output is conveyed to the rest of the brain from the MB lobes by MB output neurons (MBONs) or, from the cerebellar cortex, by Purkinje cells. The arbors of the DANs and MBONs overlap in the MB lobes and define a series of 15 compartments (*Aso et al., 2014a*; *Gao et al., 2019* ; *Figure 2*; *Figure 1—video 2*); similarly, overlap between the arbors of climbing fibers and Purkinje cells define zones along the GC parallel fibers.

The online version of this article includes the following video(s) for figure 1:

**Figure 1—video 1.** Introduction to the MB.

https://elifesciences.org/articles/62576#fig1video1

**Figure 1—video 2.** Introduction to MB compartments.

https://elifesciences.org/articles/62576#fig1video2

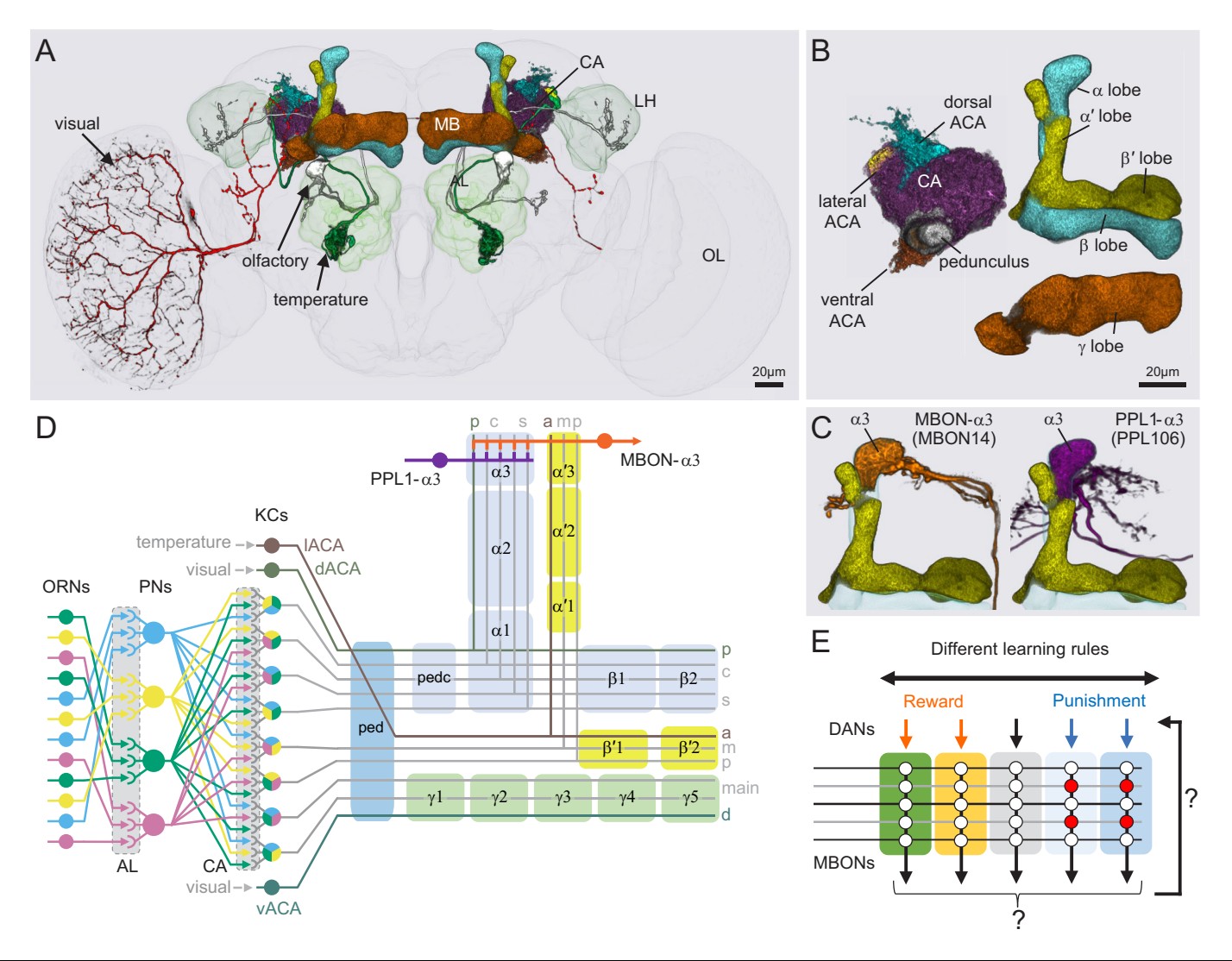

**Figure 2.** Anatomy of the adult *Drosophila* MB. Diagram of structure and information flow in the MB. (**A**) An image of the brain showing subregions of the MB (see panel B for more detail) and examples of the sensory pathways that provide information to the KCs. Projection neurons (PNs) from the 51 olfactory glomeruli of the antennal lobe (AL) extend axons to the calyx (CA) of the MB and the lateral horn (LH). A total of 126 PNs, using a threshold of 25 synapses, innervate the CA and two innervate the lACA. Six olfactory PNs from the DL3 glomerulus are shown (white). Also shown is a visual projection neuron, aMe12 (red) that conveys information from the optic lobe (OL) to the ventral accessory calyx (vACA) and a thermosensory projection neuron (green) that conveys cold temperature information from arista sensory neurons in glomerulus VP3 to the lACA; the positions of the accessory calyces are shown in (**B**). See *Figure 1—video 1* for additional details. (**B**) Subregions within the MB. The γ lobe, CA, and pedunculus are displayed separately from other lobes; their normal positions are as shown in panel A. Color-coding is as in panel A. (**C**) The MB output neuron (MBON14) whose dendrites fill the α3 compartment at the tip of the vertical lobe is shown along with the dopaminergic neuron (PPL106), whose axonal terminals lie in the same compartment. See *Figure 1—video 2* for more detailed examples of the structure of a compartment. (**D**) A schematic representation of the key cellular components and information flow during processing of sensory inputs to the MB. Olfactory receptor neurons (ORNs) expressing the same odorant receptor converge onto a single glomerulus in the AL. A small number (generally 3 – 4) of PNs from each of the 51 olfactory glomeruli innervate the MB, where they synapse on the dendrites of the ~2000 Kenyon cells (KCs) in a globular structure, the CA. Each KC exhibits, on average, six dendritic 'claws', and each claw is innervated by a single PN. The axons of the KCs project in parallel anteriorly through the pedunculus (ped) to the lobes, where KCs synapse onto the dendrites of MB output neurons (MBONs). KCs can be categorized into three major classes α/β, α′/β′, and γ, based on their projection patterns in the lobes (*Crittenden et al., 1998*). The β, β′, and γ lobes constitute the medial lobes (also known as horizontal lobes), while the α and α′ lobes constitute the vertical lobes. These lobes are separately wrapped by ensheathing glia (*Awasaki et al., 2008*). The α/β and α′/β′ neurons bifurcate at the anterior end of the ped (pedc) and project to both the medial and vertical lobes (*Lee et al., 1999*). The γ neurons project only to the medial lobe. Dendrites of MBONs and terminals of modulatory dopaminergic neurons (DANs) intersect the longitudinal axis of the KC axon bundle, forming 15 subdomains or compartments, five each in the α/β, α′/β′, and γ lobes (numbered α1, α2, and α3 for the compartments in the α lobe from proximal to distal and similarly for the other lobes; *Aso et al., 2014a*; *Tanaka et al., 2008*). Additionally, one MBON and one DAN innervate the

*Figure 2 continued on next page*

*Figure 2 continued*
core of the distal pedunculus (pedc) intersecting the α/β KCs. In the current work, we further classified KCs into 14 types, 10 main types and four unusual embryonic born KCs, named KCγs1-s4 (see *Figure 3*); the main KC types have their dendrites in the main calyx, with the following exceptions: The dendrites of γd KCs form the ventral accessory calyx (vACA; *Aso et al., 2009*; *Butcher et al., 2012*); those of the α/βp KCs form the dorsal accessory calyx (dACA; *Lin et al., 2007*; *Tanaka et al., 2008*); and the dendrites of a subset of α′/β′ cells form the lateral accessory calyx (lACA) (*Marin et al., 2020*; *Yagi et al., 2016*). These accessory calyces receive non-olfactory input (*Tanaka et al., 2008*). Different KCs occupy distinct layers in the lobes as indicated (p: posterior; c: core; s: surface; a: anterior; m: middle, main and d: dorsal). Some MB extrinsic neurons extend processes only to a specific layer within a compartment. (E) Individual compartments serve as parallel units of memory formation (see *Aso and Rubin, 2016*). Reward or punishment is conveyed by dopaminergic neurons, and the coincidence of dopamine release with activity of a KC modifies the strength of that KC's synapses onto the MBONs in that compartment. The circuit structure by which those MBONs combine their outputs to influence behavior and provide feedback to dopaminergic neurons are investigated in this paper.

The online version of this article includes the following figure supplement(s) for figure 2:

**Figure supplement 1.** The extent of the hemibrain volume and key to brain area nomenclature.

understanding of these circuits at the resolution of individual neurons and synapses will require the synergistic application of a variety of experimental methods together with theory and modeling. Many of the required methods are well developed in *Drosophila,* where the circuits underlying learning and memory are less complex than in mammals, and where detailed anatomical knowledge of the relevant circuits, which we believe will be essential, has just now become available. Here, we provide analysis of the complete connectome of a circuit involved in parallel processing of associative memories in adult fruit flies. The core architecture of this circuit is strikingly similar to that of the vertebrate cerebellum (*Figure 1*; *Laurent, 2002*; *Farris, 2011*; *Litwin-Kumar et al., 2017*).

The MB is the major site of associative learning in insects (reviewed in *Heisenberg, 2003*; *Modi et al., 2020*), and species that perform more complex behavioral tasks tend to have larger MBs (*O'Donnell et al., 2004*; *Sivinski, 1989*). In the MB of each brain hemisphere, sensory stimuli are represented by the sparse activity of ~2000 Kenyon cells (KCs) whose dendrites form a structure called the MB calyx and whose parallel axonal fibers form the lobes of the MB (*Figure 2*; *Figure 1—video 1*).

The major sensory inputs to the *Drosophila* MB are olfactory, delivered by ~150 projection neurons (PNs) from the antennal lobe to the dendrites of the KCs in the MB calyx (*Bates et al., 2020b*). KCs each receive input from an average of six PNs. For a KC to fire a spike, several of its PN inputs need to be simultaneously activated (*Gruntman and Turner, 2013*). This requirement, together with global feedback inhibition (*Lin et al., 2014a*; *Papadopoulou et al., 2011*), ensures a sparse representation where only a small percentage of KCs are activated by an odor (*Honegger et al., 2011*; *Perez-Orive et al., 2002*). The MB has a three layered divergent-convergent architecture (*Huerta et al., 2004*; *Jortner et al., 2007*; *Laurent, 2002*; *Litwin-Kumar et al., 2017*; *Shomrat et al., 2011*; *Stevens, 2015*) in which the coherent information represented by olfactory PNs is expanded and decorrelated when delivered to the KCs (*Caron et al., 2013*; *Zheng et al., 2020*). But the degree to which the structure of the sensory input representation is maintained by the KCs has been debated. We explore this issue, taking advantage of a nearly comprehensive dataset of KC inputs and outputs.

While best studied for its role in olfactory associative learning, the MB also receives inputs from several other sensory modalities. A subset of projection neurons from the antennal lobe delivers information about temperature and humidity in both the larva (*Eichler et al., 2017*) and the adult (*Frank et al., 2015*; *Liu et al., 2015*; *Marin et al., 2020*; *Stocker et al., 1990*). Taste conditioning also requires the MB and is believed to depend on specific KC populations, although the relevant inputs to these KCs have not yet been reported (*Kirkhart and Scott, 2015*; *Masek and Scott, 2010*). We identified one likely path for gustatory input to the MB.

*Drosophila* MBs are also known to be able to form memories based on visual cues (*Aso et al., 2014b*; *Brembs, 2009*; *Vogt et al., 2016*; *Vogt et al., 2014*; *Liu et al., 1999*; *Zhang et al., 2007*). Until a few years ago, it was thought that visual input reached the *Drosophila* MBs using only indirect, multisynaptic pathways (*Farris and Van Dyke, 2015*; *Tanaka et al., 2008*) as direct visual input from the optic lobes to the MBs, well known in Hymenoptera (*Ehmer and Gronenberg, 2002*), had not been observed in any dipteran insect (*Mu et al., 2012*; *Otsuna and Ito, 2006*). In 2016, *Vogt et al., 2016* identified two types of visual projection neurons (VPNs) connecting the optic lobes

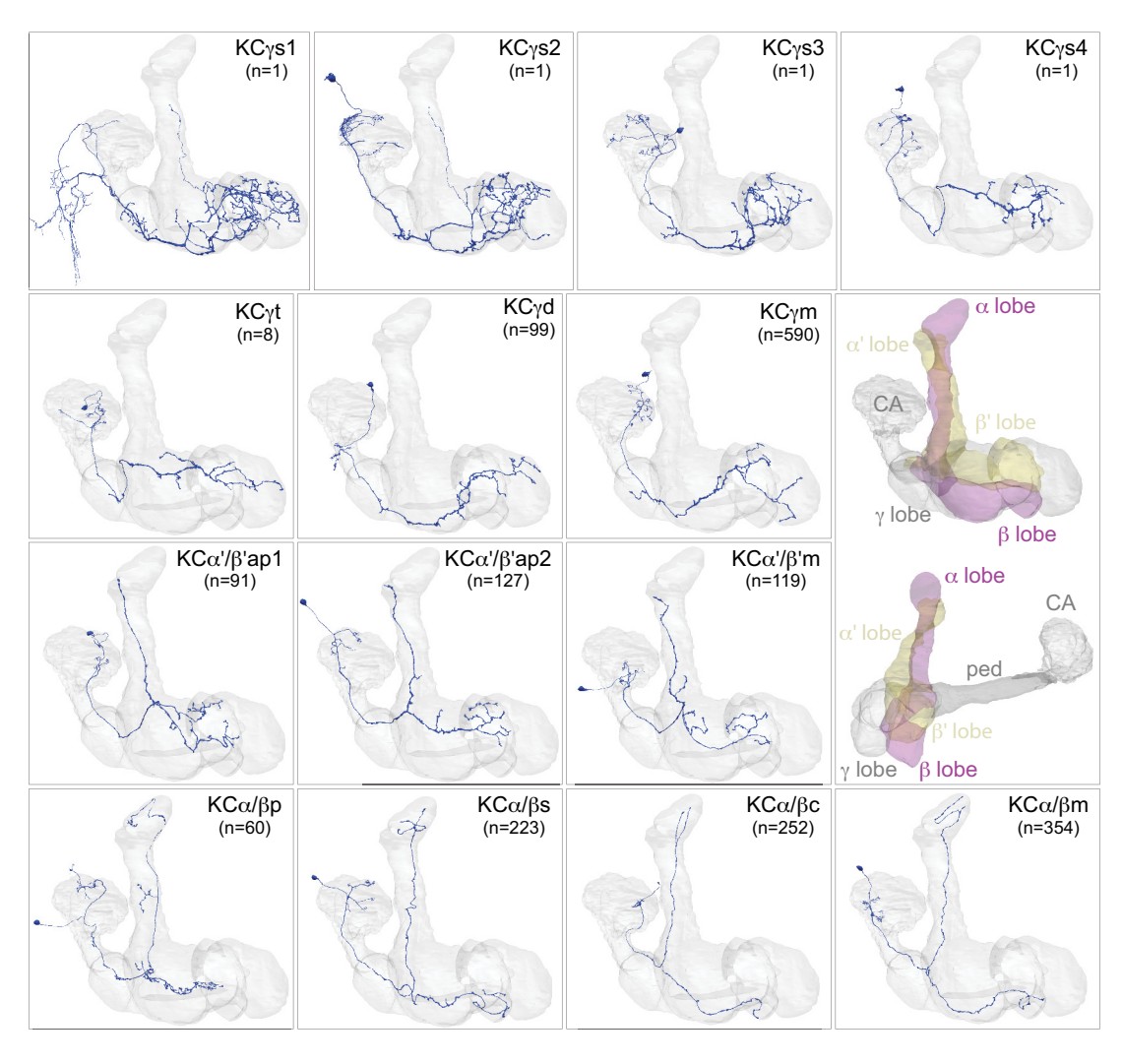

**Figure 3.** Kenyon cells. Each panel shows a representative neuron of the indicated KC subtype together with the outline of the MB lobes and CA in gray , in a perspective view from an oblique angle to better display neuronal morphology. The insert on the right provides a key to the position of the individual lobes, the pedunculus (ped) and CA; the upper image presents the same view as KC subtype panels and the lower image shows a rotated view to better visualize the ped and CA. The numbers (n=) indicate the number of cells that comprise each KC subtype in this animal; the number of KCs is known to vary between animals (reviewed in *Aso et al., 2009*). Several of the KC subtypes are defined here for the first time, based on morphological clustering, as described in *Figure 4*. Although reconstruction of 78 KCα/β was incomplete in the CA as a consequence of a small area of reduced image quality, it did not affect morphological clustering, which was based on simplified axonal skeletons. More information about each of these cell types is shown in *Figure 3—video 1*. Additional intrinsic and extrinsic neurons with processes in the MB are shown in *Figure 3—figure supplement 1*.

The online version of this article includes the following video and figure supplement(s) for figure 3:

**Figure supplement 1.** APL, DPM, SIFamide, OA-neurons and other modulatory neurons.

**Figure 3—video 1.** KC types.

https://elifesciences.org/articles/62576#fig3video1

**Figure 3—video 2.** APL and DPM.

https://elifesciences.org/articles/62576#fig3video2

**Figure 3—video 3.** SIFamide and octopaminergic neurons.

https://elifesciences.org/articles/62576#fig3video3

and the MB and additional connections have been observed recently by light microscopy (*Li et al., 2020*). We found that visual input was much more extensive than previously appreciated, with about 8% of KCs receiving predominantly visual input, and present here a detailed description of neuronal

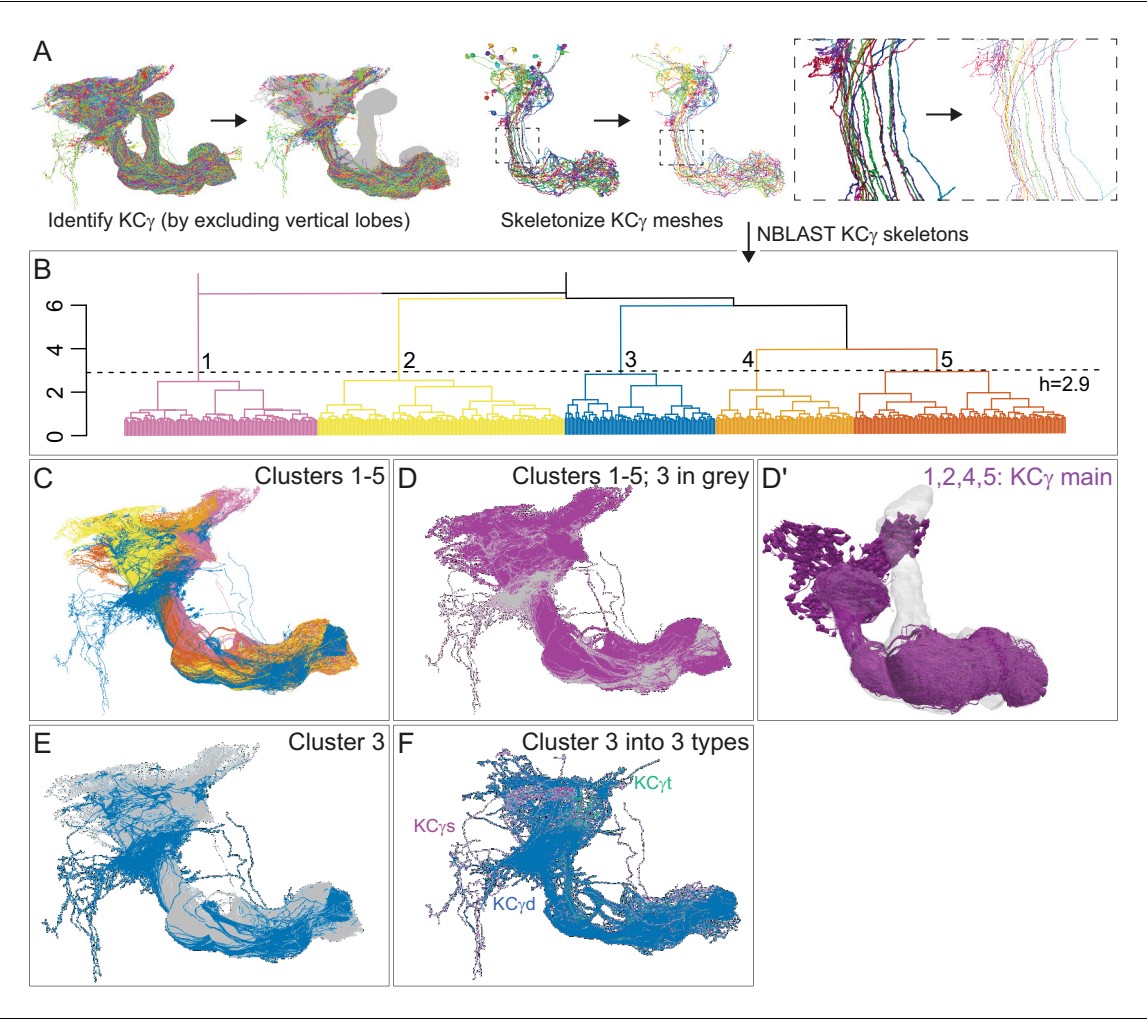

**Figure 4.** Morphological hierarchical clustering reveals previously unrecognized KC subtypes. (**A**) KC typing workflow, using KCγ as an example. All γ KCs in the population of annotated KCs were identified by excluding all KCs with axons in the vertical lobes. The space filling morphologies of γ KCs were converted to skeletons (enlarged in the dashed box for clarity) and then NBLAST all-by-all neuron clustering (*Costa et al., 2016*) was used to reveal morphological groups. (**B**) Morphological hierarchical clustering of KCγ based on NBLAST scores is shown, cut at height 2.9 (dashed line), which produces five clusters. (**C**) KCγ skeletons of those five clusters are shown, color-coded as in (**B**). (**D**) KCγ skeletons in clusters 1, 2, 4, and 5, which includes all 590 KCγm, are shown in magenta; cluster 3 is shown in gray. (**D'**) Space filling morphologies of KCγm (clusters 1, 2, 4, and 5) shown in magenta with the MB in gray. (**E**) KCγ skeletons in cluster 3 (blue), which includes all KCγd, and clusters 1 – 2 and 4 – 5 (gray), which correspond to the KCγm type, are shown. (**F**) KCγ skeletons from cluster 3, cut at height 1.3, which produces six sub-clusters corresponding to three color-coded subtypes: green, 3.1 (eight KCγt); magenta, 3.2 (four KCγs); and blue, 3.3 – 3.6 (99 KCγd); see *Figure 4—figure supplement 1* for details. The online version of this article includes the following video and figure supplement(s) for figure 4:

**Figure supplement 1.** Successive rounds of whole neuron morphological hierarchical clustering reveal novel KCγ subtypes.
**Figure supplement 2.** Three distinct morphological subtypes of KCα′/β′.
**Figure supplement 3.** Further explanation of KC subtype nomenclature.
**Figure supplement 4.** Four distinct morphological subtypes of KCα/β.
**Figure 4—video 1.** KC lineages.
https://elifesciences.org/articles/62576#fig4video1

pathways connecting the optic lobe and the MB. Visual sensory input appears to be segregated into distinct KC populations in both the larva (*Eichler et al., 2017*) and the adult (*Vogt et al., 2016*; *Li et al., 2020*), as is the case in honeybees (*Ehmer and Gronenberg, 2002*). We found two classes of KCs that receive predominantly visual sensory input, as well as MBONs that get the majority of their input from these segregated KC populations.

MBONs provide the convergence element of the MB's three layer divergent-convergent circuit architecture. Previous work has identified 22 types of MBONs whose dendrites receive input from specific axonal segments of the KCs. The outputs of the MBONs drive learned behaviors. Approximately 20 types of dopaminergic neurons (DANs) innervate corresponding regions along the KC axons and are required for associative olfactory conditioning. Specifically, the presynaptic arbors of the DANs and postsynaptic dendrites of the MBONs overlap in distinct zones along the KC axons, defining the 15 compartmental units of the MB lobes (*Aso et al., 2014a*; *Mao and Davis, 2009*; *Takemura et al., 2017*; *Tanaka et al., 2008*; *Figure 2*; *Figure 1—video 2*). A large body of evidence indicates that these anatomically defined compartments of the MB are the units of associative learning (*Aso et al., 2012*; *Aso et al., 2010*; *Aso et al., 2019*; *Aso et al., 2014a*; *Aso et al., 2014b*; *Aso and Rubin, 2016*; *Berry et al., 2018*; *Blum et al., 2009*; *Bouzaiane et al., 2015*; *Burke et al., 2012*; *Claridge-Chang et al., 2009*; *Isabel et al., 2004*; *Jacob and Waddell, 2020*; *Krashes et al., 2009*; *Lin et al., 2014b*; *Liu et al., 2012*; *Owald et al., 2015*; *Pai et al., 2013*; *Perisse et al., 2016*; *Qin et al., 2012*; *Plaçais et al., 2013*; *Schwaerzel et al., 2003*; *Séjourné et al., 2011*; *Trannoy et al., 2011*; *Yamagata et al., 2015*; *Zars et al., 2000*).

The DANs innervating different MBON compartments appear to play distinct roles in signaling reward vs. punishment, novelty vs. familiarity, the presence of olfactory cues and the activity state of the fly (*Aso et al., 2010*; *Aso and Rubin, 2016*; *Burke et al., 2012*; *Cohn et al., 2015*; *Hattori et al., 2017*; *Liu et al., 2012*; *Sitaraman et al., 2015b*; *Tsao et al., 2018*). These differences between DAN cell types presumably reflect in large part the nature of the inputs that each DAN receives, but our knowledge of these inputs is just emerging (*Otto et al., 2020*) and is far from comprehensive. DANs adjust synaptic weights between KCs and MBONs with cell type-specific rules and, in at least some cases, these differences arise from the effects of co-transmitters (*Aso et al., 2019*). In general, a causal association of KC responses with the activation of a DAN in a compartment results in depression of the synapses from the active KCs onto MBONs innervating that compartment (*Hige et al., 2015*; *Handler et al., 2019*). Different MB compartments are known to store and update non-redundant information as an animal experiences a series of learning events (*Berry et al., 2018*; *Felsenberg et al., 2017*; *Felsenberg et al., 2018*). In rodent and primate brains, recent studies have revealed that dopaminergic neurons are also molecularly diverse and encode prediction errors and other information based on cell type-specific rules (*Hu, 2016*; *Menegas et al., 2018*; *Poulin et al., 2020*; *Watabe-Uchida et al., 2017*).

MBONs convey information about learned associations to the rest of the brain. Activation of individual MBONs can cause behavioral attraction or repulsion, according to the compartment in which their dendrites arborize (*Aso et al., 2014b*; *Owald et al., 2015*; *Perisse et al., 2016*). The combined output of multiple MBONs is likely to be integrated in downstream networks, but we do not understand how memories stored in multiple MB compartments alter these integrated signals to guide coherent and appropriate behaviors. Prior anatomical studies implied the existence of multiple layers of interneurons between MBONs and descending motor pathways (*Aso et al., 2014a*). What is the nature of information processing in those layers? Anatomical studies using light microscopy provided the first hints. MBONs from different compartments send their outputs to the same brain regions, suggesting that they might converge on shared downstream targets. DANs often project to these same brain areas, raising the possibility of direct interaction between MBONs and DANs. The functional significance of such interactions has just begun to be investigated (*Felsenberg et al., 2017*; *Felsenberg et al., 2018*; *Ichinose et al., 2015*; *Jacob and Waddell, 2020*; *Pavlowsky et al., 2018*; *Perisse et al., 2016*; *Zhao et al., 2018b*), and studies of the *Drosophila* larva, where a connectome of a numerically less complex MB is available (*Eichler et al., 2017*), are providing valuable insights (*Eschbach et al., 2020a*; *Eschbach et al., 2020b*; *Saumweber et al., 2018*).

The recently determined connectome of a portion of an adult female fly brain (hemibrain; see *Figure 2—figure supplement 1*) provides connectivity data for ~22,500 neurons (*Scheffer et al., 2020*). Among them, ~2600 neurons have axons or dendrites in the MB, while ~1500 neurons are directly downstream of MBONs (using a threshold of 10 synapses from each MBON to each downstream target) and ~3200 are upstream of MB dopaminergic neurons (using a threshold of five synapses from each upstream neuron to each DAN). Thus we will consider approximately one-third of the neurons in the central brain in our analysis of the MB ensemble.

Throughout the paper we set synaptic thresholds in order to focus our descriptions and analyses on the most strongly connected neurons. In the above analysis, we chose a higher threshold for

MBON connections to downstream targets than for DAN inputs because the typical MBON has many more output synapses than a DAN has input synapses. At a thresholds of five synapses, DANs have a median of 31 different input neurons, but if we increased the threshold to 10 synapses this would decrease to only six different neurons. In contrast, at the threshold of 10 synapses, MBONs are connected to a median of 90 downstream neurons. There were some limitations resulting from not having a wiring diagram of the full central nervous system, as we lacked complete connectivity information for neurons with processes that extended outside the hemibrain volume (*Figure 2—figure supplement 1*). We were generally able to mitigate these limitations by identifying the corresponding neurons in other EM or light microscopic datasets when the missing information was important for our analyses. Thus the hemibrain dataset was able to support a nearly comprehensive examination of the full neural network underlying the MB ensemble.

Studies of the larval MB are providing parallel information on the structure and function of an MB with most of the same cell types, albeit fewer copies of each (*Eichler et al., 2017*; *Eschbach et al., 2020a*; *Eschbach et al., 2020b*; *Saumweber et al., 2018*). The microcircuits inside three MB compartments in the adult were previously described (*Takemura et al., 2017*) and we report here that the overall organization of these three compartments is conserved in a second individual of a different gender. More importantly, we extend the analysis of microcircuits within the MB lobes to all 15 compartments, revealing additional aspects of spatial organization within individual compartments.

In the current study, we were able to discover new morphological subtypes of KCs and to determine the sensory inputs delivered to the dendrites of each of the ~2000 KC. We found considerable structure in the organization of those inputs and unexpectedly high levels of visual input, which was the majority sensory input for two classes of KCs. This segregation of distinct sensory representations into channels is maintained across the MB, such that MBONs, by sampling from different KCs, have access to different sensory modalities and representations. We discovered a new class of 'atypical' MBONs, consisting of 14 cell types, that have part of their dendritic arbors outside the MB lobes, allowing them to integrate input from KCs with other information; at least five of them make strong, direct synaptic contact onto descending neurons that drive motor action. We describe how MBONs from different compartments interact with each other to potentially integrate and transform the signals passed from the MB to the rest of the brain, revealing a number of circuit motifs including multi-layered MBON-to-MBON feedforward networks and extensive convergence both onto common targets and onto each other through axo-axonal connections. Finally, we analyzed the inputs to all 158 DANs that innervate the MB. We found extensive direct feedback from MBONs to the dendrites of DANs, providing a mechanism of communication within and between MB compartments. We also found groups of DANs that share common inputs, providing mechanistic insights into the distributed parallel processing of aversive and appetitive reinforcement and other experimental observations.

## Results

### An updated MB cell type catalog

The MB can be divided into three distinct parts: the calyx, the pedunculus, and the lobes (*Figure 2*; *Figure 1—video 1*). The calyx is the input region for sensory information; KCs have their dendrites in the calyx where they receive inputs from projection neurons. The calyx has subregions: the main calyx (CA) and three accessory calyces. The CA gets over 90% of its sensory information from olfactory projection neurons, whereas the smaller accessory calyces are sites of non-olfactory input. The lobes are the main output region of the MB; the axons of the KCs make synapses along their length, as they transverse the lobes, to the dendrites of the MBONs. The pedunculus consists of parallel KC axonal fibers that connect the CA and the lobes and is largely devoid of external innervation in the adult. Voltage-gated sodium channels are concentrated in the proximal peduculus where they are likely to serve as the initiation point for KC action potentials (*Ravenscroft et al., 2020*). There are five MB lobes: α, β, α′, β′, and γ. In *Drosophila*, the α and α′ lobes are often called the vertical lobes, and the β, β′, and γ lobes are collectively called the medial, or horizontal, lobes. Each lobe is further divided into compartments by the innervation patterns of DANs and MBONs (*Figure 2*; *Figure 1—video 2*). Although the individual lobes are surrounded by glia and some glia extend fine processes into the lobes, there does not appear to be a glial-based boundary between compartments

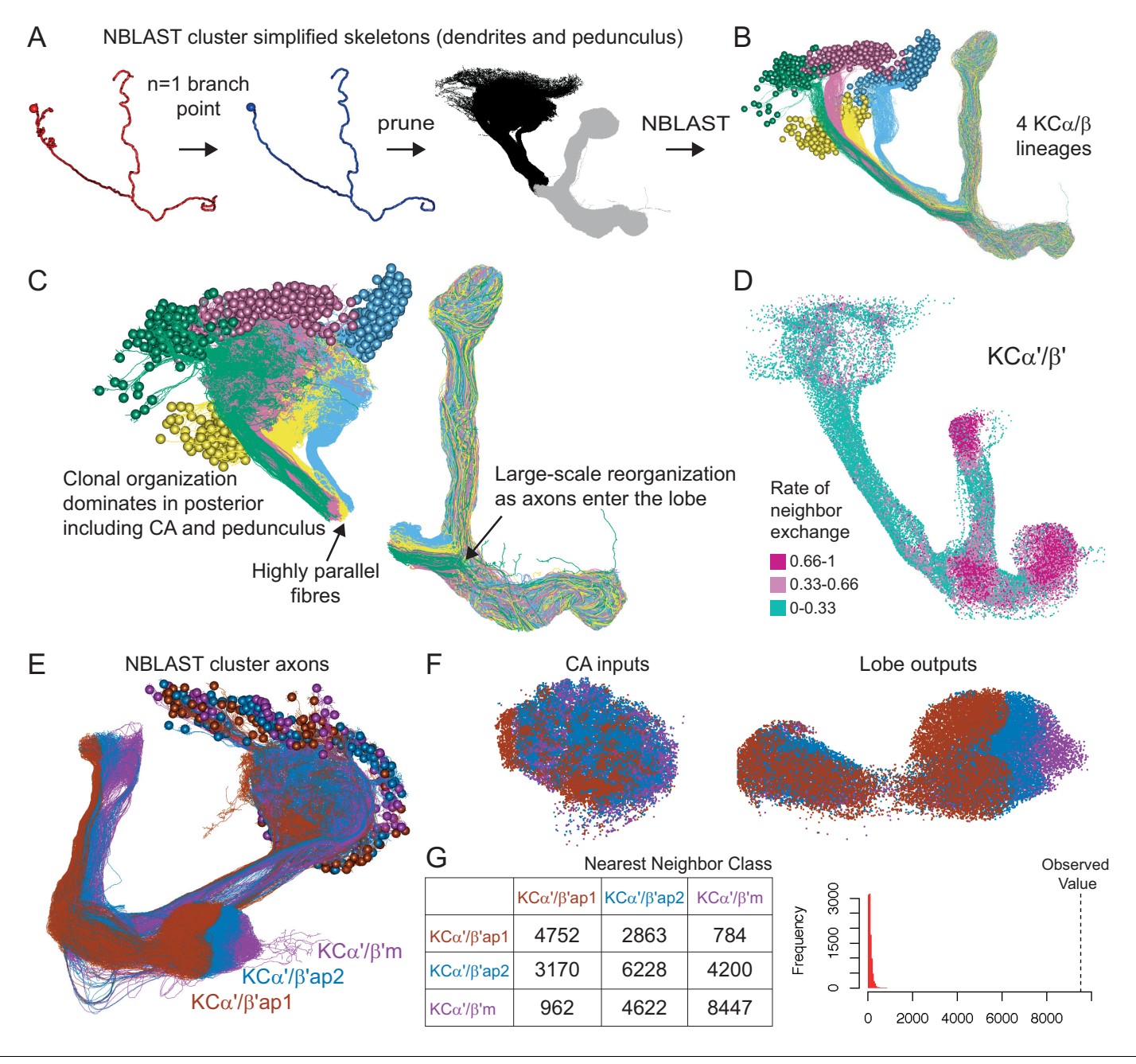

**Figure 5.** Organizational features of KC projections. (**A**) KCs were simplified to skeletons with one major branch point which were used as input for NBLAST all-by-all whole neuron clustering. (**B**) This clustering revealed the four clonal units that make up the mushroom body (shown here for KCα/β). (**C**) Full neuronal morphologies of the four clusters show that the positions of the neurons in the CA are strongly influenced by this fourfold clonal unit structure, which is also reflected in the arrangement of the highly parallel fibers in the pedunculus. On entering the lobes, the axons reorganize, and the neurons from the four clusters become intermingled. (**D**) Visualization of the rate of change in KC neighbors quantified as the fraction of 10 nearest neighbors that change compared with a position 5 μm closer to the soma. Large values imply a rapid change in the neighbors of individual KC fibers, which is observed at the entry and tips of the KCα′/β′ lobe as illustrated here. *Figure 5—figure supplement 1B* shows similar visualizations for the KCγ and KCα/β lobes; rapid change in neighbors is seen throughout the KCγ lobe while KCα/β neurons show an intermediate rate of change. (**E**) NBLAST clustering of the axons of α′/β′ KCs reveals three clear laminae in the vertical and horizontal lobes, which correlate with a layered organization in the CA and correspond to the three KC α′/β′ subtypes. (**F**) A similar organization is seen for KCα′/β′ dendrites in the CA and axon outputs in the lobes. (**G**) As a statistical test for the correlated lobe/CA organization into three subtypes, each synapse in the CA was matched with its closest neighbor from another neuron and the subtype of that neighbor recorded. The contingency table (left) shows that nearest neighbor synapses were most commonly from the same subtype. A permutation test (n = 10,000) confirmed that this statistic was far higher than expected by chance (right).

*Figure 5 continued on next page*

*Figure 5 continued*

The online version of this article includes the following figure supplement(s) for figure 5:

**Figure supplement 1.** Additional organizational features of KC projections.

**Figure supplement 2.** KC-to-KC synapses.

(*Ito et al., 1997*; *Kremer et al., 2017*; *Takemura et al., 2017*). In the hemibrain volume, glial cell processes were identified but were not analyzed further, preventing us from exploring the possibility of synapses between glia and neurons (*Scheffer et al., 2020*).

We compared the morphology of each EM reconstructed neuron to light-microscopy images of genetic driver lines that had been used previously to define the cell types in the MB. Guided by these comparisons, we assigned cell type names that corresponded to established names to the extent possible. In some cases, the availability of full EM reconstructed morphologies allowed us to discern additional subtypes. We also discovered an entirely new class of MBONs, the atypical MBONs, that differed from previously described MBONs in having dendrites both inside and outside the MB lobes. The next few sections describe this updated catalog of MB cell types, including KCs (*Figures 3*, *4*, *5*; *Figure 3—video 1*, *Figure 4—video 1*), other MB intrinsic and modulatory neurons (*Figure 3—figure supplement 1*; *Figure 3—video 2* and *3*), DANs (*Figure 6*), MBONs (*Figure 7*), and atypical MBONs (*Figure 8*; *Figure 8—figure supplements 1–15*; *Figure 8—video 1*, *2*, *3*, *4*, *5*, *6*, *7*, *8*, *9*, *10*, *11*, *12*, *13*, *14*). Most MB cell types were named based on light-microscopy analyses of their specific innervation areas inside the MB. For instance, MBON-α3 has its dendrites in the third compartment of the α lobe. MBONs and DANs also have synonymous names based on numbering (e.g. MBON14 for MBON-α3), which are primarily used in this report; *Figure 6—figure supplement 1* shows the neurons contained in each MB compartment and lists their alternative names.

## KCs: the major MB intrinsic neurons and conveyors of sensory identity

Associative memories in the MB are stored as altered synaptic weights between KCs, which represent sensory information, and their target MBONs (*Bouzaiane et al., 2015*; *Cassenaer and Laurent, 2012*; *Hige et al., 2015*; *Owald et al., 2015*; *Pai et al., 2013*; *Perisse et al., 2016*; *Séjourné et al., 2011*). Each of the 15 MB compartments is unique in cellular composition, and individual compartments can exhibit internal substructure, which we discuss later in the paper. In this next section we consider the different types of KCs that project to each compartment. In later sections, we examine how the various KC types receive distinct sensory information from projection neurons in the calyces and then connect differentially with MBONs to provide each MBON cell type with access to a different sensory space to use in forming memories. The number of KCs connected with each MBON also varies significantly from 122 (MBON10) to 1694 (MBON11), which can influence memory capacity and specificity.

We identified 1927 KCs in the right brain hemisphere. There are three major KC classes: α/β, α′/β′, and γ KCs. KCs are sequentially generated from four neuroblasts in the order of γ, α′/β′, and α/β (*Figure 4—video 1*; *Crittenden et al., 1998*; *Ito et al., 1997*; *Lee et al., 1999*; *Zhu et al., 2003*). The axons of α/β and α′/β′ KCs bifurcate at the distal peduncle to innervate the α and β lobes or the α′ and β′ lobes, respectively. The axons of γ KCs also branch but are confined to the γ lobe. Genetic driver lines, immunohistochemistry and single-cell morphology has revealed further subtypes (*Aso et al., 2009*; *Lin et al., 2007*; *Strausfeld et al., 2003*; *Tanaka et al., 2008*). Here, we grouped 1927 KCs into 14 subtypes (*Figure 3*; *Figure 3—video 1*) by applying NBLAST morphological clustering (*Bates et al., 2020a*; *Costa et al., 2016*) to each major class of KCs (*Figure 4*). Despite the dominance of olfactory input to the MB, all three major classes of KCs were found to contain small subsets dedicated to non-olfactory information.

### γ KCs

KCs that innervate the γ lobe (KCγ) have been traditionally divided into two groups, dorsal (KCγd) and main (KCγm). Axons of γd KCs innervate the dorsal layer of the γ lobe and γm KCs innervate the rest of the γ lobe. The dendrites of γd KCs arborize in the ventral accessory calyx (vACA), whereas those of γm KCs are found in the CA (*Aso et al., 2014a*; *Aso et al., 2009*; *Vogt et al., 2016*). We

identified the 701 γ KCs by excluding any KCs that innervated the vertical lobes (*Figure 4*). We then converted their 3D morphologies (meshes) into skeletons and performed an all-by-all NBLAST, which allowed us to define new KCγ subtypes and enumerate the members of each type: 590 KCγm, 99 KCγd, eight KCγt neurons with dendrites in the anterior CA (targeted preferentially by thermo/hygrosensory neurons), and four unique KCγs neurons sampling from one or more accessory calyces (*Figure 4—figure supplement 1*). All NBLAST clusters were validated by an independent clustering based on connectivity, CBLAST (*Scheffer et al., 2020*), and the small number of discrepancies were resolved by manual inspection (see Materials and methods). The birth order of these subtypes could not be definitively determined from the data, but the relative positions of their axons in the peduncle are consistent with the KCγs being generated first, followed by the KCγd and KCγt, and finally the KCγm (*Figure 5—figure supplement 1*).

## α′/β′ KCs

We identified 337 α′/β′ KCs, which could be divided into three subtypes using all-by-all NBLAST on their axons (*Figure 4—figure supplement 2*). The axons of each subtype formed a distinct layer in both the vertical and horizontal lobes. We named these subtypes to be consistent with prior nomenclature based on split-GAL4 driver lines (*Figure 4—figure supplement 3*). There are 91 α′/β′ap1 (*Figure 4—figure supplement 2B,B′*), 127 α′/β′ap2 (*Figure 4—figure supplement 2D,D′*), and 119 α′/β′m (*Figure 4—figure supplement 2C,C′*) KCs. While the somas of these subtypes do not segregate into clear clusters within each presumed neuroblast lineage, their axon layers suggest that they are generated in succession: α′/β′ap1, then α′/β′ap2, and finally α′/β′m (*Figure 4—figure supplement 2B–D*). Moreover, we found that each subtype's axon layer was correlated with the position of its dendrites in the CA (*Figure 5E*). The dendrites of the α′/β′ap1 KCs lie in the lateral accessory calyx and anterior CA, areas that are preferentially targeted by thermo/hygrosensory sensory projection neurons (Figure 12B).

## α/β KCs

We identified 889 α/β KCs, which could be divided into four subtypes using all-by-all NBLAST on their axons (*Figure 4—figure supplement 4*). The first subtype corresponds to the 60 KCα/βp that form the posterior layer of the α and β lobes (*Figure 4—figure supplement 4C,C′*); these are the first α/β KCs to be born and have been referred to as pioneer KCα/β neurons for this reason (*Lin et al., 2007*; *Zhu et al., 2003*). The remaining three groups form concentric layers (surface, middle, and core) in both the α and β lobes, yielding 223 KCα/βs (*Figure 4—figure supplement 4D,D′*), 354 KCα/βm (*Figure 4—figure supplement 4F,F′*), and 252 KCα/βc (*Figure 4—figure supplement 4E,E′*). The somata of each of these three subtypes appear to form distinct clusters, while the arrangement of their axon layers indicates that they are generated in the order KCα/βs, KCα/βm, and KCα/βc (*Figure 4—figure supplement 4D–F*). Dendrites of α/βp KCs form the dorsal accessory calyx (dACA), while the rest of α/β KCs have dendrites in the CA. Our classification of α/β KC subtypes is consistent with prior light-level studies (*Tanaka et al., 2008*; *Zhu et al., 2003*).

Each of the four neuroblasts whose progeny form the MB lobes is thought to generate all classes of KCs, but their exact contributions have been difficult to assess. There is no labeling of neuroblast origin in EM images, but neurons derived from the same neuroblast tend to have cell bodies in close proximity and primary neurites bundled into the same fiber tract. Tight groupings of cell bodies are particularly evident for α/β KCs. To classify α/β KCs into four neuroblast groups, we applied NBLAST to simplified skeletons of α/β KCs whose axons in the lobes had been removed (*Figure 5A*). As expected, we found four equal-sized groups of α/β KCs that we believe are each the descendants of a single neuroblast (*Figure 5B*). These four groups form subregions in the CA and pedunculus, but their axons are scrambled in the lobes (*Figure 5C*) as previously demonstrated by genetic methods (*Lin et al., 2007*; *Zhu et al., 2003*).

Upon entering the MB lobes, the axons of each KC type project to spatially segregated layers in the lobes (*Figure 4—video 1*), with the exception of γm KC axons which meander along the length of the horizontal lobe (*Figure 5—figure supplement 1B*). This maintained segregation is most prominent for α′/β′ KCs but is also seen in α/β KCs (*Figure 5E–G*). The dendrites of each KC type also tend to be found in the same region of the CA (*Figure 4—video 1*; *Leiss et al., 2009*; *Lin et al., 2007*; *Zheng et al., 2020*), which, in some cases, appears to support input specialization.

These features of the spatial mapping from CA to lobes and the organization of the parallel fiber system presumably evolved to facilitate associative learning. This spatial arrangement gives each MBON the possibility of receiving mixed or segregated sensory information depending on where that MBON extends its dendrite within the different KC layers, and, similarly, gives DANs the potential ability to modify strengths of synapses from KCs conveying specific sensory information. KCs make synapses to neighboring KCs in the calyx, pedunculus, and lobes. These were described for the α lobe in *Takemura et al., 2017*, and *Figure 5—figure supplement 2* provides a summary for the entire MB.

## DANs: the providers of localized neuromodulation

For associative learning to occur, the neuronal pathways that convey punishment or reward must converge with those that convey sensory cues. In the fly brain, this anatomical convergence takes place in the compartments of MB lobes: sensory cues are represented by the sparse activity of KCs and reinforcement signals by the DANs that innervate the MB lobes. DANs have been traditionally grouped into two clusters, PPL1 and PAM, based on the position of their cell bodies (*Figure 6*). PPL1 DANs innervate the vertical lobes and generally convey punishment, whereas PAM DANs innervate horizontal lobes and generally convey reward (*Aso et al., 2019*; *Aso et al., 2012*; *Aso et al., 2010*; *Aso and Rubin, 2016*; *Burke et al., 2012*; *Claridge-Chang et al., 2009*; *Felsenberg et al., 2018*; *Felsenberg et al., 2017*; *Huetteroth et al., 2015*; *Jacob and Waddell, 2020*; *Lin et al., 2014b*; *Liu et al., 2012*; *Mao and Davis, 2009*; *Schwaerzel et al., 2003*). There is also a DAN from the PPL2ab cluster, PPL201 (*Figure 3—figure supplement 1G*), that innervates the CA (*Mao and Davis, 2009*; *Tanaka et al., 2008*; *Zheng et al., 2018*) and has been reported to play a role in signaling saliency (*Boto et al., 2019*). We defined six PPL1 DAN cell types (PPL101-PPL106; see *Figure 1—video 2* for PPL106) and 15 PAM DAN cell types (PAM01-PAM15). These cell type classifications are consistent with previous studies (*Aso et al., 2014a*), except for the addition of one new type, PAM15 (γ5β′2a). There is only a single cell per PPL1-DAN cell type in a hemisphere, and axons of each cell broadly arborize in the compartment(s) they innervate. In contrast, there are between 3 and 26 cells per PAM DAN cell type, and the axonal terminals of an individual PAM DAN occupy only a portion of the compartment it innervates (see *Figure 29—video 1* for PAM11 and *Figure 29—video 3* and Figure 32 for PAM12). Thus, it is possible to further subdivide the members of PAM DAN cell types in *Figure 6* into smaller groups based on morphology and connectivity as described in *Otto et al., 2020*. We present an extensive analysis of such subtypes later in the paper (Figures 27–37).

## MBONs: the MB's conduit to the brain for learned associations

The representations of sensory cues and memory traces encoded in KC axon terminals have been reported to be read out by a network of 22 types of MBONs (*Aso et al., 2014a*; *Takemura et al., 2017*). We found all the previously described MBON types in the hemibrain volume (*Figure 7*), except for MBON08 which is not present in the imaged fly. MBONs can be categorized into three groups by their transmitters, which also correspond to anatomical and functional groups. Dendrites of glutamatergic MBONs arborize in the medial compartments of the horizontal lobes, which are also innervated by reward-representing PAM DANs. Most cholinergic MBONs arborize in the vertical lobes, in compartments that are also innervated by punishment-representing PPL1-DANs. GABAergic MBONs also arborize in compartments innervated by punishment-representing DANs. As described above, axon fibers of distinct types of olfactory and non-olfactory KCs form layers in the MB lobes. Each MBON arborizes its dendrites in a subset of layers where they receive excitatory, cholinergic synapses from KCs (*Barnstedt et al., 2016*; *Takemura et al., 2017*). These KC synapses are known to be presynaptically modulated by dopamine (*Davis, 2005*; *Hige et al., 2015*; *Kim et al., 2007*). Within the MB lobes, MBONs also receive input from APL (*Liu and Davis, 2009*) and DPM (*Waddell et al., 2000*) as well as from DANs (*Takemura et al., 2017*).

   MBONs generally project their axons outside the MB lobes, with the exception of three feedforward MBONs that project to other MB compartments (*Aso et al., 2014a*). As discussed in detail below (Figures 18–25, *Figure 22—video 1–3*), MBONs most heavily innervate dorsal brain areas such as the CRE, SIP, and SMP (Figure 18), make direct connections to the fan-shaped-body of the central complex (Figures 19 and 20), tend to converge on common targets (Figure 21), form a multi-

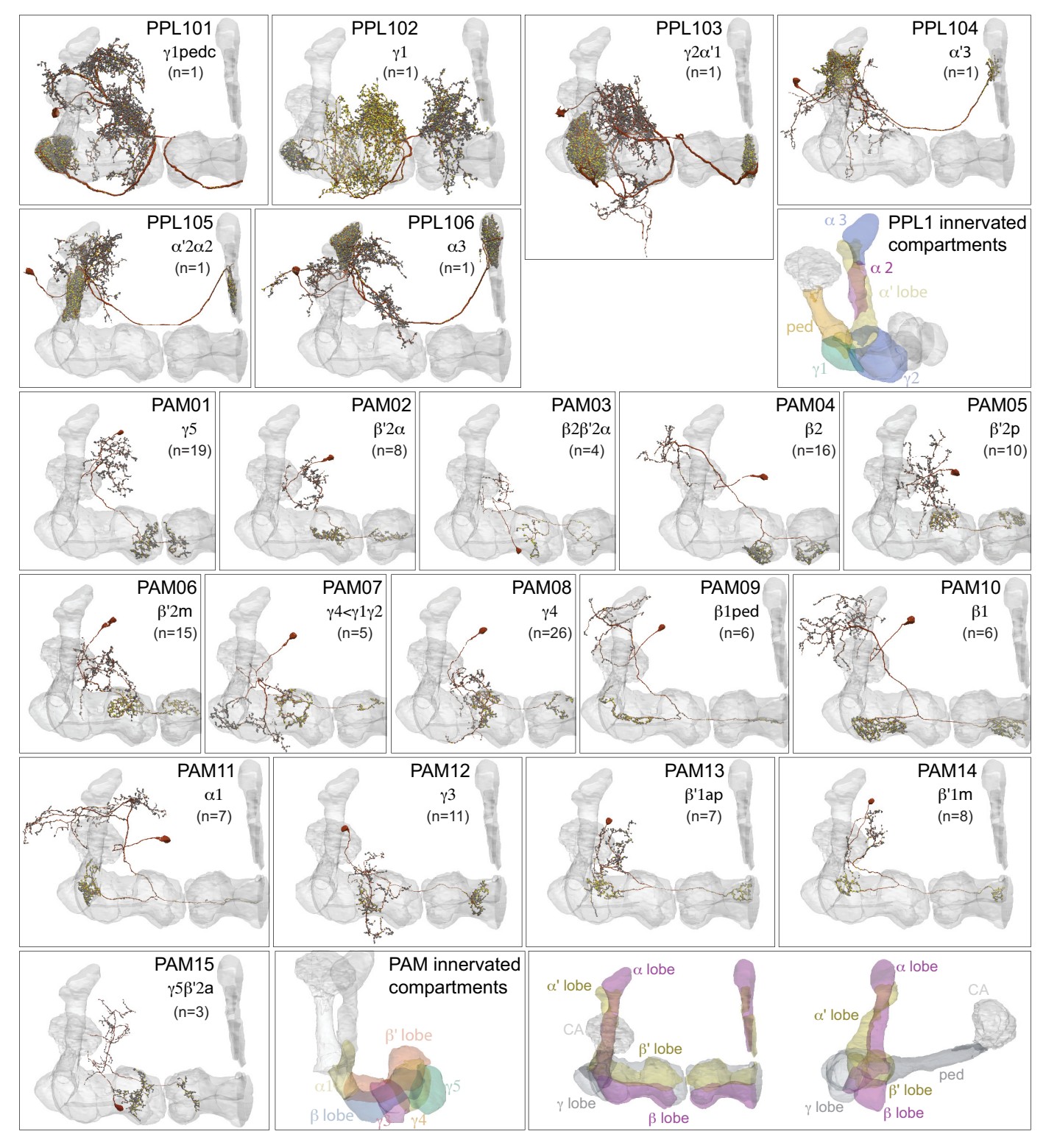

**Figure 6.** Dopaminergic neurons (DANs). Each panel shows a DAN cell type, with its name, the compartment(s) it innervates and the number of cells of that type per brain hemisphere indicated; the outline of the MB lobes and CA are shown in gray , in a perspective view from an oblique angle to better display neuronal morphology. *Figure 6—figure supplement 1* shows which DANs, MBONs, and KCs are found in each compartment. PPL1 dopaminergic neurons are divided into six cell types, PPL101, PPL102, PPL103, PPL104, PPL105, and PPL106. As a population, the PPL1 neurons innervate the α' lobe, α lobe compartments 2 and 3, and γ lobe compartments 1 and 2, as illustrated. There is only one PPL1 DAN of each type per

*Figure 6 continued on next page*

*Figure 6 continued*

hemisphere, but they send their axons bilaterally to innervate the same MB compartments, although less densely, in the other brain hemisphere (see *Aso et al., 2014a*). PPL102 differs in morphology and polarity from the other PPL1 DANs and is likely to perform different functions. For this reason, it has not been included in certain analyses of DANs. All other compartments are innervated by PAM DANs, as illustrated: PAM01, PAM02, PAM03, PAM04, PAM05, PAM06, PAM07, PAM08, PAM09, PAM10, PAM11, PAM12, PAM13, PAM14, and PAM15. Unlike the PPL1 DANs, multiple PAM DANs of the same cell type innervate the same compartment, and in some cases the same compartment has different PAM DAN types innervating different subdomains of the compartment. MB lobes are shown in gray. A single representative neuron is shown for each cell type in magenta, with gray dots indicating postsynaptic sites and yellow dots indicating presynaptic sites. Images showing the identity of the MB lobes are shown in the lower right. The online version of this article includes the following figure supplement(s) for figure 6:

**Figure supplement 1.** Table of cell types found in each MB compartment.
**Figure supplement 2.** MBON neurotransmitter predictions.
**Figure supplement 3.** Synapse number distribution for MBON outputs and DAN inputs.

layer feedforward network employing axo-axonal synapses (Figure 24), and provide input to the dendrites of DANs (Figure 26).

## Atypical MBONs: integrators of information from inside and outside the MB lobes

We identified 14 additional types of MBONs that differ from MBONs previously described in the adult. We refer to these cell types as 'atypical MBONs' in that their dendritic arbors, in addition to having extensive KC input within the MB lobes, extend outside the MB lobes into adjacent brain areas (*Figure 8*, *Figure 8—figure supplements 1–14*, *Figure 8—video 1*, *2*, *3*, *4*, *5*, *6*, *7*, *8*, *9*, *10*, *11*, *12*, *13*, *14*). We reclassified MBON10 and MBON20 as atypical MBONs since these two cell types had significant dendritic arbors outside the MB lobes (*Figure 8*, *Figure 8—figure supplements 1* and *2*, *Figure 8—video 1* and *2*). Twelve of the 14 atypical MBON types innervate the horizontal lobes. Unlike typical MBONs, six of the atypical MBONs have significant innervation in ventral neuropils, in particular the LAL. For each of the atypical MBONs we provide a figure supplement (*Figure 8—figure supplements 1–14*) that provides information on its top inputs and outputs as well as a video (*Figure 8—video 1*, *2*, *3*, *4*, *5*, *6*, *7*, *8*, *9*, *10*, *11*, *12*, *13*, *14*) that displays additional morphological and connectivity features. We used newly developed machine vision methods (*Eckstein et al., 2020*; Methods) to predict the neurotransmitters of these MBONs (*Figure 6—figure supplement 2*), as the specific GAL4 drivers that would be required to determine transmitters by antibody staining or RNA profiling were not available.

As these MBONs are described here for the first time, no experimental data yet exists on their function(s) or physiology. Nevertheless, their connectivity provides clues. Each atypical MBON is poised to integrate information conveyed by KCs with additional inputs to the portion of its dendritic arbor that lies outside the MB lobes. *Figure 8—figure supplement 15* shows which brain regions supply input to each of the atypical MBONs. Frequently, these inputs include the outputs of other MBONs; nine of the 14 atypical MBONs have at least two other typical or atypical MBONs among the top 10 inputs to their dendrites that lie outside the lobes. Some atypical MBONs receive sensory information directly. MBON28 (α′3) receives strong multiglomerular PN input, with three mPNs among its top 10 inputs outside the MB. MBON24 (β2γ5)'s top two inputs outside the MB are putative suboesophageal zone output neurons (SEZONs), likely to convey mechanosensory or gustatory information based on their arbors traced in the FAFB volume (*Otto et al., 2020*; *Zheng et al., 2018*) that contains the full SEZ. One atypical MBON, MBON30 (γ1γ2γ3), is the only MBON that receives significant input directly from the central complex; the nine cells of one fan-shaped body (FB) columnar cell type, FB2-5RUB (FR1), converge in a small brain area called the rubus where they make nearly 500 synapses onto MBON30 (*Figure 8—video 9*).

Such features suggest that the atypical MBONs might be involved in information convergence. Six of the atypical MBONs project to the LAL, positioning them to connect more directly with the motor network than typical MBONs which send their outputs to dorsal brain areas, a feature we explore in detail later in the paper.

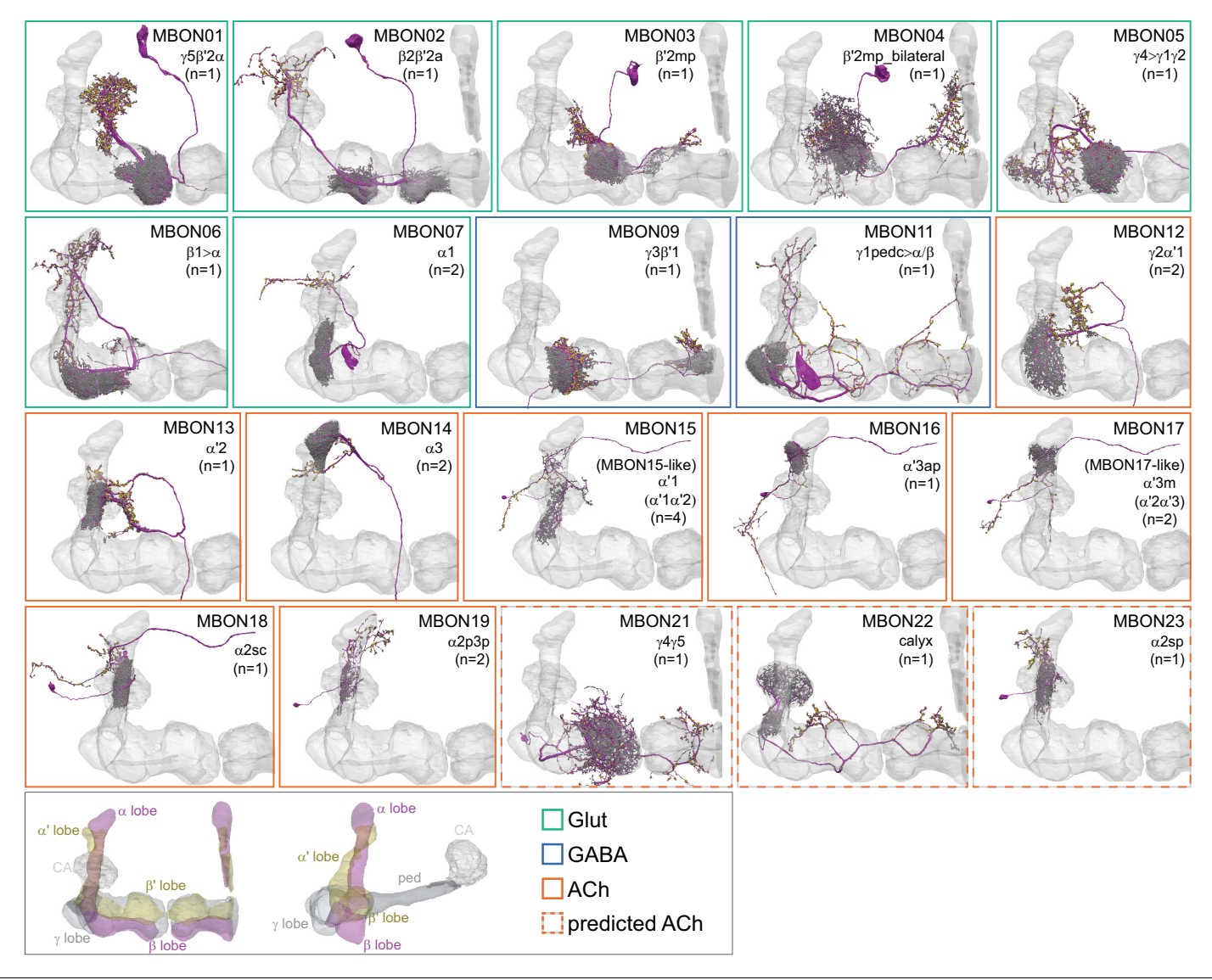

**Figure 7.** Mushroom Body Output Neurons (MBONs). Each panel shows one of the previously described 20 types of MBONs, with its name, the compartment(s) it innervates and the number of cells of that type per brain hemisphere indicated (*Aso et al., 2014a*; *Takemura et al., 2017*); the outline of the MB lobes and CA are shown in gray, in a perspective view from an oblique angle to better display neuronal morphology. MB lobes are shown in gray. A single representative neuron is shown for each cell type (magenta), with gray and yellow dots indicating postsynaptic and presynaptic sites, respectively. The bounding box for each neuron is color-coded by the neurotransmitter used by that MBON; dashed boxes are used where the transmitter type is based on computational prediction (see *Figure 6—figure supplement 2*). The lower panel shows the neurotransmitter color code as well as diagrams of the MB in which the different lobes are indicated; the left diagram is in the same orientation as the other panels. These MBONs are considered to be typical in that their dendritic arbors are confined to the MB lobes. We reclassified MBON10 and MBON20 (*Aso et al., 2014a*) as atypical MBONs since their dendrites extend outside the MB lobes. MBON08, defined by split-GAL4 line MB083C (*Aso et al., 2014a*), was not found in the hemibrain volume. For the other 21 MBON types, we found only minor differences with previous studies (*Aso et al., 2014a*; *Takemura et al., 2017*). For example, MBON15 (α'1) and MBON17 (α'3m), which each were described as having two cells in *Aso et al., 2014a*; *Takemura et al., 2017*, had additional cells in the hemibrain that were similar in morphology, but had some connectivity differences, that we refer to as MBON15-like and MBON17-like. However, since our observations are based on a single individual, we did not split them into separate cell types. Links to the neuPrint records of these MBON types are as follows: MBON01, MBON02, MBON03, MBON04, MBON05, MBON06, MBON07, MBON09, MBON11, MBON12, MBON13, MBON14, MBON15 (including MBON15-like), MBON16, MBON17 (including MBON17-like), MBON18, MBON19, MBON21, MBON22, and MBON23.

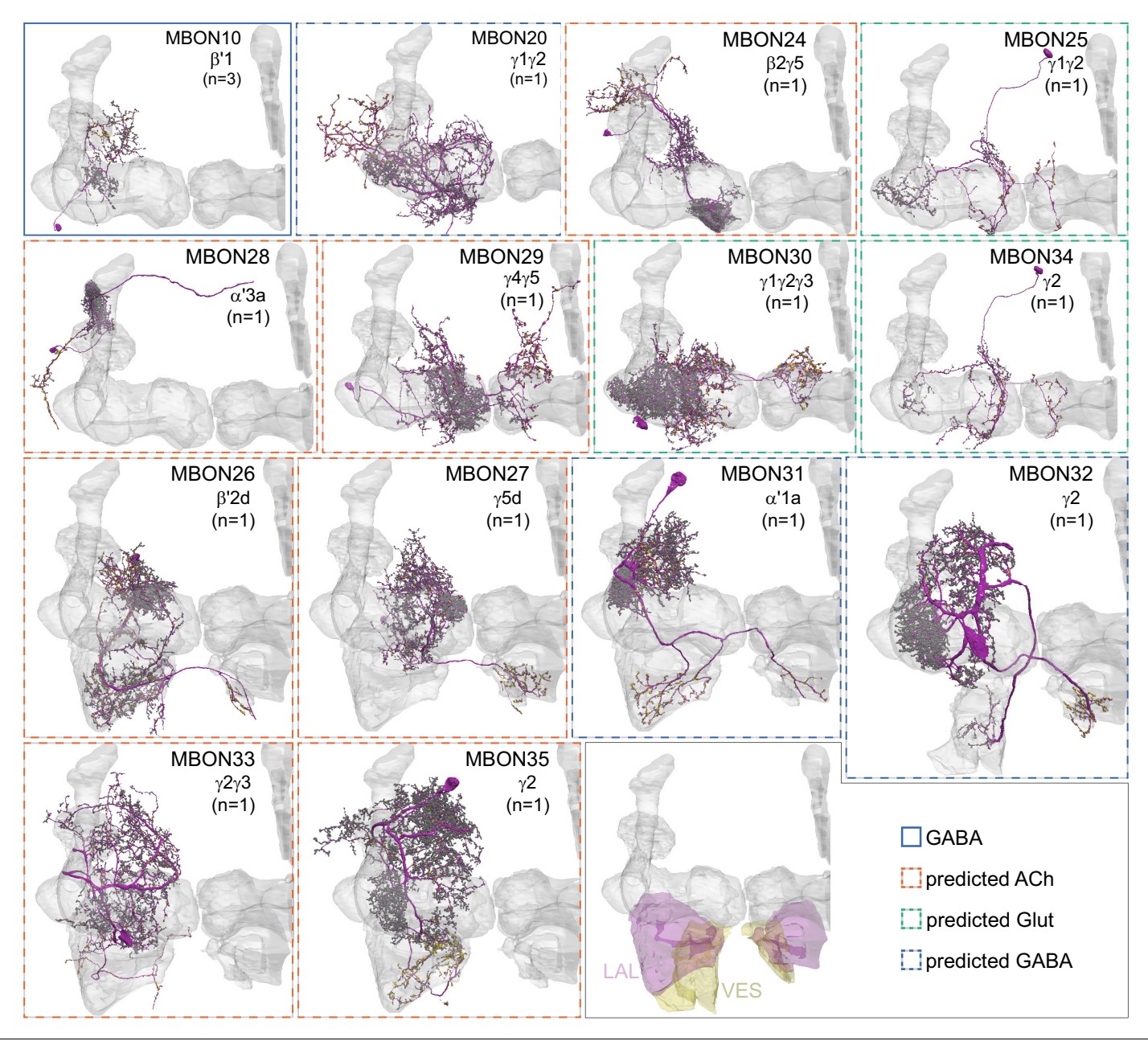

**Figure 8.** Atypical MBONs. Each panel shows one of the 14 types of atypical MBONs, with its name, the compartment(s) it innervates and the number of cells of that type per brain hemisphere indicated. *Figure 6—figure supplement 1* shows which DANs, MBONs, and KCs are found in each compartment. MB lobes are shown in gray and in the bottom right panel the lateral accessory lobe (LAL) and vest (VES) brain areas are highlighted. Neuronal morphologies are shown with dark gray dots and yellow dots indicating postsynaptic and presynaptic sites, respectively. The MBONs shown in this figure are considered to be atypical in that their dendritic arbors are only partially within the MB lobes. Twelve of these types were discovered in the course of the current study. The other two, the GABAergic MBON10 and MBON20, were described by *Aso et al., 2014a*, but we have reclassified them here as atypical MBONs because they have dendrites both inside and outside the MB lobes. The three MBON10s in the EM volume have 69 , 68, and 50% of their postsynaptic sites outside the MB lobes; one is shown here. All the other atypical MBONs occur once per hemisphere. Unlike most typical MBONs that innervate brain areas that are dorsal to the MB, six of the atypical MBONs innervate areas that are ventral to the MB. The LAL is a target of several atypical MBONs, and one also innervates the VES. More detailed information about each of the atypical MBONs can be found in *Figure 8—figure supplements 1–14* and *Figure 8—videos 1–14*. *Figure 8—figure supplement 15* compares the non-MB inputs to these MBONs. The bounding box for each MBON is color-coded by the neurotransmitter used by that MBON; dashed boxes are used where the transmitter type is based on computational prediction (see *Figure 6—figure supplement 2*). Links to the neuPrint records of these MBON types are as follows: MBON10, MBON20, MBON24, MBON25, MBON26, MBON27, MBON28, MBON29, MBON30, MBON31, MBON32, MBON33, MBON34, and MBON35.

*Figure 8 continued on next page*

*Figure 8 continued*

The online version of this article includes the following video and figure supplement(s) for figure 8:

**Figure supplement 1.** Atypical MBON10.
**Figure supplement 2.** Atypical MBON20.
**Figure supplement 3.** Atypical MBON24.
**Figure supplement 4.** Atypical MBON25.
**Figure supplement 5.** Atypical MBON26.
**Figure supplement 6.** Atypical MBON27.
**Figure supplement 7.** Atypical MBON28.
**Figure supplement 8.** Atypical MBON29.
**Figure supplement 9.** Atypical MBON30.
**Figure supplement 10.** Atypical MBON31.
**Figure supplement 11.** Atypical MBON32.
**Figure supplement 12.** Atypical MBON33.
**Figure supplement 13.** Atypical MBON34.
**Figure supplement 14.** Atypical MBON35.
**Figure supplement 15.** Atypical MBON input distribution by brain region and similarity of inputs to different MBONs.
**Figure 8—video 1.** Atypical MBON10.
https://elifesciences.org/articles/62576#fig8video1
**Figure 8—video 2.** Atypical MBON20.
https://elifesciences.org/articles/62576#fig8video2
**Figure 8—video 3.** Atypical MBON24.
https://elifesciences.org/articles/62576#fig8video3
**Figure 8—video 4.** Atypical MBON25.
https://elifesciences.org/articles/62576#fig8video4
**Figure 8—video 5.** Atypical MBON26.
https://elifesciences.org/articles/62576#fig8video5
**Figure 8—video 6.** Atypical MBON27.
https://elifesciences.org/articles/62576#fig8video6
**Figure 8—video 7.** Atypical MBON28.
https://elifesciences.org/articles/62576#fig8video7
**Figure 8—video 8.** Atypical MBON29.
https://elifesciences.org/articles/62576#fig8video8
**Figure 8—video 9.** Atypical MBON30.
https://elifesciences.org/articles/62576#fig8video9
**Figure 8—video 10.** Atypical MBON31.
https://elifesciences.org/articles/62576#fig8video10
**Figure 8—video 11.** Atypical MBON32.
https://elifesciences.org/articles/62576#fig8video11
**Figure 8—video 12.** Atypical MBON33.
https://elifesciences.org/articles/62576#fig8video12
**Figure 8—video 13.** Atypical MBON34.
https://elifesciences.org/articles/62576#fig8video13
**Figure 8—video 14.** Atypical MBON35.
https://elifesciences.org/articles/62576#fig8video14

## Sensory inputs to the KCs: the calyces

Sensory information is conveyed to the dendrites of KCs in specialized MB structures called calyces. The main calyx (CA) contains the dendrites of 90% of the KCs; the dendrites of the remaining KCs are in one of three accessory calyces, each with a specialized function and distinct KC composition. Our work confirms and extends prior descriptions of the calyces (*Aso et al., 2009*; *Bates et al., 2020b*; *Butcher et al., 2012*; *Marin et al., 2020*; *Tanaka et al., 2008*).

### Main calyx (CA)

Olfactory sensory neurons (ORNs) that express the same odorant receptor project their axons to the same glomerulus in the antennal lobe. We typed the ORNs, the antennal lobe projection neurons

(PNs), and their targets in the rest of the hemibrain and have described their full connectivity elsewhere (*Schlegel et al., 2020*). Uniglomerular PNs (uPNs) arborize their dendrites in a single glomerulus and thus receive direct sensory signals from one type of ORN. These olfactory uPNs are the major sensory inputs to the main calyx (*Figure 9A*). PNs branch when they enter the CA and their axons terminate in an average of ~6 synaptic boutons, although the number varies between PN cell types (*Figure 9—figure supplement 1*) and across individuals (*Gao et al., 2019*). The number of boutons per KC also differs between KC cell types in the CA: KCα'/β', 4.40 ± 1.58; KCα/β, 4.67 ± 1.78; KCγ, 8.49 ± 2.17. The synaptic boutons from a single PN can be spread over a large fraction of the calyx. Thermo- and hygrosensory uPNs and multiglomerular PNs (mPNs) can also terminate in the CA, mainly in the anterior (*Figure 9B*). PNs that convey particular odor scenes (for example decaying fruit or pheromones) appear to target specific areas of the CA (*Figure 9C*; *Figure 9—figure supplement 2*). We identified a subset of γm KCs that, while receiving input from olfactory PNs in CA, also extended dendritic claws outside the main calyx to receive gustatory input from a single PN delivering both hygrosensory and gustatory information, providing the first description of a pathway for gustatory information to reach KCs (*Figure 9—figure supplement 3*).

## Ventral accessory calyx

The dendrites of the 99 γd KCs surround the base of the CA in a loose ring and form the ventral accessory calyx (vACA; *Figure 10*). The accessory calyces are thought to be specialized for non-olfactory information, and indeed we found that visual projection neurons (VPNs) from the medulla (ME) and the lobula (LO) are the predominant inputs to the γd KCs. While the LO is largely contained in the hemibrain, the ME is not. In many cases, we were able to confirm predicted ME VPNs by matching neuronal fragments with their complete counterparts in FAFB or light-microscopy images for conclusive annotation of ME VPNs (*Figure 10—figure supplement 1*).

Although the γd KCs respond to light and are required for learning the predictive value of color (*Vogt et al., 2016*; *Vogt et al., 2014*), we have little definitive insight into the type of visual information conveyed by their VPN inputs. In bees, ME and LO VPNs convey specific chromatic, temporal, and motion features, including sensory information required for associations bees make during foraging tasks (*Paulk and Gronenberg, 2008*). The largest group of VPNs in *Drosophila* are ipsilateral, unilateral ME neurons. The ME VPNs have dendrites in the outer part of the ME (up to layer M8); the LO VPNs primarily arborize in the deeper layers of the LO (Lo4-Lo6). These layer patterns are consistent with a possible role in conveying information about color and intensity but notably exclude optic lobe regions that are strongly associated with motion vision such as the lobula plate, ME layer M10 and LO layer Lo1 (*Borst, 2014*). The LO VPNs are also clearly distinct from the well-studied lobula columnar cells which respond to visual features such as visual looming or small moving objects (*Wu et al., 2016*). The optic lobe has a retinotopic organization. Several ME and LO VPNs have arbors that are restricted to parts of the ME and LO and thus are predicted to preferentially respond to stimuli in different parts of the visual field. However, we did not observe evidence for a high-resolution spatial map formed by KC inputs. Local visual interneurons (LVINs) that do not themselves arborize in the optic lobe but are downstream of VPNs convey additional visual information (*Figure 10A*). The connections from VPNs and LVINs onto KCγd dendrites are spatially segregated (*Figure 10B*). LVINs are discussed in more detail below.

## Dorsal accessory calyx

The dendrites of the 60 α/βp KCs define the dorsal accessory calyx (dACA) and receive predominantly visual input (*Figure 11A*). We found that the VPNs that directly connect to α/βp KCs come mostly from the ME (*Figure 11*), with a much smaller contribution from the LO; VPNs projecting from the LO to the dACA have also been recently noted by *Li et al., 2020*. However, unlike in the vACA, indirect visual input conveyed by LVINs outweighs direct VPN input (*Figure 11A*), and VPN and LVIN inputs are less segregated (*Figure 11B*). Among these LVINs, a cluster of 13 morphologically similar neurons (SLP360, SLP362 and SLP371) contributes over 50% of input from all LVINs. One LVIN, MB-CP2, which has been suggested to integrate multi-sensory inputs (*Zheng et al., 2018*), is the single strongest dACA input neuron (*Figure 11—figure supplement 2E*) and also seems to relay input from the subesophageal zone (SEZ).

As described above, local visual interneurons (LVINs) make up a substantial portion of the inputs to γd KCs (*Figure 10—figure supplement 2*) and α/βp KCs (*Figure 11—figure supplement 1*) in the vACA and dACA, respectively. LVINs get input from multiple VPNs, as well as nonvisual inputs, and then convey this integrated information to KCs. The neuronal morphologies and connectivity patterns of the most strongly connected LVINs are shown in *Figure 10—figure supplements 3–5* for the vACA and *Figure 11—figure supplements 2,3* for the dACA. We observed that clusters of LVINs, or sometimes single LVINs, receive input from distinct subpopulations of VPNs. Moreover, some of the LVINs that receive inputs from similar VPN subpopulations tend also to receive similar nonvisual inputs (*Figure 11—figure supplements 4,5*). These observations suggest that, rather than simply relaying visual information, LVINs may perform more complex processing, including integration of visual and nonvisual signals.

### Lateral accessory calyx

The lateral accessory calyx (lACA) is a small subcompartment of the CA innervated by 14 α′/β′ap1 KCs (*Yagi et al., 2016*) and one KCγs2 (*Figure 12*). The lACA, which has been recently described in detail in *Marin et al., 2020*, is thought to be a thermosensory center as >90% of its input comes from two PNs: the slow-adapting, cooling air-responsive PN, VP3+ vPN (*Frank et al., 2015*; *Liu et al., 2015*; *Stocker et al., 1990*), which solely targets the lACA (*Jenett et al., 2012*), and the predicted warming air-responsive VP2 adPN (*Marin et al., 2020*). Other inputs to the lACA are described in *Figure 12* and the inputs to KCγs2 are separately detailed in *Figure 12—figure supplement 2*. Most KCs in the lACA also receive inputs in the CA from olfactory and thermo/hygrosensory PNs. There are direct connections to DN1a clock neurons within the lACA (*Alpert et al., 2020*; *Marin et al., 2020*; *Yagi et al., 2016*), as well as from the DN1a neurons and the aMe23/DN1-like neuron to the 5th s-LNv and one LNd (*Figure 12—figure supplement 1*), which could play a role in entrainment of the circadian clock by temperature (*Figure 12—figure supplement 2*) or the adjustment of sleep patterns to different temperatures (*Yadlapalli et al., 2018*; *Alpert et al., 2020*). These connections appear to reflect a function of the lACA that is distinct from its role as a site of thermosensory inputs to KCs.

## Randomness and structure in sensory inputs to KCs

Sensory input to the MB calyces shows clear structure across modalities, with visual VPN/LVIN and thermo/hygrosensory PN input targeted to specific KC types (as described above and discussed more fully below). This raises the question of whether olfactory inputs, in particular from uPNs, also exhibit structure in their inputs to the KCs. PN synapses onto KCs in the CA have a characteristic structure in which each bouton is surrounded by a claw-like KC process (*Leiss et al., 2009*; *Yasuyama et al., 2002*; *Figure 9—figure supplement 1F*; *Figure 9—video 1*), which are strikingly reminiscent of the mossy fiber-granule cell synapses found in the vertebrate cerebellum (*Huang et al., 2013*). Each KC has an average of 5.6 dendritic claws in the CA and requires simultaneous inputs from a combination of PNs to spike (*Gruntman and Turner, 2013*). The synapses between the VPNs and the KCγd dendrites have a more typical morphology, lacking the claw-like structure seen in PN-to-KC synapses in the CA (*Figure 10—video 1*).

Previous work suggested that KCs sample olfactory uPNs without apparent structure in both the larva (*Eichler et al., 2017*) and the adult (*Caron et al., 2013*), but a recent analysis of the FAFB EM dataset identified convergence of specific PNs that was inconsistent with random sampling (*Zheng et al., 2020*; see also *Gruntman and Turner, 2013*). Developmental mechanisms have the potential to bias PN-KC connections. Both PN and KC cell types are generated in a highly stereotyped developmental order (*Lee et al., 1999*; *Yu et al., 2010*), and PNs flexibly adjust the number of their presynaptic boutons based on the availability of KC dendrites (*Elkahlah et al., 2020*).

To look for potential structure in olfactory uPN inputs to KCs, we computed a binary uPN-to-KC connection matrix using a threshold of five synapses. We performed principal components analysis (PCA) on this connectivity, which can provide indications of structure (*Caron et al., 2013*; *Eichler et al., 2017*), and compared the results with PCA on synthetic connectivity matrices constructed by assuming KCs randomly sample their inputs in proportion to the total number of KC connections formed with each uPN (*Figure 13A*). Three principal components (PCs) are clearly larger than the corresponding values in the random model. Much of this deviation from the random model

is due to differential sampling of olfactory glomeruli by γ, α′/β′, and α/β KCs. In particular, input to α′/β′ and α/β KCs is more strongly skewed toward specific highly-represented glomeruli, most notably DP1m and DM1, while the distribution for γ KCs is more uniform (*Figure 13B*). This suggests that the random model should be extended to allow for uPN connection probabilities that depend on both the uPN and KC types. However, some deviations from the extended random model are still present in the data, as can be seen when PCA is performed on lobe-specific connectivity matrices (*Figure 13A*). Note, for example, the first PC for the α/β KCs (also see *Figure 13—figure supplements 1,2*).

We reasoned that this residual structure might arise from the spatial organization of inputs in the CA, so we analysed the spatial arrangement of uPN-to-KC connections and its impact on the KC odor representation. We determined the centroid locations of uPN axonal boutons within the CA by using spatial clustering of PN-to-KC synapses. Boutons belonging to uPNs from the same glomerulus were nearer, on average, than those from different glomeruli (*Figure 14A*). From these distributions, we computed the average number of boutons within a given radius of each centroid (*Figure 14B*). These neighboring boutons were used to construct models with PN-to-KC connectivity randomly shuffled within a specified radius r (*Figure 14C*). This produces models in which large-scale organization (at spatial scales greater than r) is preserved, while local organization (at spatial scales less than r) is random (note that the model and the data are identical for r = 0). We computed statistics that quantified properties of the KC representation for our shuffled models, as a function of r. The first is the participation ratio of the PN-to-KC weight matrix, which quantifies how uniformly represented each glomerulus is across the inputs to all KCs. The second statistic is the dimension of the KC representation in a model in which KCs fire sparsely in response to odors that activate random patterns of PNs. Previous work has shown that this quantity determines the ability of a linear readout of KC activity to perform odor discrimination (*Litwin-Kumar et al., 2017*). Our analysis reveals that the participation ratio and dimension are lower for the true data than for the shuffled models, as expected from non-random structure, although the effect is modest (*Figure 14D*). Noticeable effects are present when the length scales for random shuffling is greater than approximately 10 μm. The effect is strongest for α/β and α′/β′ KCs, while the effect of spatial organization of the γ KC inputs appears to be minimal.

This analysis indicates that uPN inputs depend on KC subtype and on the spatial organization of connections within the main calyx due to locally restricted sampling. This spatial structure has only a modest effect on the dimensionality of the KC olfactory representation for simulated odors that activate random ensembles of PNs, and its effect on KC inputs does not noticeably persist at the level of KC outputs to MBONs (*Figure 13—figure supplement 2*). Nevertheless, it might be relevant for specific odor categories. More broadly, our analysis of sensory inputs to KCs reveals clear specialization of function across modalities supported by segregated modality-specific connectivity within the accessory calyces as well as non-olfactory input to the main calyx (see *Figure 15—figure supplement 1A*).

## Segregation of information flow through the MB

If the segregation of information seen in the KCs, especially across sensory modalities, is maintained at the level of MB output, this could have important implications for the specificity of learned associations. To explore this possibility, we first computed the fraction of inputs that KC types receive from PNs conveying different sensory modalities (*Figure 15—figure supplement 2A*) and then we computed the fraction of input to each MBON provided by KCs specialized for different sensory modalities (*Figure 15—figure supplement 2B,C*). The PN-to-KC input fractions (*Figure 15—figure supplement 2A*) quantify the specialization of KCs for particular sensory modalities as described above. For example, α/βp, γd, and γs1 KCs receive more than 90% of their input from VPNs and LVINs, while γs2,3,4 KCs receive more than 50% of their input from thermo-hygrosensory PNs. From this information, we divided KCs into three groups: 1664 olfactory KCs, 102 olfactory + thermo/hygrosensory KCs, and 161 olfactory + visual KCs. We then computed the fraction of KC input to each of the MBONs that came from each of these three KC groups (*Figure 15—figure supplement 2B,C*; this connectivity is also shown more finely divided into KC types in *Figure 15—figure supplement 3*). Typical MBONs innervating the α′/β′ lobes show a gradation of input from KCs that receive thermo/hygrosensory inputs, with MBON16 (α′3ap) having 36% of its effective input coming from this source. Typical MBONs of the α/β lobes similarly show varying degrees of KC input of the

olfactory + visual class. Typical MBONs of the γ lobe and CA are driven predominantly by olfactory KCs. Atypical MBONs are more strongly innervated by KCs with non-olfactory input.

By multiplying matrices of PN-KC and KC-MBON connectivity fractions, we computed an effective PN-to-MBON connectivity (*Figure 15*, *Figure 15—figure supplement 1B,C*). This effective connectivity shows a surprisingly high amount of non-olfactory sensory input to the MBONs, with some MBONs predominantly devoted to non-olfactory modalities. While the majority of the effective input to the typical MBONs is olfactory, 16% of the effective input to MBONs innervating the α/β lobes, and 9% for the γ lobe MBONs, is visual, with one of the two MBON19 (α2p3p)s having 62% visual effective input (*Figure 15*, *Figure 15—figure supplement 1*). Thermo-hygrosensory PNs constitute 7% of the effective input to the α'/β' MBONs. Compared to typical MBONs, atypical MBONs have a higher fraction of their input from non-olfactory sources: 24% of the effective input to atypical MBONs innervating the α'/β' lobes is thermo-hygrosensory, and 26% of the effective input to γ-lobe atypical MBONs is visual. In particular, MBON26 (β'2d) has 30% thermo-hygrosensory effective input, and MBON27 (γ5d) has 80% visual effective input. Thus, these MBONs may preferentially participate in non-olfactory or multimodal associative memories.

## Downstream targets of MBONs

The preceding analysis revealed segregated pathways through the MB that carry distinct sensory signals. We next investigated the extent to which these pathways remained segregated in the outputs of the MBONs by comparing the similarity of PN inputs between pairs of MBONs with the similarity of their output targets. For this purpose, we used cosine similarity, which measures the alignment between the inputs (or outputs) of the two neurons, without being affected by the total number of synapses they make. It also takes into account synapse counts, so that stronger connections influence the measure more than weaker connections, without using any arbitrary cutoff threshold. If two neurons have no shared partners, their cosine similarity will be 0, and if they target the exact same partners with the exact same relative strength, their cosine similarity will be 1.

The similarity of the PN input to pairs of MBONs reflects the selectivities seen in *Figure 15* as well as revealing some more subtle structure (*Figure 16*, left panel). These similarities are, at least to some extent, preserved at the output level (*Figure 16*, right panel), particularly for the similarities among MBONs innervating the same lobes (*Figure 16—figure supplement 1*; results are quantified in *Figure 16—figure supplement 2*). This suggests that the parallel processing of different sensory modalities is, in some cases, preserved all the way from the PNs to the downstream targets of the MBONs. Beyond this organization, unbiased clustering of MBONs based solely on their output connectivity reveals additional groups of MBONs that have similar downstream targets but different effective PN input (*Figure 16—figure supplements 3,4*). For example, MBON06 (β1>α), and MBON18 (α2sc) innervate different dendritic compartments (*Figure 16—figure supplement 4*) but have a high cosine similarity score based on their output (*Figure 16—figure supplement 3*). Conversely, MBON20 (γ1γ2) and MBON25 (γ1γ2) have low cosine similarity based on their output although their dendrites innervate the same compartments.

To look for other factors underlying the patterns of MBON output, we considered the sign of MBON output as implied by neurotransmitter identity. Acetylcholine is highly predictive of excitation, as is GABA for inhibition. The situation for glutamate is more ambiguous. There are established cases of glutamate having inhibitory (*Liu and Wilson, 2013*) and excitatory (*Johansen et al., 1989*) action. Most cells appear to express both inhibitory and excitatory receptors for glutamate; for example, all six MBON and all ten DAN cell types for which RNA profiling data exists express both inhibitory GluClalpha and excitatory NMDA type receptors (*Aso et al., 2019*). Activation of cholinergic and glutamatergic MBONs has been associated with opposing behaviors, avoidance and approach, respectively (*Aso et al., 2014b*; *Owald et al., 2015*; *Perisse et al., 2016*), but the extent to which these result from the action of these MBONs on shared targets has not been established.

The degree of overlap in outputs between pairs of MBONs organized by neurotransmitter type (*Figure 17*) reveals structure that can be summarized by grouping MBONs by their transmitters (*Figure 17—figure supplement 1*). Among typical MBONs, both cholinergic and glutamatergic MBONs show similarity in their outputs with other MBONs of the same transmitter type but, interestingly, there is a matching degree of similarity in the outputs across cholinergic and glutamatergic MBONs. This is not seen in the similarities between MBONs using either of these transmitters and GABAergic MBONs (*Figure 17—figure supplement 1*). Thus, assuming an inhibitory role for glutamate, the

convergence of cholinergic and glutamatergic MBONs on a common downstream target is likely to produce a 'push-pull' effect, in which competing excitatory and inhibitory influences on the common target drive opposite behaviors. Among atypical MBONs, neither the push-pull effect nor the disproportionate similarity in the outputs of cholinergic MBONs is observed.

## The distribution of the MBON outputs differs between MBONs using different neurotransmitters

The analyses above reveal MBONs with similar downstream targets but do not identify those targets. We computed the propensity of MBONs, grouped by neurotransmitter type, to project to different brain regions (*Figure 18*); our results confirm and extend observations made by light microscopy (*Aso et al., 2014a*). All MBON types project strongly to the CRE, while cholinergic and glutamatergic MBONs (but not GABAergic MBONs) project strongly to the SMP, and to a lesser extent the SIP and SLP. This pattern suggests that the CRE, SMP, SIP, and SLP may be sites of 'push-pull' interactions as described above. GABAergic MBONs are notable in that the vast majority of their projections are to the CRE, and several atypical MBONs are unique in their strong projections to the LAL. Overall, these results suggest strong biases in output connectivity patterns of MBON neurotransmitter types, viewed at the coarse level of brain area. Finer structure is also observable in the data. For instance, the downstream projection targets of MBONs exhibit spatial biases even within individual brain areas (*Figure 18—figure supplement 1*). Importantly, although MBONs of a given neurotransmitter type often exhibit related connectivity biases, they are certainly not homogeneous, consistent with unique behavioral roles for individual MBONs (*Figure 18—figure supplement 2*).

## Neuronal pathways connecting the MB and the CX

The MB and central complex (CX), both highly structured centers, are known to carry out sophisticated computations. Communication from the MB to the CX is likely to be important for conveying information about learned associations of sensory cues and external rewards or punishment (*Aso et al., 2014a*; *Aso et al., 2014b*; *Owald et al., 2015*), novelty (*Hattori et al., 2017*) and sleep need (*Sitaraman et al., 2015a*; *Dag et al., 2019*), which, in turn, are expected to influence navigation, sleep, and other activities governed by the CX. Recent transsynaptic tracing experiments have suggested the presence of connections from a few MBONs to the CX (*Scaplen et al., 2020*). We describe here the complete network of strong (based on synapse number) neural pathways connecting the MB to the CX. These connections are also discussed in a companion manuscript on the connectome of the CX (*Hulse et al., 2020*).

We found that direct communication from the MB to the CX is limited to two pathways (*Figure 19*). The most prevalent is MBONs connecting to the dendrites of fan-shaped body (FB) tangential neurons; 22 out of 34 MBON types make such direct connections and both typical and atypical MBONs participate. In addition, three atypical MBONs make connections in the LAL to the dendrites of three CX cell types that have axonal terminals in the nodulus (NO) (*Figure 19—figure supplement 1A*). The FB has been divided by anatomists into multiple layers, numbered ventrally to dorsally from one to nine (*Figure 20*; *Figure 20—video 1*). Each tangential neuron's arbors within the FB lie predominantly in a single layer and its dendrites project laterally in the CRE, SMP or SIP (morphologies are shown in *Figure 20—video 1*). We found that MBONs preferentially target FB layers four (FBl4) and five (FBl5). Fifteen MBON types target FBl4 and account for 62% of all MBON to FB synapses; they innervate 58% of FB cell types found in this layer. Thirteen MBON types target FBl5 and account for 30% of all MBON to FB synapses; they innervate 43% of FB cell types found in this layer (*Figure 20*). Thirty-seven different FB cell types get direct synaptic input from MBONs, and some cell types get input from multiple MBON cell types; the fraction of each cell type's input that comes from MBONs is shown in *Figure 19*. One cell type, CREFB4, has an unusually high level of MBON input, receiving nearly 20% of its synaptic input from a combination of MBON09 (γ3β′1) and MBON21 (γ4γ5) (*Figure 19—figure supplement 1B,C*). Among the MBONs that output onto FB neurons, MBON21 (γ4γ5), MBON33 (γ2γ3), and MBON34 (γ2) devote the highest fractions of their synaptic output connecting to FB neurons, at 16, 9, and 8%, respectively.

We also looked at MBON to CX connections that are mediated by a single interneuron. We separately determined strong connections, where we required at least 50 synapses from the MBON to

the interneuron and then at least 50 synapses from the interneuron to the CX neuron, as well as weaker connections where these thresholds were set at 20 synapses. For these thresholds, we found that the only connected CX neurons were FB tangential neurons and NO neurons (eight and two at thresholds of 20 and 50, respectively). We found that all 34 MBON types output to FB neurons when the threshold of 20 synapses at each connecting point was used (*Figure 20B*). Moreover, for this threshold all FB layers except layer 9 receive indirect input from MBONs. Increasing the threshold to 50 synapses results in a connectivity pattern that, at the level of which FB layers are connected, is very similar to that displayed by direct connections (compare *Figure 20* panels A and C). The direct and indirect connections appear to target similar FB neurons. Over 90% of FB cell types that are directly postsynaptic to MBONs also get indirect input through a single interneuron; conversely, 76% of FB cell types connected by an interneuron, when using the 50-synapse threshold, also get direct connections. A single interneuron can get input from multiple MBONs and then itself make output onto multiple FB cell types. *Figure 20—figure supplement 1* and *Figure 20—videos 2– 4* show examples of this circuit motif.

Connections from the CX to the MB are much more limited. In particular, we identified no cases where CX neurons provide direct input to MB intrinsic cell types such as KCs, APL, or DPM. Indirect connections are also rare. We did identify three cases where CX neurons were upstream of DANs. FB tangential neurons often have mixed arbors outside the FB and we found layer 6 neurons that are presynaptic to PAM09 (β1ped) and PAM10 (β1) and layer 7 neurons that are presynaptic to PPL105 (α′2α2) (see Figure 34). In addition, one of the 11 PAM12 (γ3) neurons (862705904) is atypical in that it receives 12% of its input from the FB columnar neuron FB2-5RUB (*Figure 29—video 3*). This same FB2-5RUB (FS1) cell type makes strong inputs onto the atypical MBON30 (γ1γ2γ3); with 451 synapses, it is MBON30's strongest input outside the MB lobes (see *Figure 8—figure supplement 9*, *Figure 8—video 9* and *Figure 25—figure supplement 1*).

## A network of convergence downstream of MBONs

We found that different MBONs frequently share downstream targets resulting in an extensive network of convergence. Such convergence would provide a mechanism to integrate the outputs from different MB compartments. At a threshold of 10 synapses, ~1550 neurons are downstream targets of at least one MBON, with nearly 40% of these (~600 neurons), getting input from at least two MBON types (*Figure 21A*). Downstream targets with multiple inputs are not unexpected given that the total of all MBON downstream targets is close to 10% of central brain neurons and individual MBONs tend to have many downstream partners (an average of 57 at the threshold we used). However, a null model in which MBON outputs are randomly sampled by central brain neurons, with weighted probabilities according to their total number of input synapses, yields ~3200 downstream targets, only 14% of which receive input from at least two MBON types. This result indicates that MBON outputs are convergent beyond what would be expected from random connectivity alone. Both typical (*Figure 21B*) and atypical (*Figure 21B′*) MBONs participate in such convergent networks at roughly similar levels, after correcting for the smaller number of atypical MBONs, and individual cells often get input from both types of MBONs. As discussed below, MBONs are highly overrepresented among the downstream targets of other MBONs. Lateral horn (LH) neurons and lateral horn output neurons (TON) are also frequent targets of MBON convergence, being particularly overrepresented among those receiving input from seven or more MBONs; for further details please see *Schlegel et al., 2020*. *Figure 21—figure supplements 1,2* show examples of convergence neurons with 7 and 11 upstream MBONs, respectively, and include MBONs with neurotransmitters of opposite sign, extreme examples of the 'push-pull' arrangement introduced in *Figure 17*.

## Three MBON types support a multilayer feedforward network in the MB lobes

MBON05 (γ4>γ1γ2), MBON06 (β1>α), and MBON11 (γ1pedc>α/β) have been previously shown, based on light microscopic data, to have a high fraction of their output directed to the dendrites of other MBONs within the MB lobes, providing pathways for a multi-layered feedforward MBON network (*Aso et al., 2014a*). All three of these MBONs are putatively inhibitory based on their use of GABA or glutamate as neurotransmitters. We have now been able to map this network comprehensively and to look at the spatial distribution of the synapses from the three feedforward MBONs to

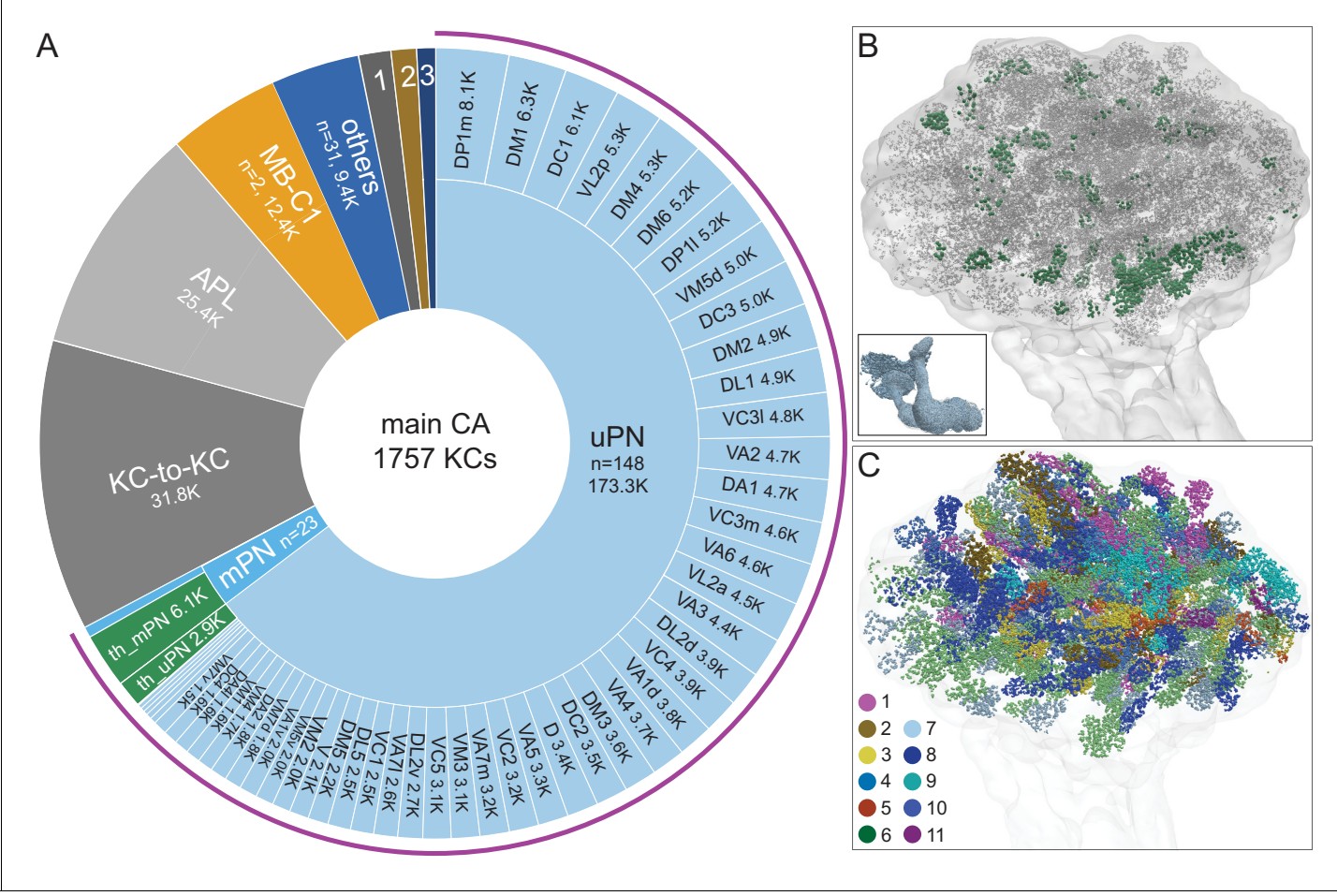

**Figure 9.** Main calyx (CA). The dendrites of 1757 KCs of the α/β, α'/β', and γ cell types define the CA. (**A**) The pie chart shows a breakdown of the inputs to these KCs. The largest source of input is from 129 uniglomerular olfactory projection neurons as judged by synapse number (uPNs; 63.6% of total input to the KCs); the number of synapses is indicated for each uPN cell type. Additional olfactory sensory input is provided by 23 multiglomerular projection neurons (mPNs). Information about temperature is provided by both 35 mPNs (th_mPN) and 19 uPNs (th_uPN). The next most prominent inputs are KC-to-KC synapses within the CA (11.9%), from APL (9.5%; see *Figure 3—figure supplement 1A*) and from MB-C1 (4.6%; see *Figure 3—figure supplement 1H*). Smaller sources of input are indicated by the numbered sectors: 1, a group of nine neurons previously described as 'centrifugal' neurons (*Bates et al., 2020a*) that innervate both CA and LH (1.3%). 2, MB-CP2 (1.0%); 3, PPL201 (see *Figure 3—figure supplement 1G*). The remaining 3.4% is provided by 31 other neurons (blue). (**B**) An image of the CA showing the locations of olfactory PN and thermo PN synapses onto KCs. The green dots representing thermo PN olfactory input synapses are of larger diameter to allow better visibility in the presence of the larger number of gray dots representing olfactory input synapses. Note the thermo PN inputs are located in the anterior and at the periphery of the CA, corresponding to the position of α'/β'ap1 and γt KC dendrites. The inset shows the orientation of the image. (**C**) Inputs from olfactory PNs are shown color-coded based on the type of olfactory information they are thought to convey (see *Bates et al., 2020b*): 1, fruity; 2, plant matter; 3, animal matter; 4, wasp pheromone; 5, insect alarm pheromone; 6, yeasty; 7, alcoholic fermentation; 8, decaying fruit; 9, pheromonal; 10, egg-laying related; 11, geosmin.

The online version of this article includes the following video and figure supplement(s) for figure 9:

**Figure supplement 1.** Distribution of the termini of olfactory PNs in the CA.

**Figure supplement 2.** Spatial arrangement in the CA of synaptic input from different PN groups.

**Figure supplement 3.** Gustatory input to a subset of KCs.

**Figure 9—video 1.** Introduction to γ main KCs.

https://elifesciences.org/articles/62576#fig9video1

their targets. It has been noted that in some cases, such synapses are clustered close to the root of the target MBON's dendrites where they would be well positioned to cause a shunting effect (*Perisse et al., 2016*; *Takemura et al., 2017*; *Felsenberg et al., 2018*). We examined all feedforward connections between MBON05 (γ4>γ1γ2), MBON06 (β1>α), and MBON11 (γ1pedc>α/β) and

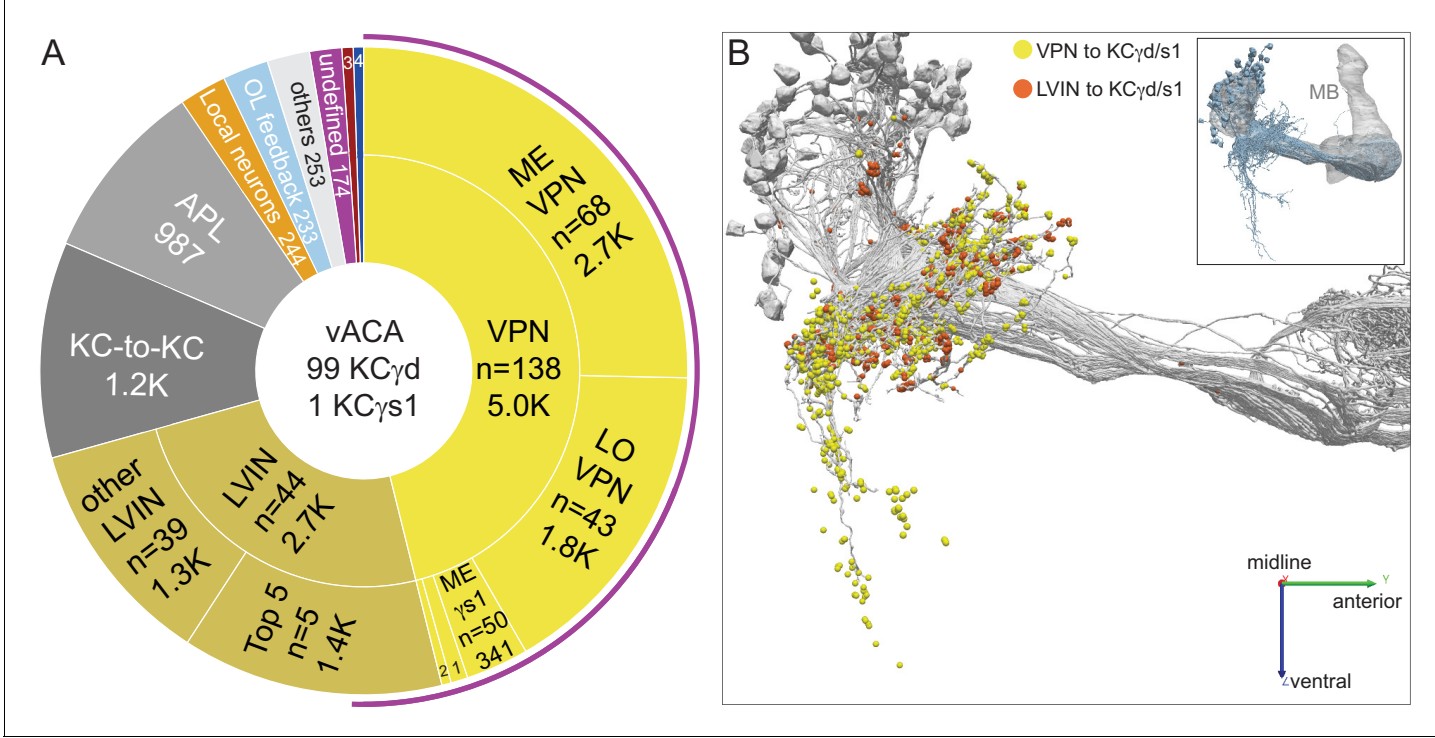

**Figure 10.** Ventral accessory calyx (vACA). The dendrites of the 99 γd and one γs1 KCs define the vACA. (**A**) The pie chart shows a breakdown of the inputs to these KCs; the number of cell (n=) and the number of the total synapses contributed by the cells in that sector are shown without applying a threshold. The majority of inputs convey visual information, either directly from visual projection neurons (VPNs; 46.1%) or through intermediate local visual interneurons (LVIN; 24.6%) that themselves receive input from VPNs. The number of VPNs shown in the pie chart counts VPNs that make as few as one synapse. When a threshold is applied that requires a VPN to make at least five synapses to a single KC, then we find 49 VPNs, including 26 ME VPNs and 21 LO VPNs. The synapses from the VPNs and the LVINs onto the KCγd dendrites do not show the claw-like structure seen in the CA (*Figure 10—video 1*). A ranking of LVINs based on the amount of visual input conveyed is shown in *Figure 10—figure supplement 2A*. More than half of the indirect input is mediated by five LVINs (Top 5), which are shown in *Figure 10—figure supplement 3B*. VPNs can be subdivided based on the location of their dendrites in either the medulla (ME) or lobula (LO), as indicated in the outer circle. There are 68 VPNs that connect to the single KCγs1, with a total of 483 synapses: 50 from the ME, 34 of which are shared with other γd KCs, and 14 from the LO, eight of which are shared with other γd KCs (represented by the numbered sector 1). The next most prominent inputs to KCs in the vACA are synapses between the KCs themselves (10.8%), from APL (9.1%), from local interneurons that do not appear to convey significant visual information (2.3%), from interneurons that send feedback from the vACA to optic lobe neurons (OL feedback; 2.2%) and neurons that leave the volume with undefined identity (undefined; 1.6%). Other sources of input are indicated by the other numbered sectors: 2, other VPN input that we could not classify as from the ME or LO, due to incomplete morphology (0.9%); 3, three putative mPNs (0.6%) (5813063239, 1442819296, 5813040515); 4, three putative SEZ cells (0.5%). The remaining 2.3% is provided by 253 interneurons that are weakly connected to these KCs, with each providing one synapse to each of less than 4 KCs (others). The fraction of input to the vACA KCs conveying visual information is indicated by the outer purple arc; it reflects the direct input from the VPNs plus the fraction of the LVIN input that represents visual input. (**B**) Synaptic connections from visual projection neurons (VPN) and local visual interneurons (LVIN) onto γd and γs1 KCs (gray), color-coded. Note the different spatial distribution of synapses from VPNs and LVINs. VPNs make synapses onto KCγd dendrites in an area ventral to the CA, previously recognized as the vACA (*Butcher et al., 2012*), as well as in a diffuse ring surrounding the base of the CA; synapses from LVINs are restricted to the ring. Additional views are shown in *Figure 10—figure supplement 4A, B*.

The online version of this article includes the following video and figure supplement(s) for figure 10:

**Figure supplement 1.** Identification of VPNs.

**Figure supplement 2.** LVINs ranked by amount of visual information conveyed.

**Figure supplement 3.** Further description of LVINs upstream of vACA KCs.

**Figure supplement 4.** Additional views of the morphologies of VPN and LVIN inputs to the vACA.

**Figure supplement 5.** Distribution of VPN inputs onto an LVIN.

**Figure 10—video 1.** Introduction to γ dorsal KCs.

https://elifesciences.org/articles/62576#fig10video1

their target MBONs and found that the same upstream neuron often has distinct synapse distributions on its different downstream MBONs (*Figure 22*), as previously observed for MBON11 (*Perisse et al., 2016*; *Felsenberg et al., 2018*). In several cases in the α lobe where such

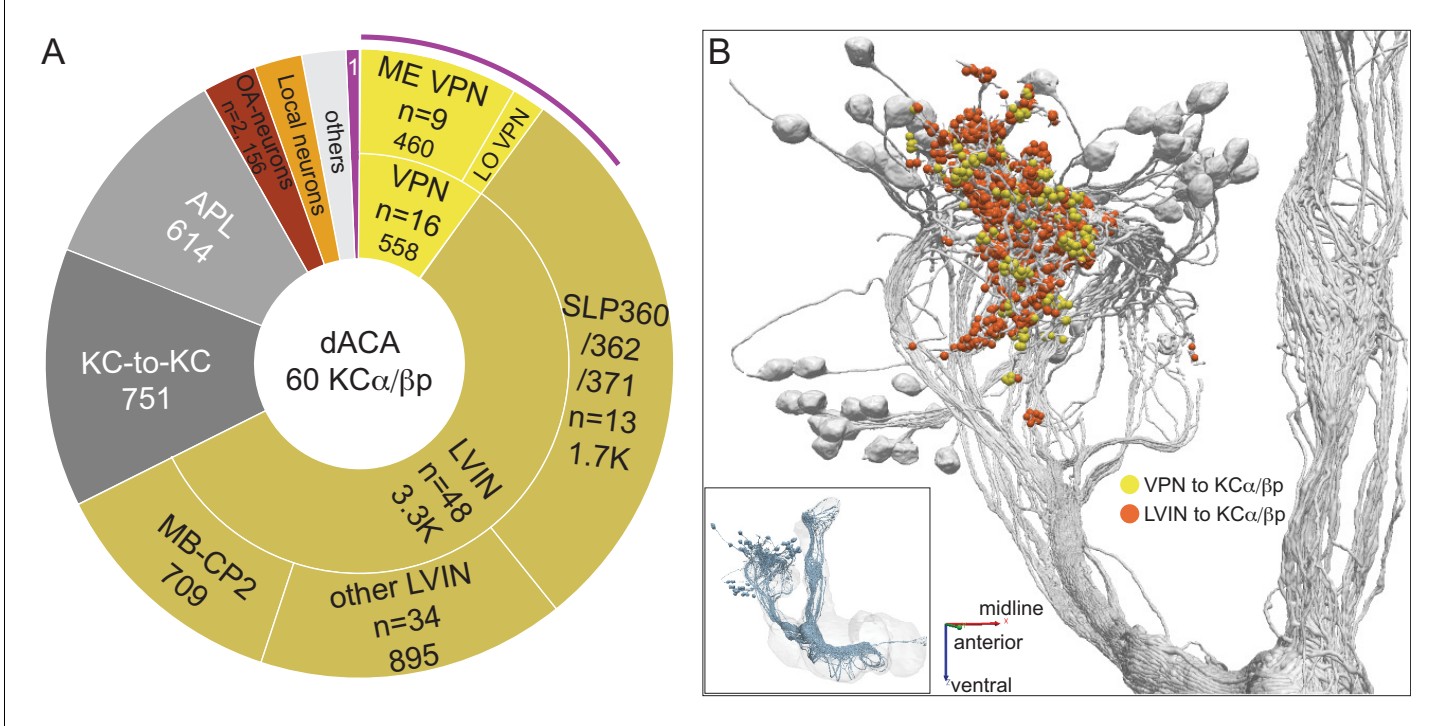

**Figure 11.** Dorsal accessory calyx (dACA). The dendrites of the 60 α/βp KCs define the dACA. The pie chart shows a breakdown of the inputs to these KCs. The majority convey visual information, either directly from visual projection neurons (VPN; 9.8%) or through intermediate local visual interneurons (LVIN; 57.8%) that receive input from VPNs (see *Figure 11—figure supplement 1*). VPNs can be subdivided based on the location of their dendrites in either ME or LO, as indicated in the outer circle. More than two-thirds of the indirect input is mediated by the LVIN cell types SLP360, SLP362, and SLP371, shown in *Figure 11—figure supplement 2C*; this SLP360/361/371 cluster of 13 neurons contributes about 30% of total input to the 60 α/βp KCs in the dACA. Neurons of similar morphology have also been observed to be presynaptic to KCα/βp in the dACA in a recent study (*Li et al., 2020*). Another LVIN, MB-CP2 (LHPV3c1) (479935033), provides 12.6% of the input to KCs in the dACA; however, only a small percentage of its inputs are visual (see *Figure 11—figure supplement 1A* and *2E*). The total visual information presented to KCs by VPNs and LVINs is indicated by the purple arc around the outer layer; it reflects the direct input from the VPNs plus the fraction of the LVIN input that represents visual input. The next most prominent inputs are KC-to-KC synapses in the dACA (13.3%), from APL (10.9%), from two octopaminergic neurons (2.8%; OA-VPM3, see *Figure 3—figure supplement 1D*, and OA-VUMa2, see *Figure 3—figure supplement 1F*); and local interneurons (n = 23; 2.3%). Remaining input, 'others', are input from 102 different neurons that are all weakly connected; and numbered sector 1 are mPNs (0.7%). The dendrites of KCα/βp neurons in the dACA (*Tanaka et al., 2008*; *Zhu et al., 2003*) are reportedly activated by bitter or sweet tastants (*Kirkhart and Scott, 2015*). However, the KCα/βp are not required for taste conditioning, which instead appears to depend on γ KCs (*Kirkhart and Scott, 2015*) and we were unable to identify strong candidates for delivering gustatory sensory information to the dACA. The PN VP5+Z adPN (5813063239) connects to two α/βp KCs has dendrites in the SEZ (*Figure 9—figure supplement 3*). But this is the only gustatory PN we can associate with the dACA, and it primarily projects to KCγm neurons through which it might participate in conditioned taste aversion (*Kirkhart and Scott, 2015*). (B) Color-coded synaptic connections from visual projection neurons (VPN; yellow) and local visual interneurons (LVIN; orange) onto α/βp KCs (gray). Note that, unlike in the vACA, there are more connections from LVINs than VPNs in the dACA.

The online version of this article includes the following figure supplement(s) for figure 11:

**Figure supplement 1.** VPN and LVIN inputs to the dACA.

**Figure supplement 2.** LVINs that conveys visual input onto α/βp KCs.

**Figure supplement 3.** Detailed morphology of a dACA LVIN.

**Figure supplement 4.** Similarity of VPN inputs to individual LVINs that innervate the vACA or the dACA.

**Figure supplement 5.** Similarity of non-visual inputs to individual LVINs.

distributions had also been mapped by *Takemura et al., 2017*, the two EM datasets are consistent. To quantitate these distributions, we compared the distances from the root of the target MBON's dendritic tree to the synapses onto it made by KCs and by feedforward MBONs. We found that, in most cases, the locations of synapses from feedforward neurons closely tracked those of KCs, showing no obvious spatial bias; an example is provided by the two feedforward MBONs that target MBON14 (α3): MBON06 (β1>α) and MBON11 (γ1pedc>α/β) (*Figure 22B*). An example of a biased spatial distribution is provided by MBON06, which feeds forward onto both MBON07 (α1) cells at

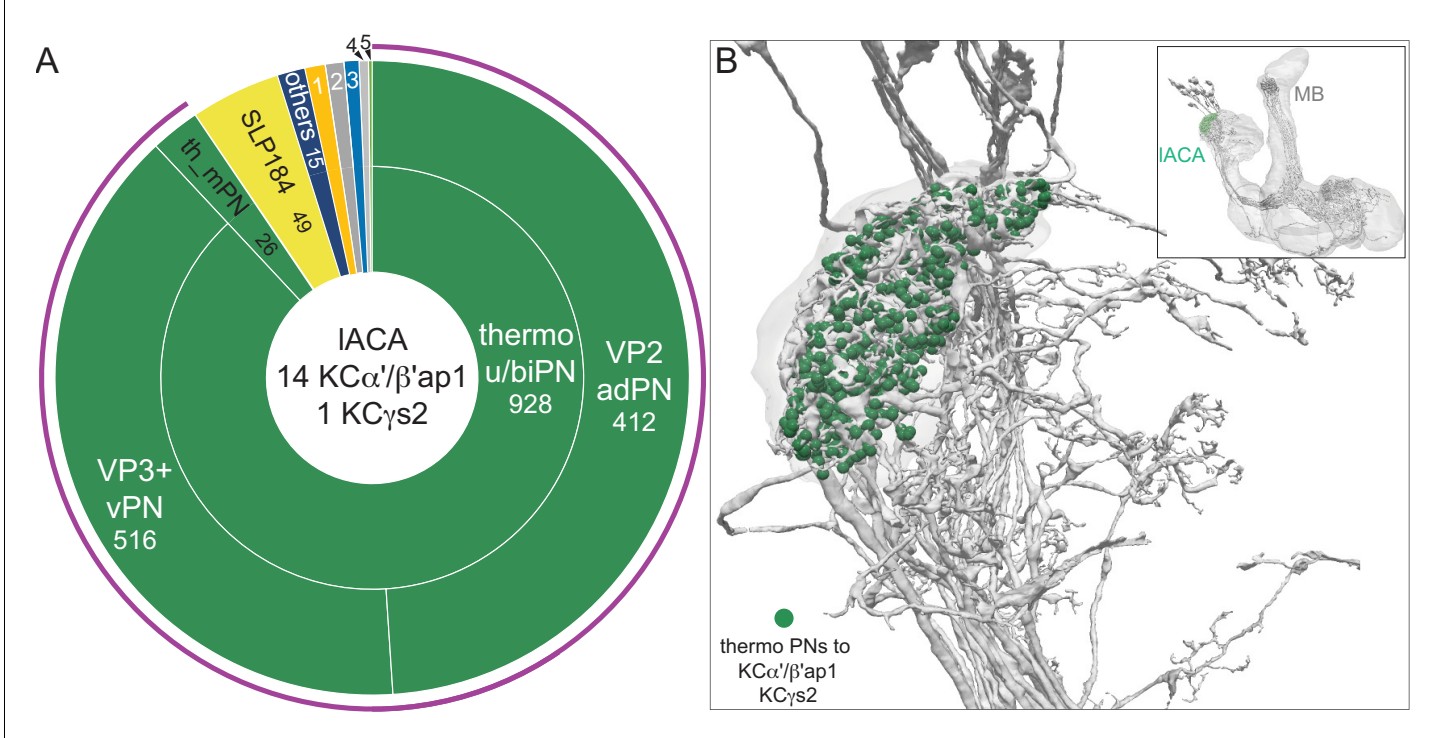

**Figure 12.** Lateral accessory calyx (lACA). The lACA is defined by the limits of the presynaptic boutons of VP2 adPN (1975878958) and VP3+ vPN (663432544); there also appear to be glia separating the lACA from main CA. Fourteen α'/β'ap1 KCs and the γs2 KC innervate the lACA. (**A**). A pie chart showing inputs to the 15 lACA-innervating KCs. The majority of input to these KCs is from the two thermosensory uPNs, VP2 adPN and VP3+ vPN, that contribute 928 synapses, of which the single γs2 KC receives 268. Other prominent inputs are one local interneuron (SLP184) and two thermo/hygrosensory mPNs. Eight interneurons (other) contribute 15 synapses. Sources of input indicated by the other numbered sectors are as follows: 1, circadian clock-associated neurons (1.0%); 2, APL (0.9%); 3, MB-C1 (0.7%); 4, KC-to-KC connections and 5, other PNs (0.2%). The total temperature information presented to KCs by PNs is indicated by the purple arc around the outer layer. Note that KCs in lACA have dramatically less KC-to-KC and APL input than those in the dACA and the vACA. (**B**) Synaptic connections from thermo uPNs (green) to KCs (gray). The inset shows the orientation of the MB.

The online version of this article includes the following figure supplement(s) for figure 12:

**Figure supplement 1.** DN1a and DNa1-like (aMe23) neurons relay temperature cues from the lACA to the circadian clock.

**Figure supplement 2.** KCγs2 and its inputs.

locations that are shifted closer to the dendritic root (*Figure 22D*). Conversely, we found MBON11 (γ1pedc>α/β) synapses onto MBON03 (β'2mp) around the edge of its dendrite far away from its root (*Figure 22E*; consistent with *Felsenberg et al., 2018*), while MBON11's synapses onto the two MBON07 cells are uniformly distributed (*Figure 22D*).

We also confirmed and extended the observation of *Takemura et al., 2017* that MBON06 and MBON11 (γ1pedc>α/β) form axo-axonal connections in the α lobe in the same compartments where they make feedforward connections onto the dendrites of other MBONs (*Figure 22—figure supplement 1*); these axo-axonal connections add an additional layer of complexity to the MBON feedforward network.

## An extensive network of MBON-to-MBON connections outside the MB

MBONs make extensive contacts with one another outside the MB lobes, as summarized in diagrammatic form in *Figure 23*. Contacts between a typical and an atypical MBON or between two atypical MBONs can be axo-dendritic, whereas contacts between typical MBONs, with the exception of the three feedforward MBONs, are almost exclusively axo-axonal. *Figure 23—figure supplement 1* shows all axo-axonal connections between typical MBON pairs that form at least 20 synapses. Such synapses are frequently observed between glutamatergic MBONs or from glutamatergic to cholinergic MBONs, as well as from GABAergic MBONs to glutamatergic MBONs, but not between

cholinergic MBONs and GABAergic MBONs. We then examined where these synapses are located on the MBON's axons. In several cases, axo-axonal synapses were highly localized and concentrated on a single axonal branch (*Figure 23—figure supplement 2*), suggesting that they might regulate synaptic transmission to only a subset of the postsynaptic MBON's downstream targets. For example, all 110 synaptic connections from MBON09 (γ3β′1) to MBON01 (γ5β′2a) are confined to a small axonal branch of MBON01. We examined the morphology of these axo-axonal synapses and found them to be indistinguishable from axo-dendritic synapses made by the same MBON (*Figure 23—figure supplement 3*).

## Atypical MBONs form a multilayer feedforward network

Atypical MBONs receive significant input from both typical and atypical MBONs, forming a multilayer feedforward network that is diagrammed in *Figure 24*. Connections occur both inside and outside the MB lobes: panel A shows axo-dendritic connections outside the MB as well as both axo-dendritic and axo-axonal connections inside the MB, while panel B shows only axo-axonal connections outside the MB. This network contains three pairs of typical MBONs that form reciprocal connections: MBON06 (β1>α) with MBON11 (γ1pedc>α/β) through axo-axonal connections within the MB (*Figure 24A*), and MBON09 (γ3β′1) with both MBON03 (β′2mp) and MBON01 (γ5β′2a) through axo-axonal connections outside the MB (*Figure 24B*). Other than the loops formed by these MBONs, the network is exclusively feedforward. MBONs 1, 5, 6, 11 and 30 provide a significant fraction of the connections in this network. It is rare for atypical MBONs to make synapses onto typical MBONs and, when these occur, they are axo-axonal (*Figure 24B*). Only MBON29 and MBON30, non-LAL-innervating atypical MBONs with axons in the dorsal brain areas, make strong axo-axonal connections to typical MBONs, primarily to MBON04 (β′2mp_bilateral). The predicted neurotransmitters for the atypical MBONs suggest they provide both excitatory and inhibitory influence on downstream neurons. MBON30, which is positioned as a hub within the network, is predicted to be glutamatergic and could therefore be either inhibitory or excitatory, depending on the receptors expressed by its postsynaptic target neurons.

The network shown in *Figure 24A* is synaptically organized into six layers, with MBON26 (β′2d) forming the top or 6th layer and MBON27 (γ5d), the 5th. Interestingly, of all the MBONs, these two are the most highly specialized for non-olfactory input; thermo-hygrosensory input in the case of MBON26 and visual for MBON27 (*Figure 15*; *Figure 25—figure supplement 1*). MBON26, at the top of the network hierarchy, receives, according to the predicted transmitter types, a mixture of excitatory and inhibitory input. MBON26 and MBON27, as well as two of the neurons in layer 4, MBON31 (α′1) and MBON32 (γ2), send approximately 35, 60, 55 and 70%, respectively, of their synaptic output to the LAL, with MBON27 and MBON32 innervating the contralateral LAL. According to their predicted transmitters, MBONs 26, 27, 31, and 32 provide convergent push-pull input to common downstream targets (*Figure 17*). The numerous pathways by which information can reach these neurons, with modality-selective input that is subject to dopamine-modulated learning combined with both learned information carried by other MBONs and with non-MB inputs, reveals the potential for atypical MBONs to perform complex input integration. We address possible functional roles for such integration in the Discussion (Figure 38).

## Atypical MBONs provide a direct path to motor control

To drive changes in behavior, such as approach or avoidance, the MB must ultimately communicate with motor neurons that lie in the ventral nerve cord (VNC). The brain is connected to the VNC by several hundred descending neurons (DNs) that pass through the neck to motor neuropils (*Namiki et al., 2018*). However, no direct MBON to DN connections have been reported, despite many DNs having dendrites in the same dorsal brain areas that serve as major MBON output sites (*Figure 18*; compare to Figure 4A of *Namiki et al., 2018*). Most DNs have their dendrites in more ventral brain regions; about a dozen lie in the LAL, a known CX (*Hanesch et al., 1989*; *Heinze and Homberg, 2009*; *Lin et al., 2013*; *Namiki and Kanzaki, 2016*; *Wolff et al., 2015*) and atypical MBON output site.

We report here the first identified direct neuronal paths from MBONs to DNs and thus to motor control. Four of the six LAL-innervating atypical MBONs are among the strongest inputs to the descending neuron (DN) DNa03, which in turn provides strong input to another DN, DNa02

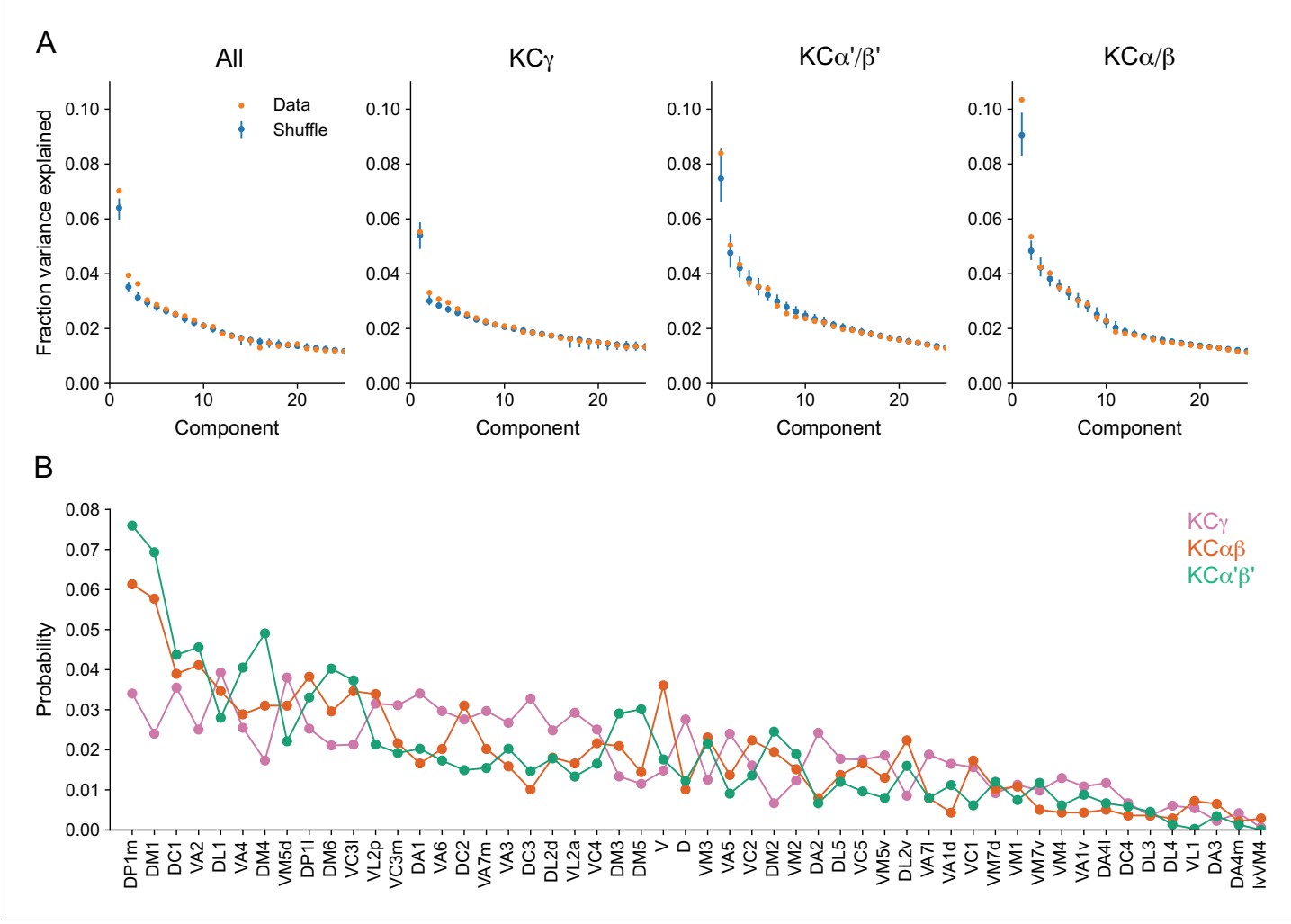

**Figure 13.** Comparison of KC input connectivity to random models. (**A**) Fraction of variance explained by components identified via principal components analysis of the olfactory uPN-to-KC input connectivity matrix—a binary matrix containing ones and zeros for present or absent connections between KCs and their inputs, at a threshold of five synapses. Results are shown for the reconstructed data (orange) and for a collection of shuffled models (blue) in which each KC retains the same total number of connections but samples among all uPNs randomly, with a probability proportional to the total number of connections each uPN makes. Such models therefore retain the degree distribution across KCs, (that is, the probability distribution of the number of claws formed by individual KCs) and the average connection probability for each uPN, but no other structure. Bars indicate 95% confidence intervals for shuffled models. Deviations in the first few components indicate structure inconsistent with a random model. Left: All KCs; Right: Analysis restricted to the indicated KC subtypes (visual α/βp and γd KCs, as well as γs KCs, were excluded). (**B**) Probability of uPN-to-KC connections from each olfactory glomerulus, sorted by most to least well-connected. Probabilities are plotted separately for the KC subtypes shown in (**A**).

The online version of this article includes the following figure supplement(s) for figure 13:

**Figure supplement 1.** Clustering KCs based on the similarity of their inputs from PNs.

**Figure supplement 2.** Similarity of KC outputs to MBONs.

(*Figure 25*). DNa02 also gets weaker, but direct, input from the ipsilateral MBON32 and from the MBON31s in both hemispheres. Asymmetrical activity in DNa02 has been implicated in the steering of a fly's walking direction (*Rayshubskiy et al., 2020*). The visual and thermo/hygrosensory sensory input these MBONs receive is predominantly ipsilateral, allowing them to convey directional sensory information that might then be used to drive approach or avoidance. In the Discussion, we present a circuit model for how the network of atypical MBONs might promote directional movement. The other two LAL-innervating MBONs, MBON33 and MBON35, also connect to neurons that we believe are DNs, but this needs to be confirmed by additional reconstruction.

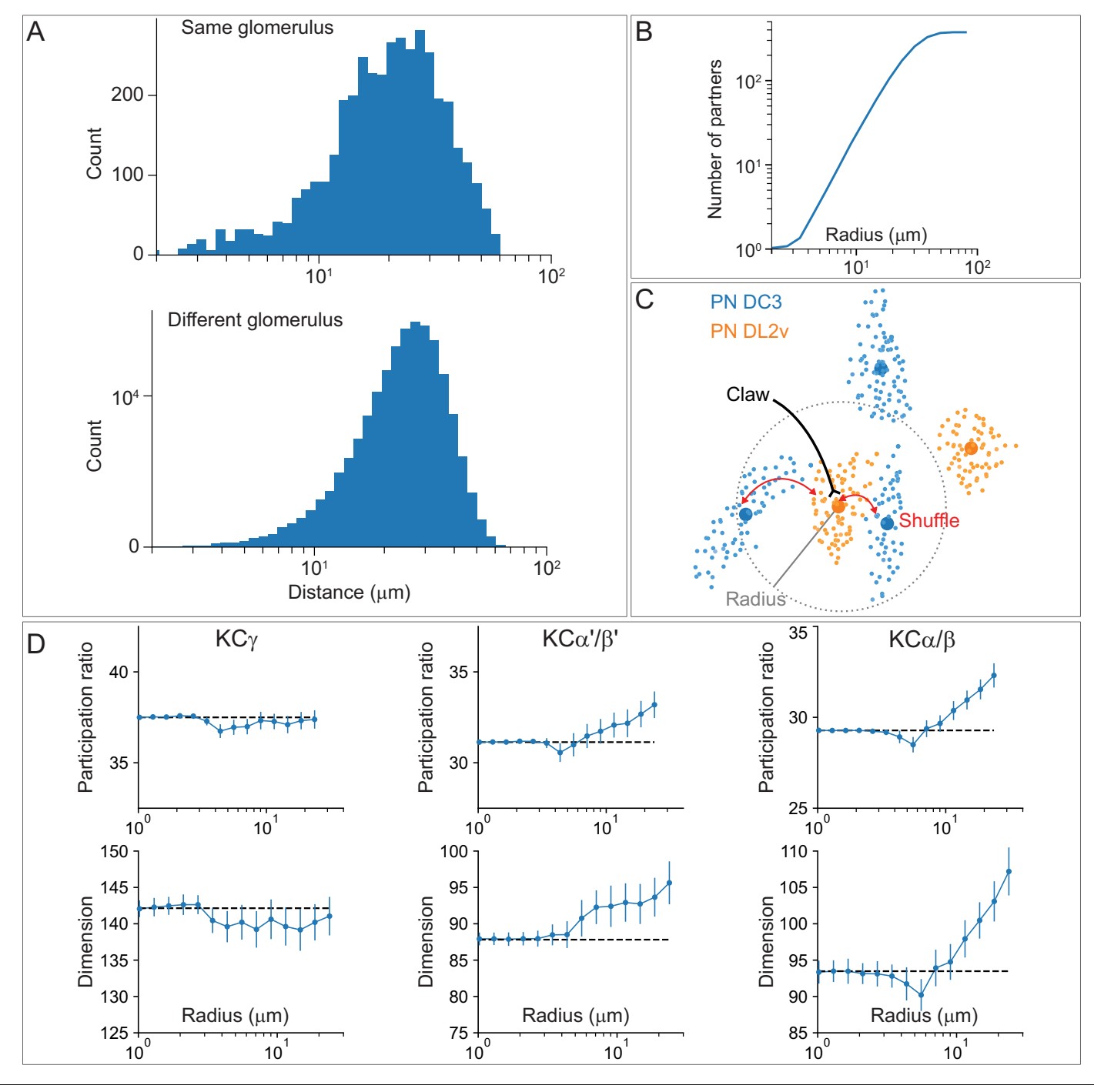

**Figure 14.** Effect of spatial organization on the KC representation. (**A**) Histogram of pairwise distances between boutons formed by PNs from the same glomerulus (top) or different glomeruli (bottom). (**B**) Average number of nearby boutons within a given radius of each PN bouton. (**C**) Illustration of shuffling procedure used to randomize PN-to-KC connectivity within a given radius. Small dots: S ynapses formed by each PN onto KCs. Large dots: Centroid location of identified boutons. Claws are randomly reassigned (red arrows) to boutons whose centroids lie within the given radius, indicated by the dotted gray line. (**D**) Participation ratio (top) and dimension (bottom) of models constructed by shuffling connectivity within a specific radius. The dependence of these two quantities on the shuffle radius shows a modest effect of the spatial organization of the CA on the KC representation in the model.

DNa03 appears to serve as a node for convergence of directional information from the optic lobes and the CX. DNa03 receives strong, direct visual input from multiple cells of a population of lobula VPNs (LT51 neurons; *Figure 25—figure supplement 1* and *Figure 25—figure supplement 2A,B*). Unlike the VPN neurons that innervate the accessory calyces we described earlier, these LT neurons get input from layers of the lobula known to be involved in feature detection (see for example, *Wu et al., 2016*). Moreover, DNa03 gets strong input from a population of columnar FB neurons, PB1-4FB1,2,4,5LAL (PFL2; *Figure 25—figure supplement 1* and *Figure 25—figure supplement 2C,D*) that are likely to convey orientation-sensitive information (discussed further in *Hulse et al., 2020*).

While six atypical MBONs innervate the LAL, the other eight have their arbors in more dorsal regions. We asked if any of these MBONs had identified DNs among their top 20 downstream targets. In this way, we were able to establish strong connections from MBON20 to two DNs, DNp42 and DNb05 (*Figure 8—figure supplement 2*). DNp42 was recently described in the FAFB EM dataset, where this strong MBON20 connection can also be observed; DNp42 is required for innate aversive olfactory behavior while its optogenetic activation causes animals to back up (*Huoviala et al., 2020*). No other identified DNs were found among the top downstream targets of any other atypical or typical MBON.

We do not mean to suggest that these are the only direct MBON to DN connections. At most one-third of the DNs expected from light-level analyses *Namiki et al., 2018* have been identified in the hemibrain v1.1 dataset. We also identified weaker connections from atypical MBONs to other putative DNs as well as strong connections to LAL neurons that might turn out to be DNs upon more extensive analysis. However, it seems likely that the connections we describe here will remain among the strongest direct MBON to DN connections. Of course, all MBONs are likely to drive DN activity through more indirect pathways, as it is through DNs that motor activity is largely controlled.

## Structure of DAN connectivity

DANs are divided into two major groups, PPL1 and PAM, that preferentially encode punishment and reward, respectively (reviewed in *Modi et al., 2020*). But there is growing evidence that DANs provide a wider range of information to the MB about novelty, locomotion, sleep state, reward or punishment omission, and safety (*Aso and Rubin, 2016*; *Cohn et al., 2015*; *Dag et al., 2019*; *Felsenberg et al., 2018*; *Felsenberg et al., 2017*; *Gerber et al., 2014*; *Handler et al., 2019*; *Hattori et al., 2017*; *Jacob and Waddell, 2020*; *Sitaraman et al., 2015b*; *Tanimoto et al., 2004*). Understanding how DANs represent the external world and internal brain state requires a systematic investigation of their upstream inputs. Recent connectomic studies of DAN inputs in the larval brain (*Eschbach et al., 2020a*) and to a subset of DANs in the adult (*Otto et al., 2020*) revealed a surprising degree of heterogeneity. To understand what neuronal pathways contribute to the signals that DANs convey, as well as how these signals might produce learning-related changes in the strength of KC-to-MBON synapses, we characterized the inputs and outputs of DANs.

## Feedback from MBONs to DANs

We found extensive, direct synaptic connections between the axonal termini of MBONs and the dendrites of DANs (*Figure 26*). All five PPL1 DANs and 99 of the 150 PAM DANs receive direct MBON input; 19 of 20 typical MBON cell types, with representatives of all three neurotransmitter types, and 12 of 14 atypical MBON cell types make direct connections to DANs. Connections were observed between MBONs and DANs that innervate the same compartment (*Figure 26A*), different compartments (*Figure 26B*) or both the same and different compartments (*Figure 26C*). Only the α1 compartment displays exclusively self-feedback. Glutamatergic, and to a lesser extent, cholinergic, MBONs exhibit a strong bias toward providing feedback to DANs of the same compartment (self-feedback). A subset of PPL1 DANs was overrepresented among DANs receiving both within-compartment and cross-compartment connections (*Figure 26C,D*). The three MBONs that have axonal termini within the MB lobes (*Figure 22*) also make axo-axonal synapses onto a subset of DANs (*Figure 26—figure supplement 1*).

Prior studies have suggested a variety of roles for feedback from MBONs to the DANs that innervate the same compartment and modulate them. For example, the nutrient-dependent consolidation of appetitive long-term memory requires the within-compartment feedback from MBON07 (α1) to

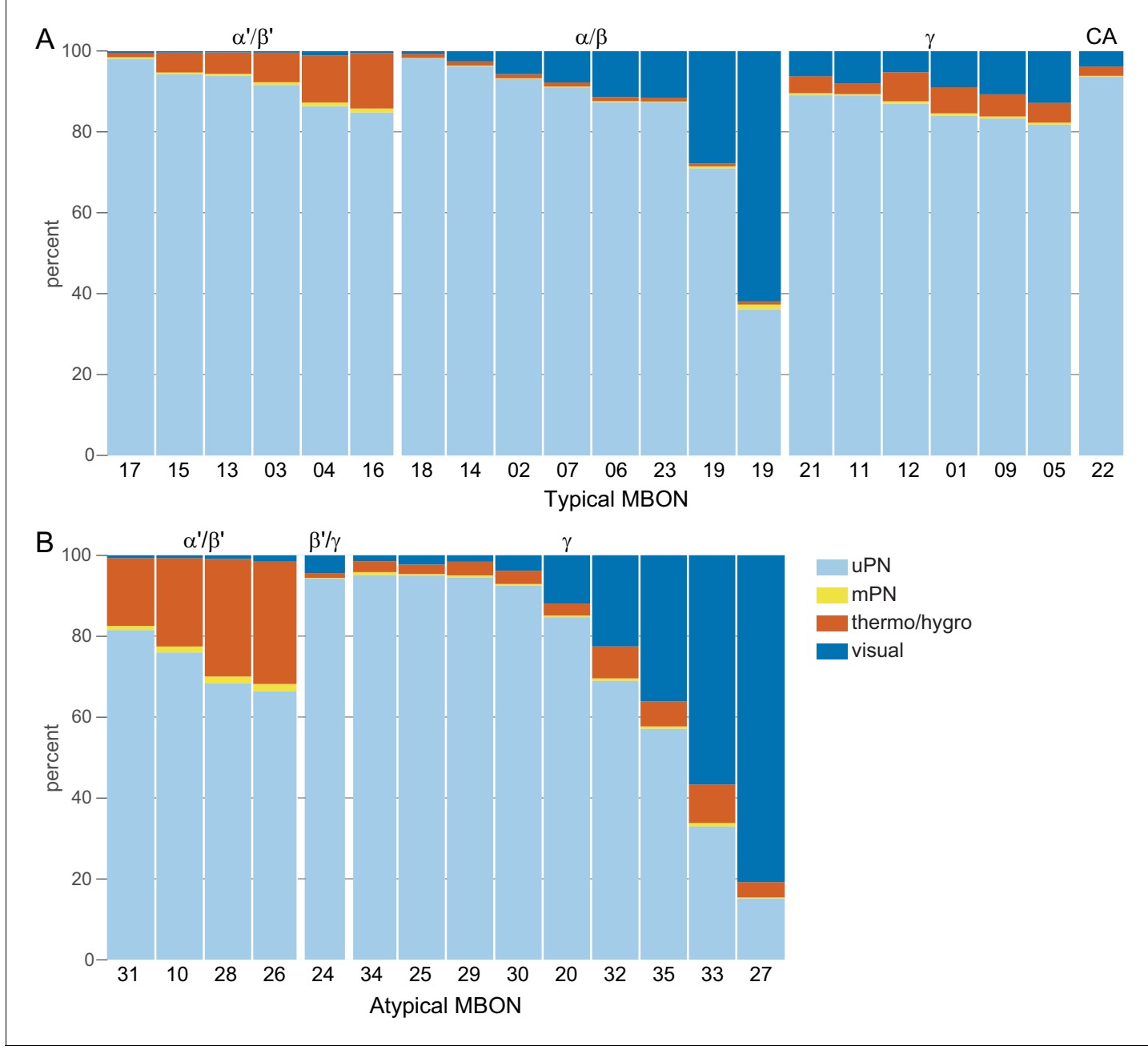

**Figure 15.** Structure in PN-KC-MBON connectivity. (**A**) Effective PN-to-MBON connectivity. MBONs within the α'/β' lobe receive input primarily from uniglomerular olfactory PNs (uPNs), but they also show a gradation of input from thermo-hygrosensory PNs. MBONs from the α/β lobes show a similar, although stronger, variation in the amount of input they receive from visual PNs. MBONs from the γ lobe are more uniformly innervated. For cases where the MBON cell type contains more than one cell, results from the individual cells have been averaged in this figure, with the exception of the two MBON19 (α2p3p)s; both MBON19s are specialized for visual input, but to different extents. (**B**) Effective PN to atypical MBON connectivity. The gradation in non-olfactory input is stronger than for typical MBONs. In particular, the majority of the input to MBON33 (γ2γ3) and MBON27 (γ5d) is visual. In this figure and its supplements, the MBONs have been grouped by the MB lobe they primarily innervate and then ordered, within these groups, according to their selectivity. MBON24 (β2γ5) receives input from both the β and γ lobes. Percentages indicate the amount of input to each MBON from a particular PN type divided by the total sensory input to that MBON.

The online version of this article includes the following figure supplement(s) for figure 15:

**Figure supplement 1.** PN connections to KC types and effective PN-to-MBON connectivity.

**Figure supplement 2.** PN-to-KC and KC-to-MBON connectivity by sensory modality.

**Figure supplement 3.** MBON connectivity by KC type.

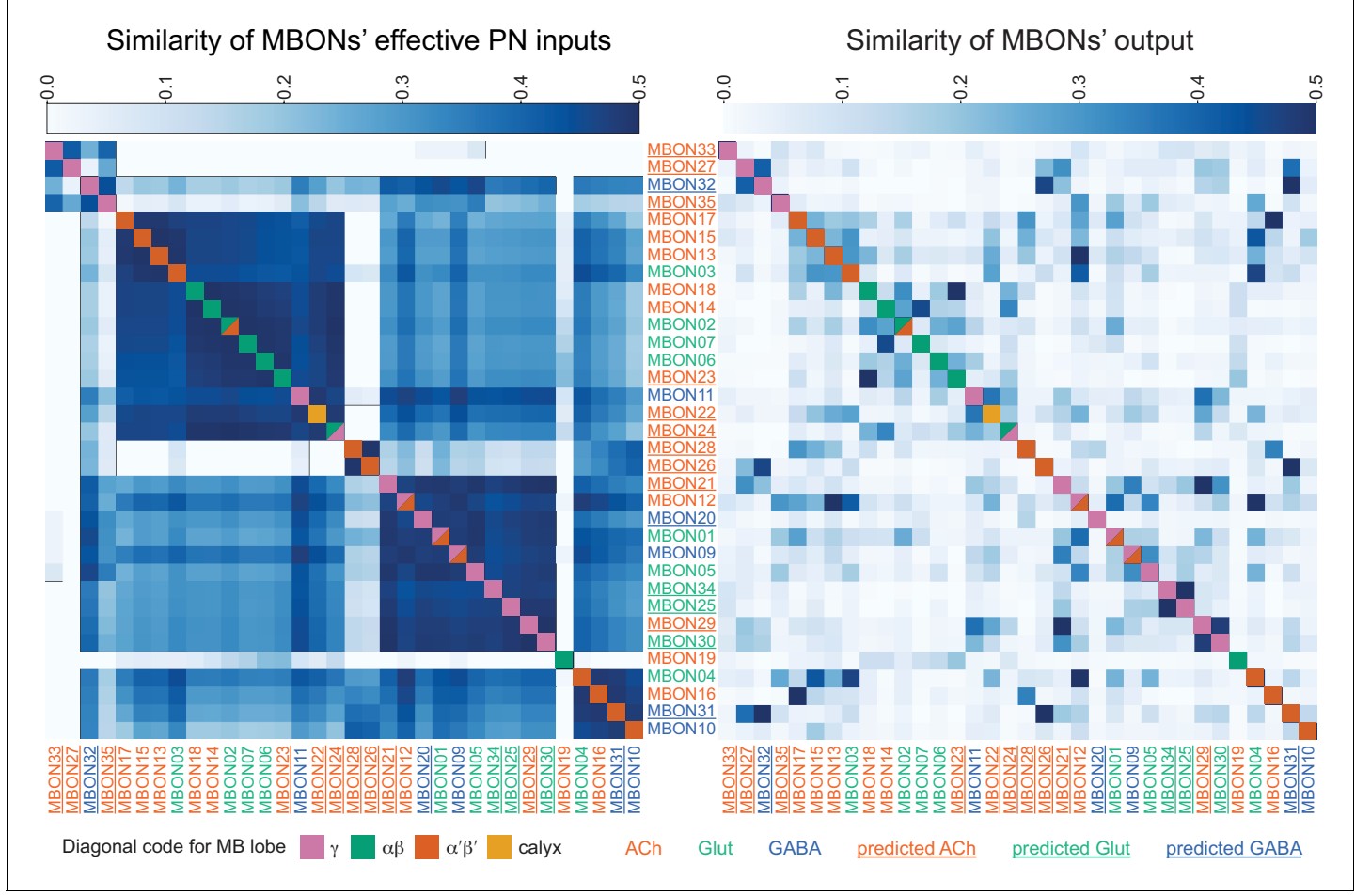

**Figure 16.** MBON input and output similarity structure. A comparison of MBON inputs and outputs through the lens of similarity structure. Left: cosine similarity of MBONs based on their effective PN inputs via KCs (computed as in *Figure 15*). Each cell of the heat map indicates the input similarity of the indicated pair of MBONs. A value of 0.5 indicates that the PN subpopulations conveying input to the two MBONs via KCs are half overlapping and half disjoint. Right: cosine similarity of MBONs based on their outputs to all neurons (unnamed neuronal fragments have been excluded). The resemblance between the left and right plots suggests that MBON outputs preserve some of the parallel structure in the PN-KC-MBON pathway. In each plot, the square in the diagonal is color-coded to indicate the MB lobe in which that MBON's dendrites lie. The MBON names are color-coded to indicate their neurotransmitter, as indicated.

The online version of this article includes the following figure supplement(s) for figure 16:

**Figure supplement 1.** MBON output similarity by lobe.

**Figure supplement 2.** Correlation of MBON output and PN input similarity.

**Figure supplement 3.** MBONs clustered by output similarity.

**Figure supplement 4.** MBON morphologies grouped by output similarity.

activate the PAM11 (α1) DANs (*Ichinose et al., 2015*). As noted above, the PAM11 (α1) DANs, specifically the PAM11-aad subtype, receive only within-compartment feedback, suggesting that consolidation may be a dedicated function of this MBON-DAN recurrent feedback loop. Consolidation of appetitive long-term memory also requires inhibition from MBON11 (γ1pedc>αβ) onto the PPL101 (γ1pedc) DANs (*Pavlowsky et al., 2018*) after training. The MBON11 (γ1pedc>αβ) to PPL101 (γ1pedc) DAN connection has also been linked to the persistence of hunger-dependent food odor seeking (*Sayin et al., 2019*). Lastly, feedback activation of PAM01 (γ5) by MBON01 (γ5β'2a) is required for the maintenance of short-term courtship memory (*Zhao et al., 2018b*) and for aversive memory extinction (*Felsenberg et al., 2018*; *Otto et al., 2020*).

In addition to direct connections, we also computed the effective connection strength from MBONs to DANs mediated by an interneuron (*Figure 26—figure supplement 2*) and the proportion of this feedback where an MBON feeds back onto a DAN that innervates the same compartment

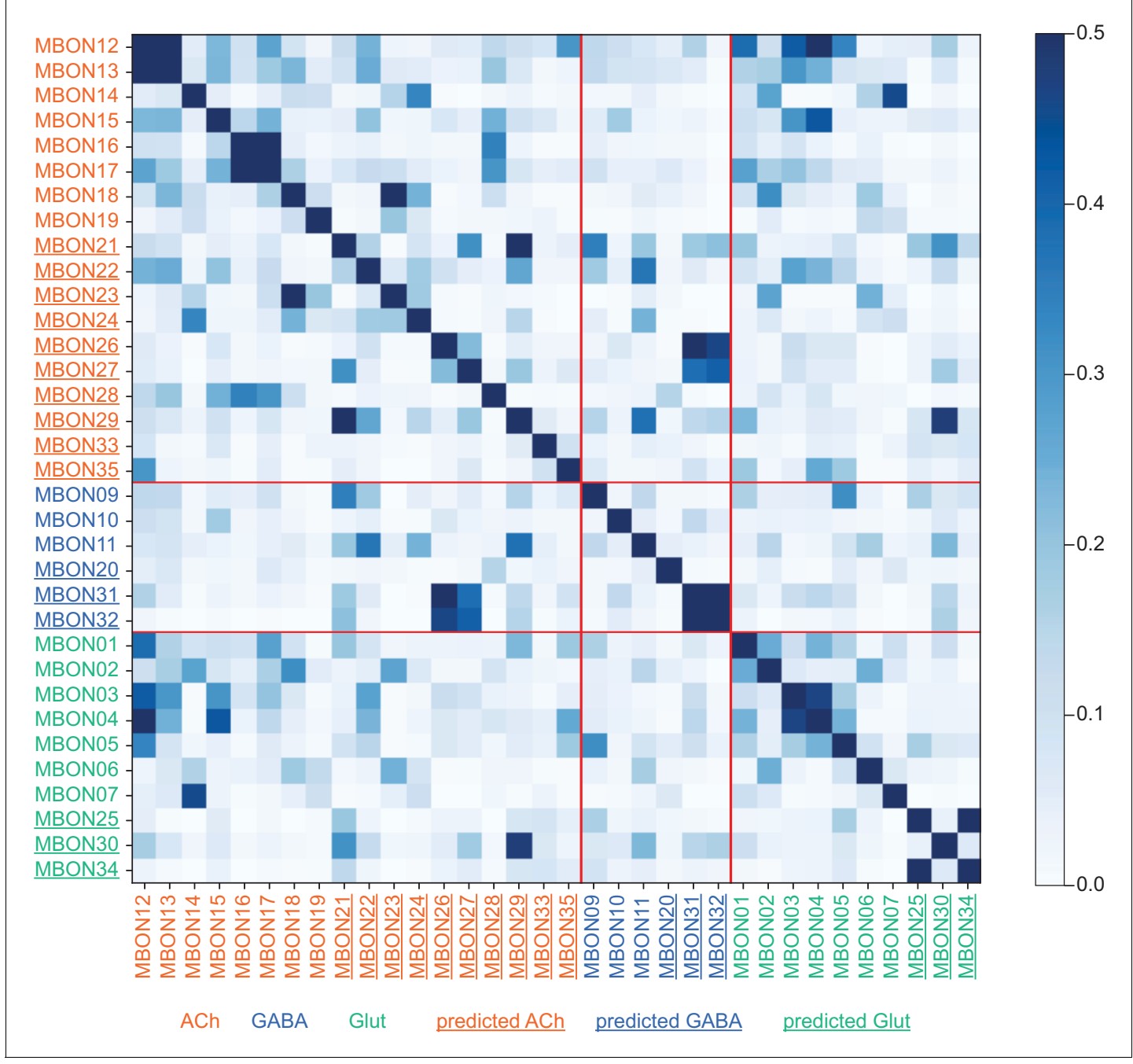

**Figure 17.** MBON output similarity by neurotransmitter. Cosine similarity of MBON outputs to all named neurons. MBONs have been grouped by neurotransmitter and then presented in numerical order. There exist many instances of convergent outputs between cholinergic and glutamatergic neurons, suggesting a widespread 'push-pull' motif in MBON output convergence patterns. This phenomenon is emphasized in *Figure 17—figure supplement 1* and notable examples are explored in further detail in *Figure 20—figure supplement 1* and *Figure 21—figure supplements 1,2*. The MBON names are color-coded by neurotransmitter with predicted transmitters underlined.

The online version of this article includes the following figure supplement(s) for figure 17:

**Figure supplement 1.** MBON output similarity pooled by neurotransmitter.

(*Figure 26—figure supplement 3*). Interestingly, most of the MBONs providing direct same-compartment feedback are glutamatergic, a bias which is also evident in indirect feedback. The presence of this motif in both direct and indirect MBON-DAN interactions is intriguing, and we consider its potential computational significance in the Discussion.

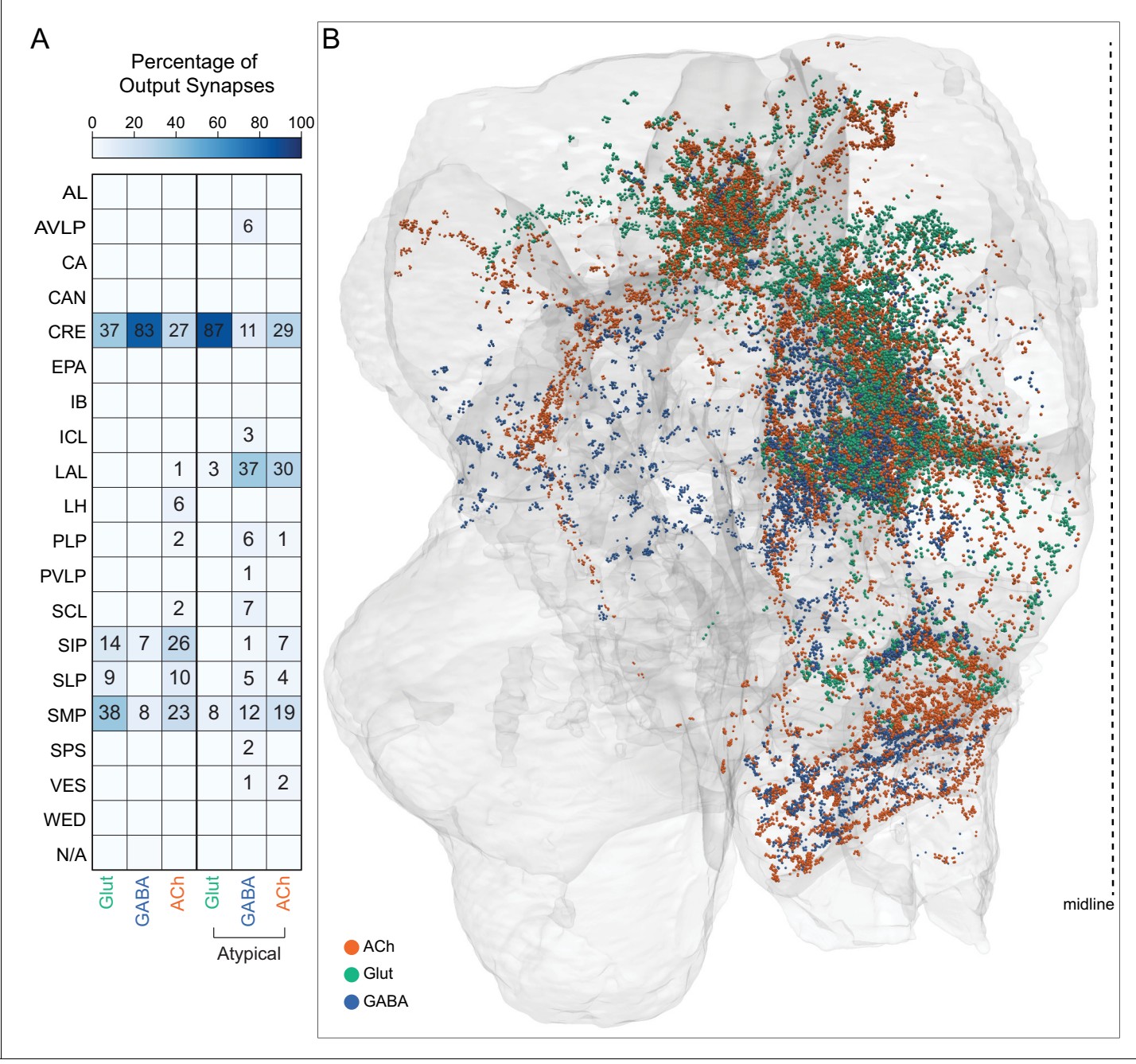

**Figure 18.** MBON output distribution by neurotransmitter. (**A**) Table indicating the percentage of output synapses of each MBON neurotransmitter type, including predicted neurotransmitters, that reside in the given brain area; typical and atypical MBONs were analyzed separately. (**B**) Visualization of the spatial position of MBON output synapses throughout the hemibrain volume, color-coded by neurotransmitter type. **Figure 18—figure supplement 1** shows the same data presented on separated brain areas. **Figure 18—figure supplement 2** shows data for individual MBONs pooled by brain area. **Figure 18—figure supplement 3** shows the same data as **Figure 18**, but with predicted neurotransmitters shown in gray. The online version of this article includes the following figure supplement(s) for figure 18:

**Figure supplement 1.** MBON output distribution by neurotransmitter presented on separated brain areas.

**Figure supplement 2.** Distribution of individual MBON outputs by brain area.

**Figure supplement 3.** Same plot as in **Figure 18** but with only synapses from neurons whose neurotransmitters had been confirmed by antibody staining ( **Aso et al., 2014a**) shown color-coded.

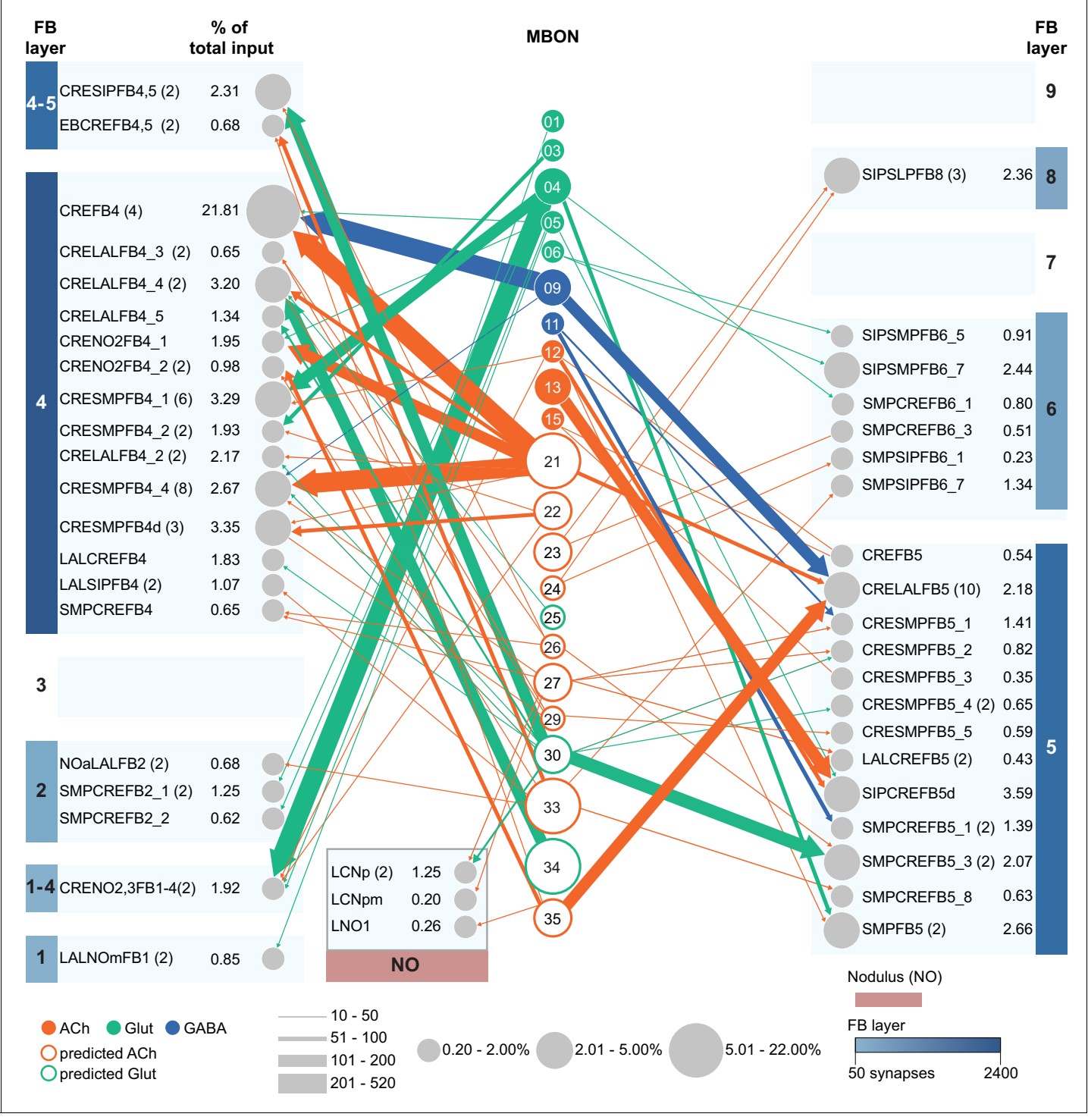

**Figure 19.** Direct connections from MBONs to the central complex (CX). In this diagram, the central column shows those MBONs that make direct connections of 10 or more synapses onto the dendrites of CX neurons. Such direct connections predominantly occur on fan-shaped body (FB) neurons. Three cell types of nodulus (NO) cell types are also direct targets (see *Figure 19—figure supplement 1A*); some FB cell types in layers 1, 2, 3, and 4 that are downstream of MBONs also have arbors in the NO. No direct targets were identified in other CX neuropils such as the ellipsoid body (EB) or protocerebral bridge (PB). The number in the circle indicates the MBON cell type; that is, 01 stands for MBON01. The size of the circle indicates what percent of that MBON's total output (based on synapse number) goes directly to FB neurons. The color of solid circles represents confirmed neurotransmitter types while the color of the outline of hollow circles represents predicted neurotransmitter types. Numbers have been pooled for cases where there are multiple MBONs of the same cell type. The outer columns show FB cell types, arranged by FB layer(s). The name of the cell type

*Figure 19 continued on next page*

*Figure 19 continued*

and, in parentheses, the number of cells of that type are indicated for those with multiple cells. The percentage of that cell type's input that comes from MBONs is given and is reflected by the size of the circle representing the cell type. All FB neuron types receive less than 5% of their total input from MBONs except for one layer 4 FB neuron type, CREFB4 (FB4R), which receives nearly 20% of its inputs from two MBONs, MBON09, and MBON21 (see *Figure 19—figure supplement 1B,C*). Direct MBON connections to the FB are concentrated in FB layers 4 and 5. Links to each of these FB and NO cell types in neuPrint, listed from FB layer 1 to FB layer 9 and then for the nodulus, are as follows: LALNOmFB1 (FB1C), CRENO2,3FB1-4 (FB1H), NOaLALFB2 (FB2A), SMPCREFB2_1 (FB2C), SMPCREFB2_2 (FB2L), CRESMPFB4_1 (FB4A), CRENO2FB4_1 (FB4C), CRESMPFB4_2 (FB4D), CRELALFB4_2 (FB4F_b), CRELALFB4_3 (FB4G), CRELALFB4_4 (FB4H), LALCREFB4 (FB4I), CRELALFB4_5 (FB4J), LALSIPFB4 (FB4L), CRENO2FB4_2 (FB4M), SMPCREFB4 (FB4N), CRESMPFB4d (FB4O), CRESMPFB4_4 (FB4P_a and FB4P_b), CREFB4 (FB4R), CRESIPFB4,5 (FB4X), EBCREFB4,5 (FB4Y), LALCREFB5 (FB5A), SIPCREFB5d (FB5AB), SMPCREFB5_1 (FB5C), CRESMPFB5_1 (FB5D), CRESMPFB5_2 (FB5E), CRESMPFB5_3 (FB5H), SMPCREFB5_3 (FB5I), SMPFB5 (FB5J), CREFB5 (FB5K), CRESMPFB5_4 (FB5L), CRESMPFB5_5 (FB5M), CRELALFB5 (FB5V), SMPCREFB5_8 (FB5X), SMPSIPFB6_1 (FB6A), SMPCREFB6_1 (FB6P), SIPSMPFB6_5 (FB6Q), SMPSIPFB6_7 (FB6R), SIPSMPFB6_7 (FB6T), SMPCREFB6_3 (FB6V), SIPSLPFB8 (FB8F_a), LCNp (LCNOp ), LCNpm (LCNOpm), and LNO1 (LNO2).

The online version of this article includes the following figure supplement(s) for figure 19:

**Figure supplement 1.** The morphology and connectivity of nodulus and CREFB4 neurons.

## Inputs to individual DANs

We characterized the inputs to the dendrites of DANs by computing cosine similarity matrices of these inputs for all pairs of the 156 DANs that innervate the MB (*Figure 27*). An enlarged view of PPL neurons is shown in *Figure 27—figure supplement 1* and a simplified matrix, in which the data is collapsed by compartment is shown in *Figure 27—figure supplement 2*. Structure at the level of lobes, compartments, and even within compartments is evident. The heterogeneity of input to different DANs is also clearly visible. In some cases, DANs receive prominent input from particular brain areas (*Figure 27—figure supplement 3*, left panel). For instance, α′3 and α′2α2 DANs receive more than half their input from the SIP, γ3, and γ4<γ1γ2 DANs from the CRE, and γ5β′2a, γ5, γ1pedc, β′2a, β′2m, β′2p, and α3 DANs from the SMP. Tracing DAN inputs back to include those inputs mediated by an interneuron reveals even more complex and broadly distributed inputs (*Figure 27—figure supplement 3*, right panel), with some distinct differences from direct inputs such as vastly increased input from the CA. In general, these input correlations suggest a heterogeneity in DAN function that could in principle support more than a dozen different learning modalities. We identified 40 putative functional groups of DANs, and an analysis of the singular values of the DAN input connectivity matrix suggests at least 20 sub-compartmental zones within the MB lobes where distinct modulation by DANs might occur.

DAN inputs need to be organized so they can convey learning-related signals to the appropriate KC-MBON synapses within the MB. Such an organization should be reflected in the relationship between the structure of DAN inputs and MBON outputs. For each pair of compartments, we computed the similarity of the upstream inputs received by DANs innervating the two compartments and the corresponding similarity of the downstream targets of MBONs from those compartments (*Figure 27—figure supplement 4*). We identified a significant positive correlation between these two similarity measures, suggesting a form of 'credit assignment' in which compartments whose MBONs control similar behaviors are innervated by DANs that receive similar reinforcement signals. This correlation suggests that the observed heterogeneity of DAN inputs is functionally-relevant.

## PAM DANs cluster into morphological subtypes that are reflected in upstream connectivity

Grouping DANs into subtypes using hierarchical clustering based on either of two different criteria, morphology or upstream connectivity, gave strikingly similar results. *Figure 28* shows comparisons of morphology- and connectivity-based clusters for PAM01 (γ5), PAM12 (γ3), and PAM11 (α1) DANs. For α1, morphology and input clustering generate an identical dendrogram, whereas for γ5 and γ3 the major groupings are identical but some of the 'within group' orders are shifted. Finding that morphology clustering reflects input clustering demonstrates that morphology is a good indicator of functionally-relevant DAN subtypes, as previously demonstrated for γ5 DANs (*Otto et al., 2020*).

*Figure 28—figure supplement 1* illustrates the use of morphology to divide PAM01 (γ5), PAM12 (γ3) and PAM11 (α1) DANs into subtypes. The most distinguishing morphological features are the

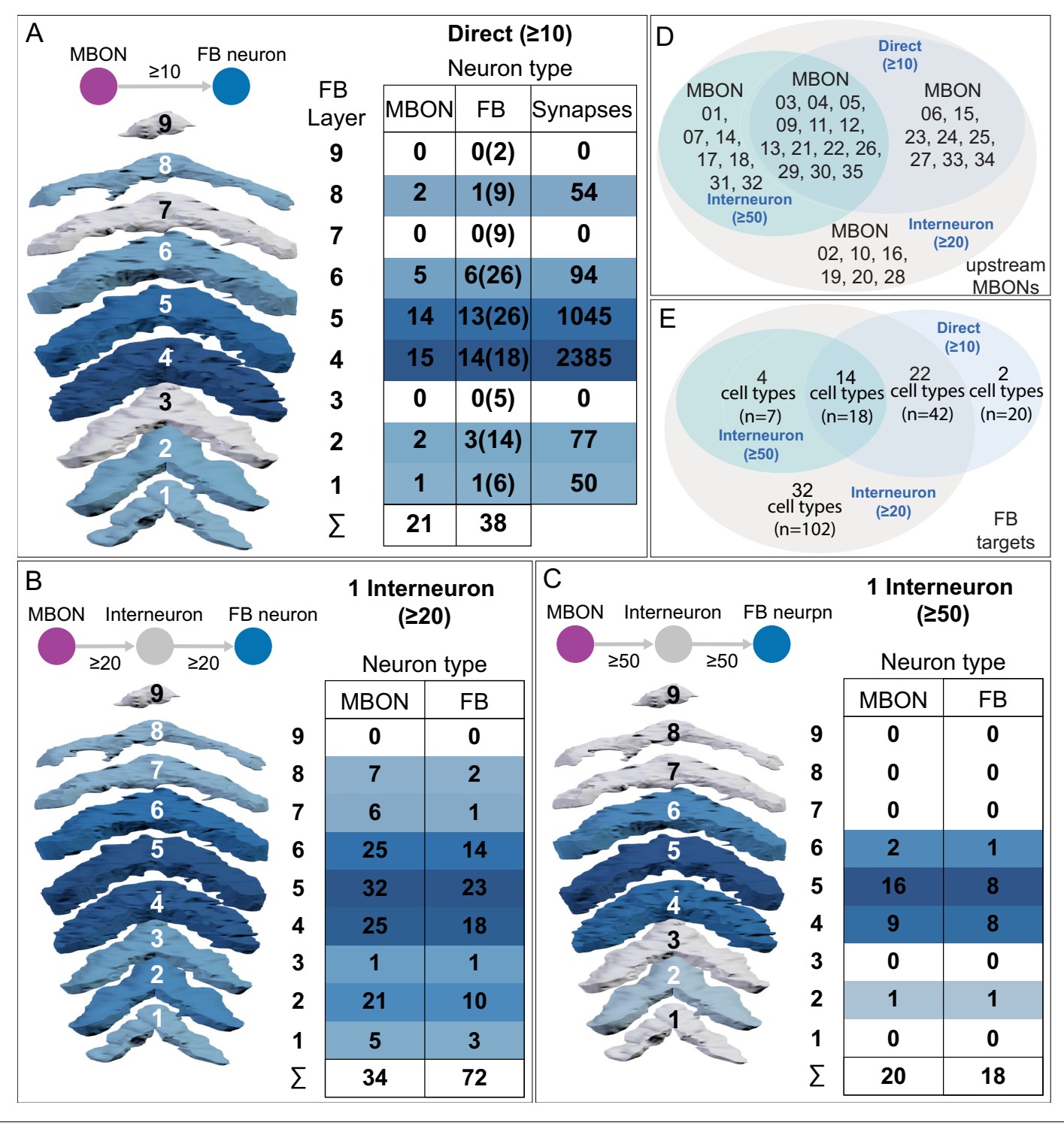

**Figure 20.** Summary of connections from MBONs to the FB. Heat maps are shown that compare the strength of connection between MBONs and FB neurons in each FB layer. (A) Direct connections. The number of MBON cell types within each layer that make synapses onto the dendrites of FB neurons, the number of FB neuron types (also within the indicated layers) they connect to, and the total number of MBON to FB neuron synapses are shown for each FB layer. The total number of cell types listed in neuPrint for each FB layer is shown in parentheses. The last row of the table shows the number of different MBON cell types that make direct connections to the FB; some MBONs connect to more than one layer. (B,C) Connections using a single interneuron. The number of MBON cell types connecting to each layer and the number of FB neuron types they connect to are shown. Connections between the MBON and the interneuron as well as between the interneuron and the FB neuron are required to have 20 or more synapses

*Figure 20 continued on next page*

*Figure 20 continued*

(**B**) or 50 or more synapses (**C**). The last row of the table shows the number of different MBON cell types that make indirect connections to the FB at this threshold; some MBONs connect to more than one layer. (**D**) Venn diagram comparing MBON types that are connected to FB neurons directly (as in A); through one interneuron with threshold of 20 synapses (as in B); and through one interneuron with threshold of 50 synapses (as in C). (**E**) Venn diagram comparing FB cell types that are targeted by MBONs directly (as in A), through one interneuron with threshold of 20 synapses (as in B), and through one interneuron with threshold of 50 synapses (as in C). The number of cell types and the total number of neurons (n) are indicated.

The online version of this article includes the following video and figure supplement(s) for figure 20:

**Figure supplement 1.** Examples of MBON to FB connectivity involving an interneuron.
**Figure 20—video 1.** MBONs providing direct input to the FB.
https://elifesciences.org/articles/62576#fig20video1
**Figure 20—video 2.** Multiple MBONs converge on an interneuron that provides input to the FB – example 1.
https://elifesciences.org/articles/62576#fig20video2
**Figure 20—video 3.** Multiple MBONs converge on an interneuron that provides input to the FB – example 2.
https://elifesciences.org/articles/62576#fig20video3
**Figure 20—video 4.** Multiple MBONs converge on an interneuron that provides input to the FB – example 3.
https://elifesciences.org/articles/62576#fig20video4

locations of dendritic and axonal fields, and the commissure in which a contralaterally projecting axon crosses the midline. Importantly, our clustering results—using cosine input similarity (*Figures 27*,*29*), input similarity and morphology (*Figure 28*)—all support the division of PAM cell types into subtypes in every compartment that is innervated by a population of DANs. As discussed in the next section, this sub-compartmental structure is likely to have important functional consequences.

## PAM DAN subtypes selectively modulate subsets of KCs and specific MBONs within single compartments

We reasoned that DAN subtypes might specifically connect to particular types of KCs and MBONs within a single compartment. Previous work suggested that the major axonal arbors of γ5 DAN subtypes innervate distinct subregions within the compartment (*Otto et al., 2020*). The dense anatomical reconstruction and connectivity of neurons in the hemibrain dataset allowed us to examine whether such differences in morphology are reflected in connectivity and, if so, whether this is a general feature of all MB compartments that are innervated by multiple DANs. We first clustered DANs within a compartment based on the similarity of their dendritic inputs and then, without changing the order determined by that clustering, asked if we could observe structure in the pattern of their outputs onto KCs (*Figure 29*). In some compartments, for example β1, β'1, and γ4, such conserved structure was indeed observed indicating that DANs with similar inputs also output onto similar sets of KCs. In other compartments known to have clear subtypes based on their dendritic inputs and morphology, such as α1, γ3, and γ5, conserved structure was not identified by this analysis. Therefore, we investigated the DANs in these three compartments in more detail, as described below, and found clear examples of subtype specificity in their synapses onto KC cell types. We also found selectivity in DAN subtype synapses onto MBONs within compartments innervated by multiple MBON cell types, including the newly discovered atypical MBONs.

Each of the four defined PAM01 (γ5) DAN subtypes is differentially connected to distinct KC populations within the γ5 compartment. This is most evident for PAM01-nc and PAM01-fb DANs whose presynaptic arbors occupy distinct regions of the compartment (*Figure 30A and B*) where they contact KC populations representing different stimulus modalities (*Figure 30D and E*; *Figure 30—figure supplements 1* and *2*; *Figure 15*). For example, PAM01-nc DANs are the major input to the visual streams of γd and γs1 KCs (*Figure 30D*). In contrast, the PAM01-fb DANs synapse predominantly with the main olfactory KCγ population, and weakly with the γt and γs1 KCs (*Figure 30E*). This arrangement suggests that DAN modulation can be specific to particular sensory modalities.

DANs are known to make synapses onto the dendrites of MBONs (*Takemura et al., 2017*), but the physiological and behavioral roles of DAN to MBON connections are much less well understood than those of DAN to KC connections; in the one case where there is functional data, DAN activation induced a slow depolarization in the postsynaptic MBON (*Takemura et al., 2017*). We found that DANs also showed selectivity within individual compartments in their targeting of MBONs. For

example, in γ5 the PAM01-fb DANs synapse onto MBON01 (γ5β′2a) and MBON29 (γ4γ5), but not MBON27 (γ5d), whose dendritic field preferentially occupies the dorsal region of the compartment. MBON27 nevertheless receives synapses from the other three classes of PAM01 DANs. Other MBONs in γ5 also receive preferential input from specific PAM01 DAN subtypes; for example, MBON24 (β2γ5) only receives synapses from the PAM01-lc DANs.

A similar but less complex subtype arrangement of DAN-KC and DAN-MBON connectivity is evident in the γ3 compartment, where there are two morphological DAN subtypes (*Figure 28—figure supplement 1E and F*; Figure 39). The PAM12-md DANs provide most of the DAN input to the γd KCs, whereas the γm and γt KCs are preferentially innervated by the PAM12-dd DANs (*Figures 30C*, 32). The three MBONs with processes in the γ3 compartment exhibit very different connectivity from γ3 DAN subtypes. PAM12-dd DANs provide all of the DAN input to MBON30 (γ1γ2γ3) and the majority of MBON09 (γ3β′1)'s input, whereas PAM12-md DANs provide the majority of MBON33 (γ2γ3)'s input.

The three α1 DAN subtypes ( *Figure 28—figure supplement 1G–I* ) also exhibit preferential KC wiring (*Figure 30C*). For example, PAM11-nc arbors only occupy a limited portion of the α1 compartment (*Figure 28—figure supplement 1I*) where they preferentially innervate the visual stream α/βp KCs, while avoiding α/βc and α/βm KCs. The α1 compartment houses the dendrites of only a single MBON type, whose two members each fill the compartment (*Figure 1—video 2*) and therefore receive input from all three PAM11 subtypes.

DAN subtype-specific wiring is also evident in the α′/β′ lobes (*Figure 30—figure supplement 2*). As an example, consider input to the thermo/hygrosensory α′/β′ap1 KCs. In the β′2a compartment, only PAM02-pd (β′2a) provides input to these KCs. Similarly, in β′2p, PAM05 DANs make abundant synapses onto the α′/β′ap1 KCs, but do not contact the α′/β′m KCs. Previous work has implicated β′2 DANs in thirst-dependent naive and learned water-seeking (*Lin et al., 2014b*; *Senapati et al., 2019*) and the pattern of connectivity described above suggests that these roles are likely to be served by specific DAN subtypes modulating specific streams of KCs.

Our analyses suggest that selective DAN subtype innervation of modality-specific streams of KCs is a general feature of all MB compartments in the horizontal lobe (*Figure 30—figure supplement 1*). Furthermore, every MB compartment that is innervated by multiple DANs contains more than one DAN subtype and generally one DAN subtype in each compartment receives direct MBON input, whereas the other subtypes do not. With DAN subtypes synapsing onto selective KCs and MBONs, this allows different populations of synaptic weights to be independently adjusted within each compartment. This arrangement is likely to provide significantly more computational bandwidth to each compartment in the MB network. We explore the functional implications of this arrangement in the Discussion (Figure 38C–D).

## Shared input neurons suggest functional grouping of DAN subtypes across compartments

Clustering DANs based on input similarity revealed groups of DANs innervating different compartments that had a significant fraction of their upstream neurons in common (*Figure 27*). To explore the neuronal pathways that provide such shared input, we selected the 50 neurons that make the largest number of synapses onto each DAN subtype. We subjected this population of 901 neurons to hierarchical clustering based on morphology, which generated 402 input neuron clusters. We then analyzed how these input neuron clusters connect to the different DAN subtypes. Interestingly, several of the input clusters contained only a single neuron. *Figure 31* shows 35 of these clusters, selected to illustrate the range of connectivity patterns; *Supplementary file 1* lists the constituent neurons for all 402 clusters. In the following paragraphs we highlight some of our key findings. *Figures 32–37* provide details of the neurons that make up several of these clusters.

Studies of aversive olfactory learning have established that key teaching signals are provided by the PPL101 (γ1pedc), PPL103 (γ2α′1), and PAM12 (γ3) DANs. Individually blocking output from these DANs impairs aversive memory formation and their forced activation can assign aversive valence to odors (*Aso et al., 2010*; *Aso et al., 2012*; *Hige et al., 2015*; *Perisse et al., 2016*; *Yamagata et al., 2016*; *Jacob and Waddell, 2020*). Our connectomics data shows that the PPL101 (γ1pedc), PPL103 (γ2α′1) and PAM12 (γ3) DANs share input, which supports the idea that they are driven in parallel in response to aversive/punishing cues. Several of the shared input pathways correspond to individual neurons that make synapses onto all of these aversively reinforcing DANs. These inputs provide the

clearest evidence to date that this collection of aversively reinforcing DANs can be triggered together (assuming that at least some of the inputs are excitatory). Sometimes PPL101 (γ1pedc) input neurons also connect to other DANs, which again provides insight into the functional relevance of recruiting ensembles of DANs. For example, the individual PPL101(γ1pedc) input neurons in clusters 1 and 4 and the group in cluster 13 provide weaker input to the PPL106 (α3) DAN. When stimulated alone during odor exposure, the PPL106 (α3) DAN requires more trials to code aversive learning than PPL101 (γ1pedc), consistent with it being relatively inefficient (*Aso and Rubin, 2016*). However, multiple spaced aversive training trials strongly depress the conditioned odor-response of MBON14 (α3) (*Jacob and Waddell, 2020*). We also found that PPL101 (γ1pedc) frequently shares input with PPL102 (γ1). Although, these DANs have overlapping innervation in the γ1 compartment, to date most attention has been focused on the PPL101 (γ1pedc) DAN. It will therefore be important to determine the role of PPL102 (γ1).

Whereas each aversively reinforcing PPL1 DAN is a single neuron, there are 10 – 11 PAM12 DANs innervating γ3 (one PAM12 appears to be incomplete in this dataset). An unexpected finding from this study is that the PAM12 (γ3) DAN population is comprised of at least two clear subtypes that are notable for the DANs with which they share input. Moreover, analysis of the inputs to the two PAM12 (γ3) DAN subtypes provides evidence that they convey opposite valence. Four single neuron clusters (clusters 1, 2, 4, and 5) preferentially connect to PAM12-dd (γ3) DANs and the negative-valence PPL1 DANs, PPL101 (γ1pedc), and PPL102 (γ1), implying that PAM12-dd might also signal negative valence. PAM12-md (γ3) DANs have very different connectivity defined by strong input from the four neurons in cluster 6 (*Figures 31*, *32D*), which also provides input to 3 of the 4 subtypes of PAM08 (γ4) DANs and PAM07 (γ4<γ1γ2) DANs. These shared inputs suggest that PAM12-md signals positive valence. It is noteworthy that prior studies have implicated PAM12 DANs in both aversive and appetitive memory. At least some PAM12 DANs are shock responsive (*Cohn et al., 2015*; *Jacob and Waddell, 2020*) and can provide aversive reinforcement (*Aso et al., 2012*), but it has also been shown that their inhibition following sugar ingestion is required and even sufficient to assign positive value to odors during learning (*Yamagata et al., 2016*). Furthermore, a recent study of spaced aversive conditioning showed that the γ3 DANs are required for flies to gradually learn that the odor that is experienced without shock is safe (*Jacob and Waddell, 2020*). It will be important to clarify how the different γ3 DANs contribute to these learning processes.

Prior functional studies have suggested that different aversively reinforcing stimuli, such as electric shock, high and low heat, and bitter taste converge onto the PPL101 (γ1pedc) and PPL103 (γ2α′1) DANs (*Das et al., 2014*; *Galili et al., 2014*; *Tomchik, 2013*). One might therefore expect to find connectivity consistent with such convergence. Interestingly, rather than finding clear streams of input to these PPL1 DANs and the PAM12 (γ3) DANs, the identified individual input neurons emanate from discrete brain areas, consistent with these neurons conveying unique information. Only one neuron in cluster 28 projects from the SEZ and connects to PPL101 (γ1pedc) and PPL106 (α3). It may therefore relay the negative valence of bitter taste. Besides this SEZ output neuron (SEZON) we did not find other obvious inputs from primary sensory processing areas, or ascending pathways that might come from the ventral nerve cord and could for example relay shock from the legs. It therefore seems likely that most information conveyed to aversively reinforcing DANs has been pre-processed en route from the periphery.

Input connectivity also revealed insights into the overall organization of the PAM DANs, uncovering a remarkable heterogeneity and revealing a highly parallel architecture of rewarding reinforcement. Prior studies have established that PAM DANs are required for flies to learn using different types of reward, and that their artificial activation (or inactivation in the case of γ3 DANs) can assign positive valence to odors during learning (*Liu et al., 2012*; *Burke et al., 2012*; *Perisse et al., 2013*; *Aso et al., 2014b*; *Huetteroth et al., 2015*; *Yamagata et al., 2015*; *Yamagata et al., 2016*). In addition, a number of studies have implicated unique populations of PAM DANs in reinforcing different positive experiences, for example, the sweet taste and nutrient value of sugar (*Huetteroth et al., 2015*; *Yamagata et al., 2015*), water (*Lin et al., 2014a*; *Shyu et al., 2017*), relative value (*Perisse et al., 2013*), the absence of expected shock (*Felsenberg et al., 2018*) and learned safety (*Jacob and Waddell, 2020*).

We found several input pathways that group the PAM DANs into both expected and unforeseen combinations. For example, the single neurons of clusters 23 (a SEZON) and 24 grouped the PAM08-md and PAM08-nc (γ4) DANs with the four subtypes of γ5 DANs. The cluster 24 neuron also

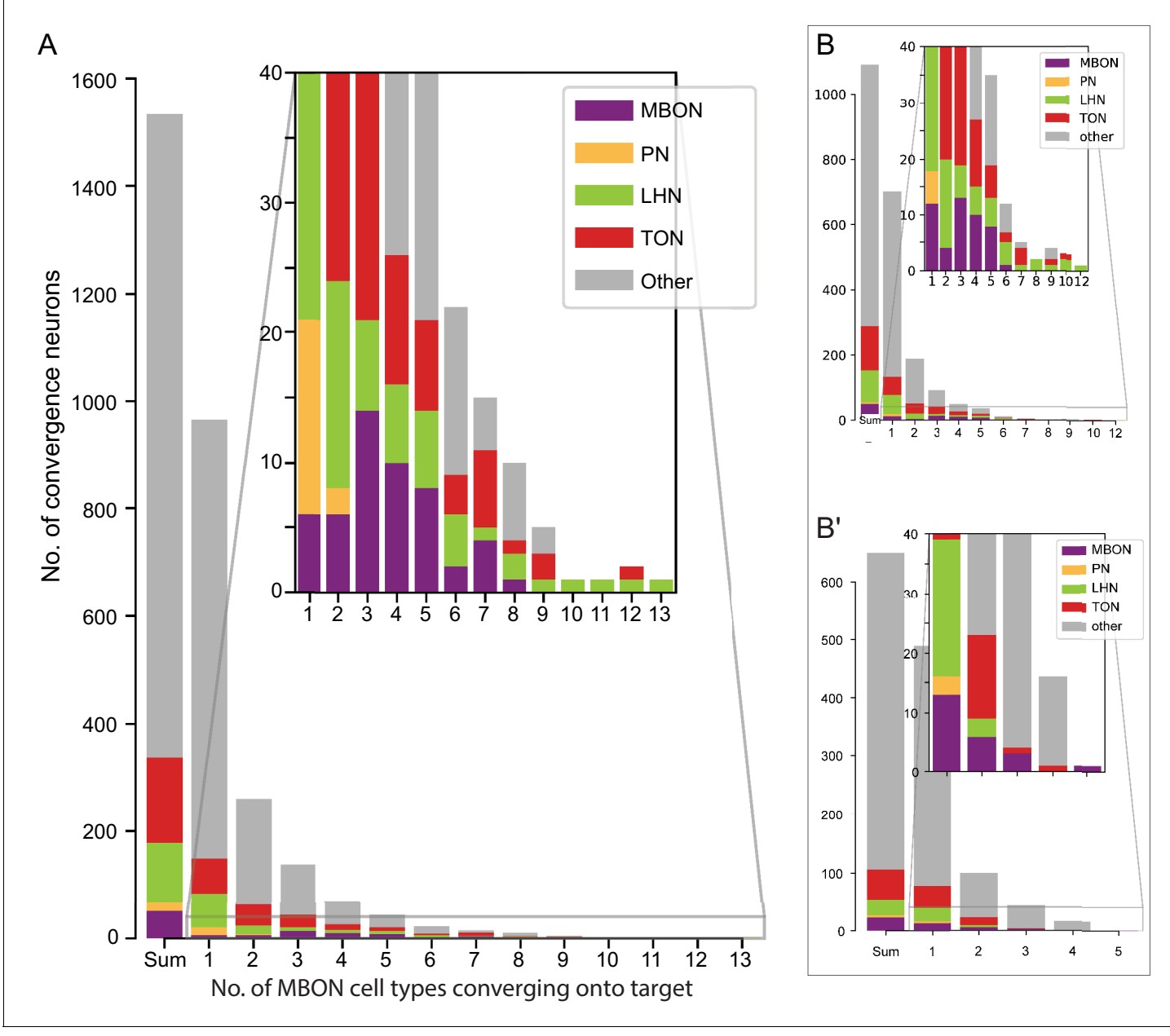

**Figure 21.** Downstream targets of MBONs often receive input from more than one MBON. (**A**) About 1550 neurons are downstream (including axo-axonal connections) of one or more of the 34 MBON cell types when a threshold of 10 synapses is used. Among them, about 600 are downstream of at least 2 MBON cell types and are therefore capable of integrating information from multiple MBONs. The inset shows an expanded view of the indicated portion of the plot. These neurons include a range of well-characterized neuron types, including MBONs, olfactory projection neurons (PNs), lateral horn neurons (LHNs), and third-order olfactory neurons (TONs; downstream targets of PNs outside the LH, see *Schlegel et al., 2020*) as well as less characterized neurons (other). Among the neurons that receive input from 10 or more MBON types, LHNs are heavily over-represented. (**B-B'**) The distribution shown in (**A**) has been divided into two separate histograms, one showing neurons downstream of multiple typical MBONs, in which atypical MBONs have not been counted (**B**), and one showing neurons downstream of multiple atypical MBONs, in which typical MBONs have not been counted (**B'**). The insets show expanded views of the indicated portions of each plot. Atypical MBONs show less convergence than the typical MBONs and do not preferentially converge onto LHNs.

The online version of this article includes the following figure supplement(s) for figure 21:

**Figure supplement 1.** A LH neuron downstream of seven MBON types.

**Figure supplement 2.** A neuron downstream of 11 MBON types.

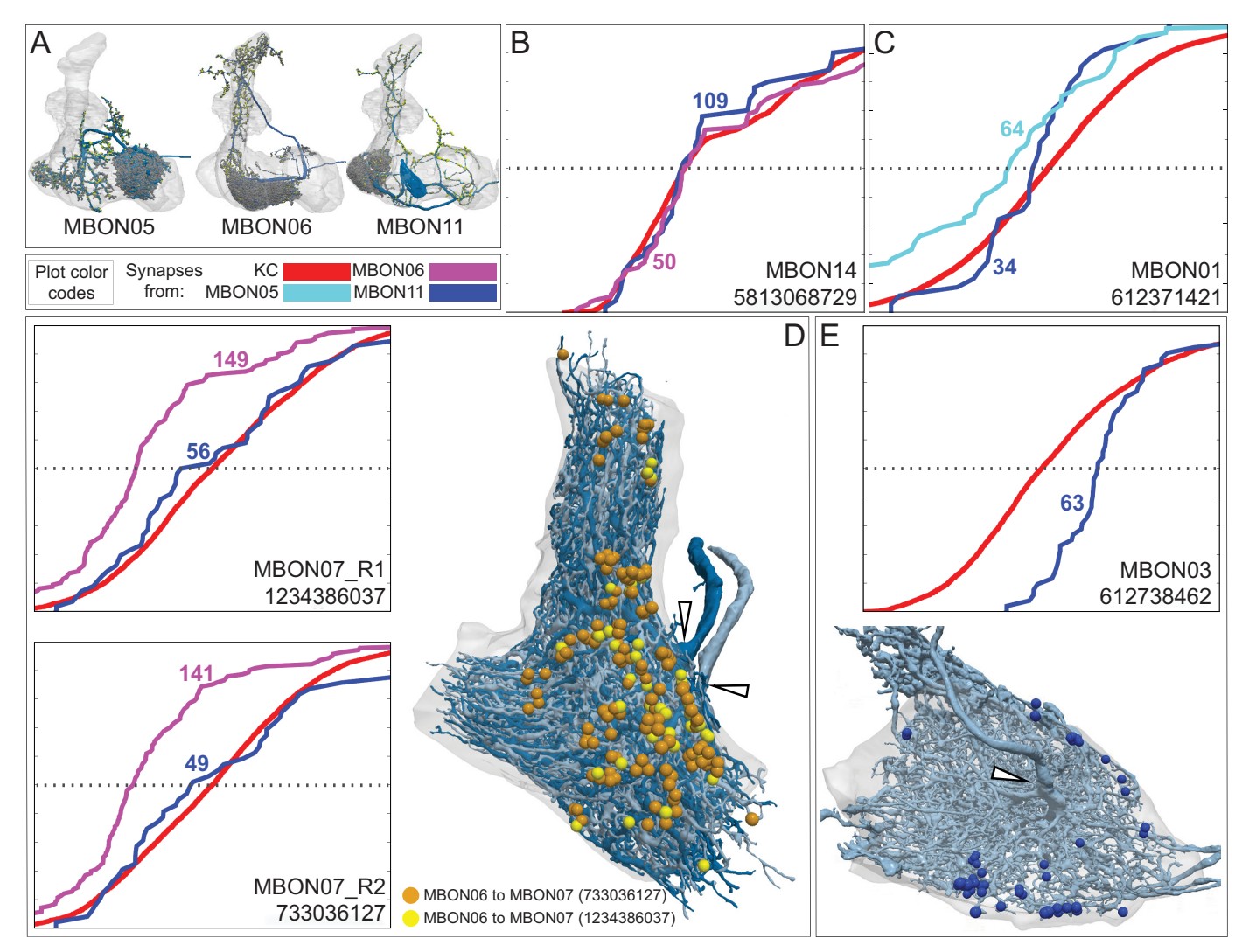

**Figure 22.** Spatial distribution of synapses from feedforward MBONs onto the dendrites of other MBONs. (**A**) The three MBON types that make feedforward connections to other compartments within the MB lobes are shown. Neurons are shown in blue with presynaptic and postsynaptic sites shown as yellow and gray dots, respectively. (**B–E**) The cumulative fraction of synaptic connections from a feedforward MBON to its target is plotted as a function of the normalized distance from the root of the target MBON's dendritic tree, on a linear scale from 0 to 1. The name of the target neuron is given in the lower right corner of the panel. The horizontal dotted line indicates the distance from the root of the target MBON's dendritic tree at which half of the synapses from KCs to that MBON have occurred. The lines in these graphs are color-coded as indicated by key shown below panel A. The numbers indicate the total number of synapses between the neuron of the same color and the target MBON. (**B**) The synapses from MBON06 (β1>α) and MBON11 (γ1pedc>α/β) onto MBON14 in α3 have a similar spatial distribution as MBON14's synapses from KCs, which tend to be uniformly distributed along MBON dendrites. (**C**) MBON01's dendrites in γ5 receive synaptic connections from MBON11 closer to MBON01's dendritic root than KC inputs, a bias even more pronounced in the distribution of synapses MBON01 (γ5β'2a) receives from MBON05 (γ4>γ1γ2). (**D**) The two MBON07 cells in the α1 compartment each receive synaptic input from MBON06 (β1>α) at sites close to the root of their dendrites (hollow arrowheads); in contrast, MBON11's synapses onto the MBON07s have a similar distribution as the input the MBON07s receive from KCs. (**E**) MBON03 (β'2mp) receives synaptic input from MBON11 shifted further from its dendritic root (hollow arrowhead) than the input it receives from KCs.

The online version of this article includes the following video and figure supplement(s) for figure 22:

**Figure supplement 1.** MBON06 and MBON11 make reciprocal axo-axonal connections in the α lobe.

**Figure 22—video 1.** Feedforward MBON05.

https://elifesciences.org/articles/62576#fig22video1

**Figure 22—video 2.** Feedforward MBON06.

https://elifesciences.org/articles/62576#fig22video2

**Figure 22—video 3.** Feedforward MBON11.

https://elifesciences.org/articles/62576#fig22video3

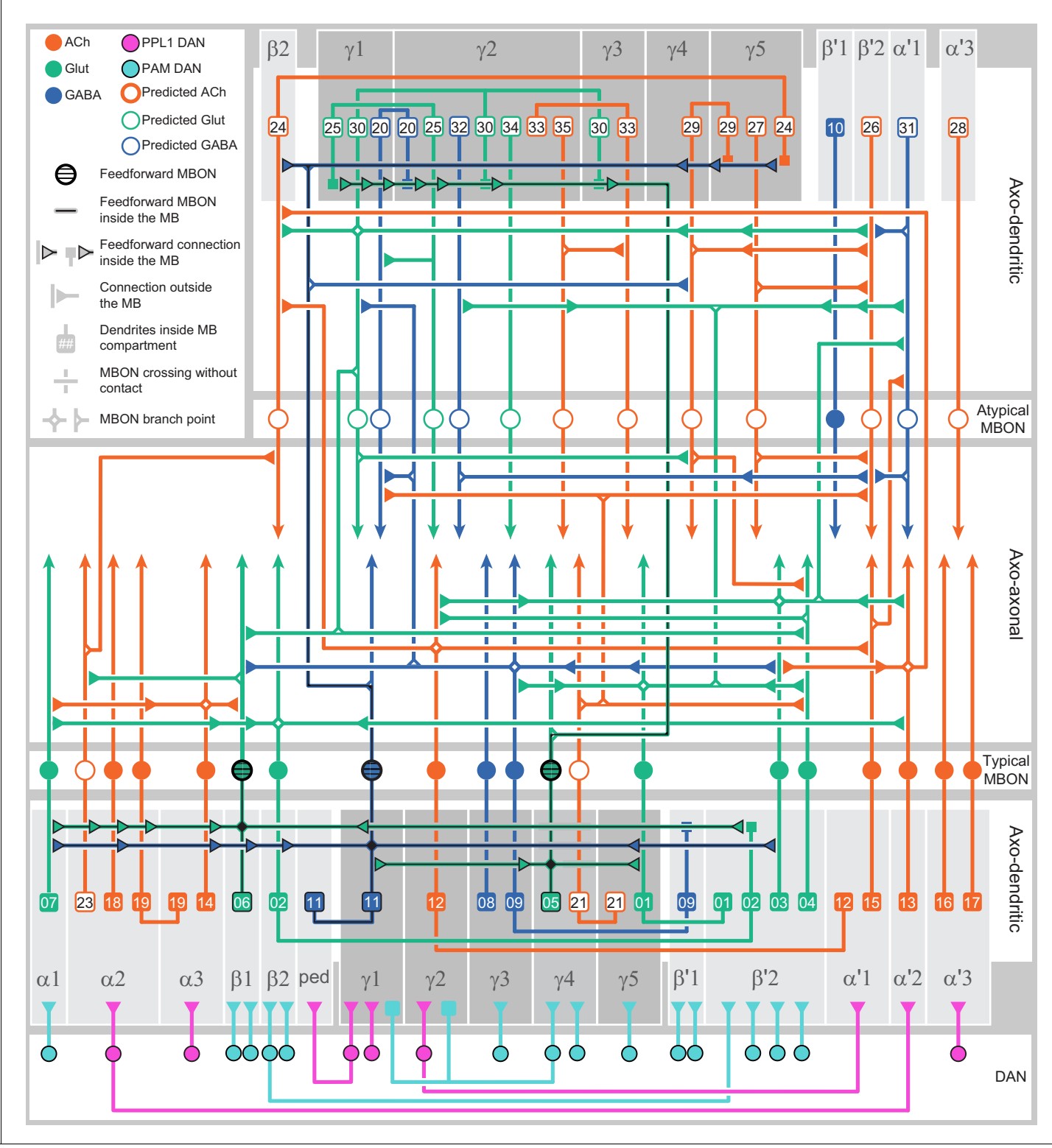

**Figure 23.** Diagram of the connections made between MBONs. A diagram showing MBON-to-MBON connections. At the bottom, gray boxes representing each of the MB compartments and the core of the distal peduncle (pedc). DAN inputs, with PPL1 and PAM DANs color-coded, are shown below the compartments (see *Figure 6—figure supplement 1* for the names of these cell types). MBON dendrites are represented as squares that contain the identification number of the MBON cell type, color-coded by neurotransmitter; see *Figure 7* for more information about these MBONs. At the top, gray boxes representing the subset of MB compartments that house the dendrites of atypical MBONs are shown; the identification numbers of the MBON cell types are given in the squares. These MBONs are described in more detail in *Figure 8*. Outputs from MBONs are shown

*Figure 23 continued on next page*

Figure 23 continued

outside the gray boxes, with the exception of three MBONs—MBON05 (γ4>γ1γ2), MBON06 (β1>α) and MBON11 (γ1pedc>α/β)—indicated by the heavier outline and striped circles representing their cell bodies, send axonal terminals (triangles) back into the MB lobes. These MBONs are described in more detail in *Figure 22* and *Figure 22—video 1*, *2* and *3*. Axo-dendritic and axo-axonal connections between MBONs are diagrammed separately, as indicated.

The online version of this article includes the following figure supplement(s) for figure 23:

**Figure supplement 1.** Axo-axonal connections among typical MBONs outside the MB presented as a connection matrix.

**Figure supplement 2.** Axo-axonal synapses between MBONs can be highly localized.

**Figure supplement 3.** Morphology of axo-axonal synapses.

provides input to a selection of β and β′ lobe innervating DANs. Cluster 7 neurons grouped a different collection of γ4 and γ5 with the PAM08-md (γ3) DANs and most PAM02 (β′2a), PAM05 (β′2p), PAM06 (β′2m) and PAM13 (β′1ap) and PAM14 (β′1m) DANs. These inputs are consistent with studies that have implicated the γ4, γ5 and β′2 compartments in reward learning, such as learning reinforced with water and the taste of sugar (*Aso and Rubin, 2016*; *Burke et al., 2012*; *Huetteroth et al., 2015*; *Lin et al., 2014b*; *Liu et al., 2012*; *Shyu et al., 2017*; *Yamagata et al., 2015*). In contrast, clusters 19 and 20 group unique subtypes of γ5 DANs with all three α1 DAN subtypes. Cluster 19 also provides strong input to both types of PAM13 DANs and the PAM14-can subtype. Prior work has shown that the γ5 and α1 DANs are required for the reinforcement of nutrient-dependent long-term sugar memory (*Huetteroth et al., 2015*; *Yamagata et al., 2015*; *Ichinose et al., 2015*). A similar complexity of combinatorial inputs is evident across most of the DAN subtypes in the PAM DAN population.

An unexpected finding was the strong and unique grouping of all subtypes of PAM04 (β2), two subtypes of PAM10 (β1) and the PAM09-vd (β1ped) subtype by the population of neurons in cluster 8. Two prior studies have shown that artificial activation of β1 and β2 DANs together can form a long-lasting appetitive memory (*Perisse et al., 2013*; *Huetteroth et al., 2015*) and the cluster 8 input suggests that their activity is likely to be genuinely coordinated. The full importance of these DANs is not currently understood, but our analysis of MBON06 (β1>α) suggests that appetitive learning via these DANs modulates a key node of the MBON network (see Discussion).

The PAM11 (α1) DANs reinforce nutrient-dependent, long-term memory (*Huetteroth et al., 2015*; *Ichinose et al., 2015*; *Yamagata et al., 2015*). Their connectivity differs from other DAN cell types in that they receive input from a few clusters of input neurons that are very selective for PAM11 DANs but do not strongly discriminate between the three α1 DAN subtypes (*Figures 31*, *33*). For example, cluster 22 is a single neuron that provides strong, highly selective input to all three PAM11 subtypes and appears to convey input from the SEZ (*Figure 33B*). Clusters 19 and 20 convey information from the LH (*Figure 33E and F*). When α1 DAN input neurons do show connections to other DANs, this input is much weaker and largely confined to positive-valence DANs (*Figure 31*).

We uncovered several additional features of the organization of the DAN system (*Figures 34–37*). Some input clusters synapse with multiple cell types, but only with specific DAN subtypes within them (*Figure 36*). There are many PAM specific clusters, such as clusters 7, 29, 32 (*Figure 36—figure supplement 1*) as well as PPL1 specific clusters, such as clusters 13, 14, and 15 (*Figure 36—figure supplement 2*). We also identified several clusters of neurons that innervate subsets of both PAM and PPL1 DANs (clusters 3, 9, 30, 33, 35; *Figure 37* and *Figure 37—figure supplement 1*). Some clusters, such as 3, 7 and 30, have such dense innervation that their most notable feature might be the DANs that they do not innervate.

Two discrete classes of dorsal fan-shaped body (FB) neurons make up clusters 10 and 31, which provide selective input to DANs of opposite valence (*Figures 31*, *34*). The arbors of the FB neurons where they synapse to these DANs are of mixed polarity and could simply serve as a local relay for other inputs. We also note that (*Hu et al., 2018*) reported that more ventral layers of the FB respond to electric shock and their artificial activation could substitute for negative-valence reinforcement, but it remains unclear what role, if any, more dorsal FB layers might play.

Neurons from the SEZ provide a major source of DAN inputs. While studies have shown that pleasant and unpleasant tastes can reinforce learning by engaging different DAN subsets (*Das et al., 2014*; *Huetteroth et al., 2015*; *Kirkhart and Scott, 2015*; *Masek et al., 2015*), the relevant input pathways are only recently emerging from connectomic studies. A comparison of inputs

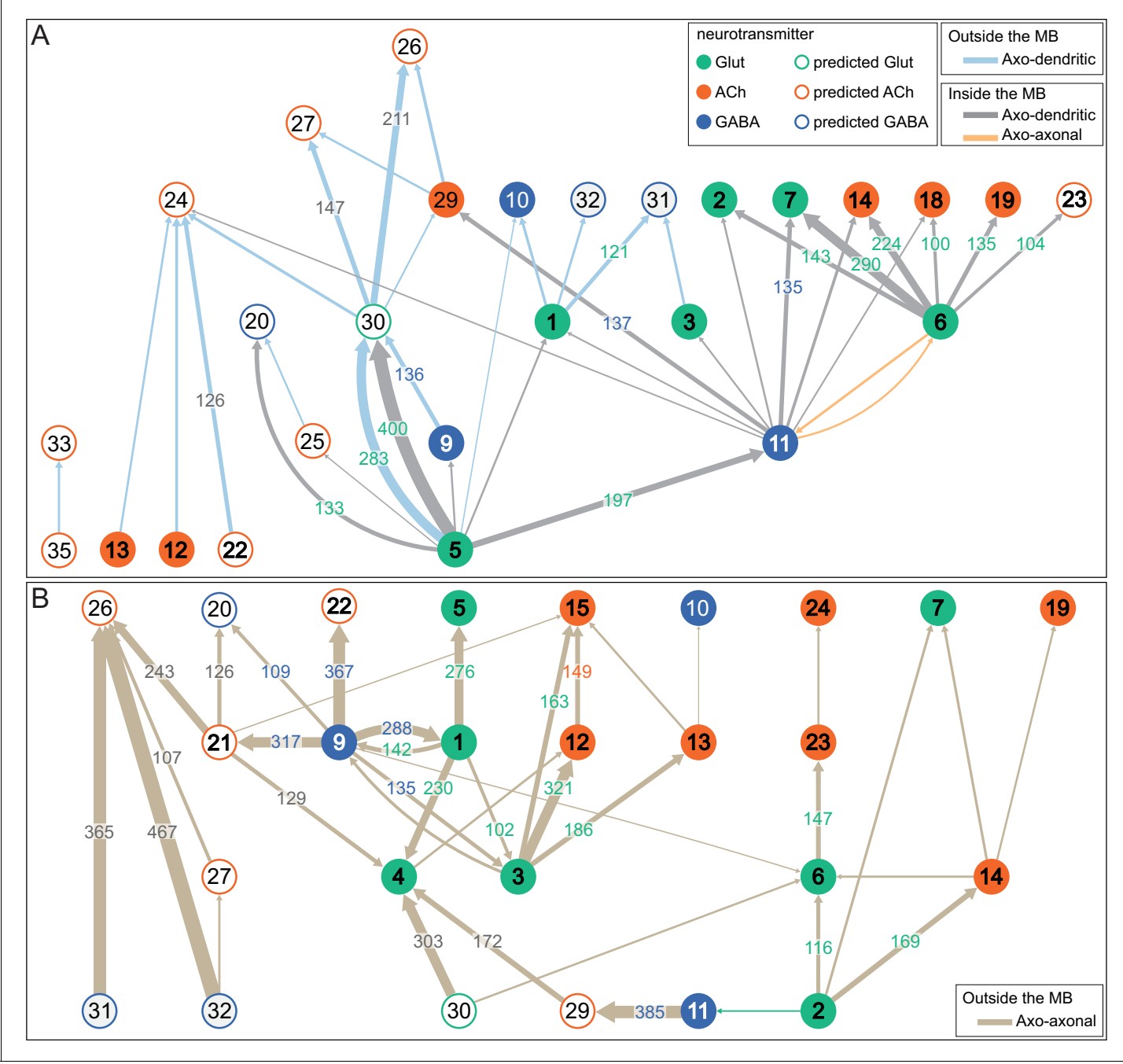

**Figure 24.** Connections between MBONs form a multi-layered feedforward network. Connections between MBONs, including both typical and atypical MBONs are shown, using a threshold of 30 synapses. The color of the arrow, as indicated in the color key, represents the nature and location of the synaptic connection. The number of synapses made in any given connection is indicated by the number in the arrow and the arrow thickness is proportional to that number. Numbers have been pooled for cases where there are two MBONs of the same cell type but only connections that occur in the right hemisphere are shown. The numbers within the circles indicate the MBON cell type; for example 1 represents MBON01. Typical MBONs are numbered with bold numerals. Circles are color-coded according to their neurotransmitter. (**A**) Axo-dendritic connections outside the MB lobes, axo-dendritic connections inside the MB lobes provided by feedforward MBONs (MBON05, MBON06, and MBON11), and the reciprocal axo-axonal connections between MBON6 and MBON11 within the MB lobes. (**B**) Axo-axonal synapses outside the MB lobes. Note that some MBONs, such as MBON24, form both axo-dendritic and axo-axonal connections outside the MB. Additional details of MBON05's innervation of MBON30 are provided in *Figure 24—figure supplement 1A*.

The online version of this article includes the following figure supplement(s) for figure 24:

**Figure supplement 1.** MBON30 displays an unusual pattern of innervation from other MBONs and FB neurons.

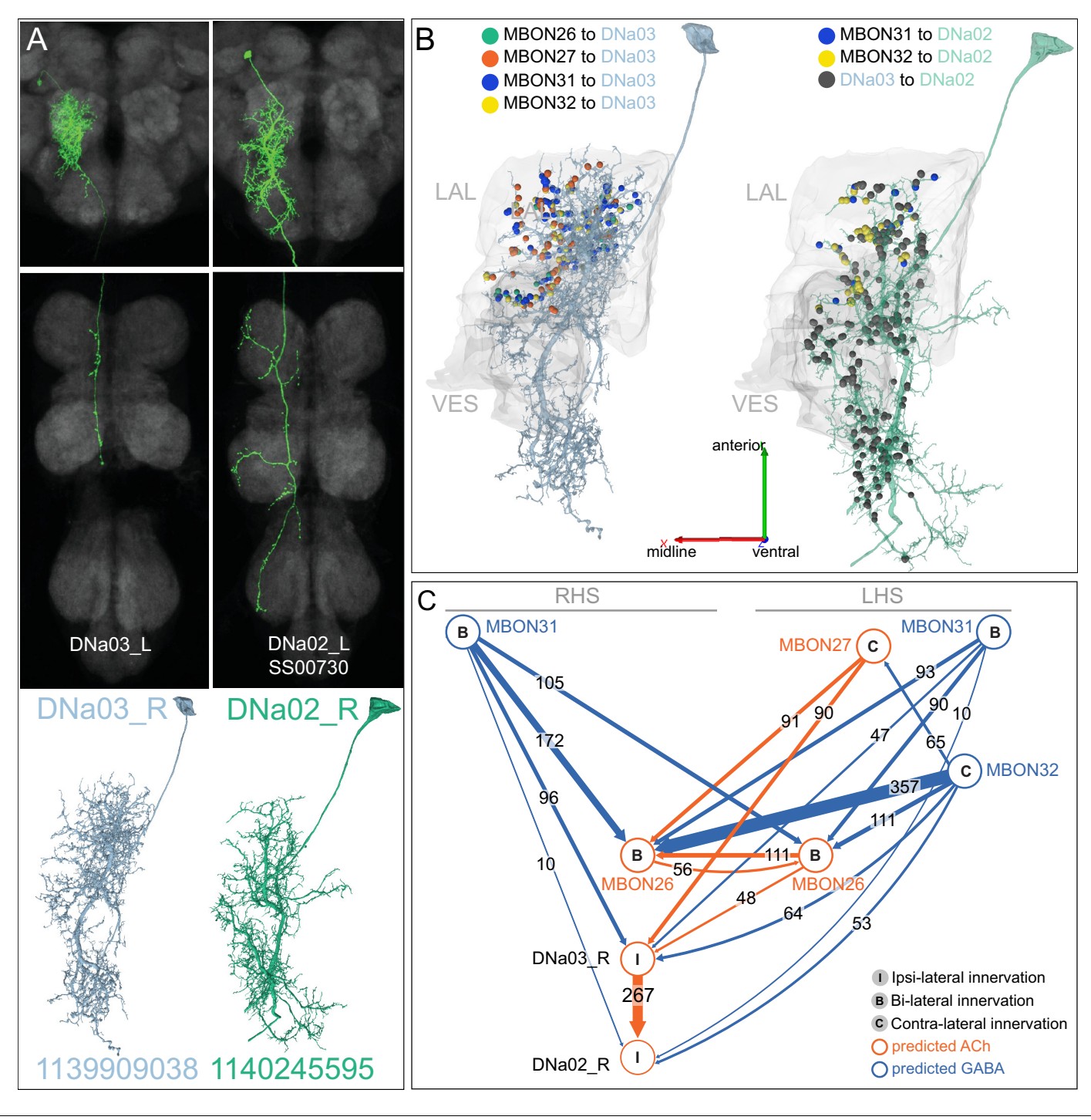

**Figure 25.** A descending pathway from atypical MBONs to the ventral nerve cord (VNC). Four LAL-innervating atypical MBONs form strong connections to descending neurons (DN). (**A**) A comparison between LM (***Namiki et al., 2018***) and EM morphologies of DNa02 and DNa03. Note that the LM images show the cells in the left hemisphere while the EM images are from the right hemisphere and that the axons of the DNs extend outside the hemibrain volume. (**B**) Synaptic connections from MBON26, MBON27, MBON31 and MBON32 to DNa03 (left) and from MBON31, MBON32, and DNa03 to DNa02 (right) are shown, color-coded. Note the connections from MBONs are restricted to the LAL while DNa03 makes synapses onto DNa02 throughout its arbor in the LAL and the VES. (**C**) Diagram of connectivity between MBONs, DNa03 and DNa02. Note that MBON27 and MBON32 have contralateral axons (***Figure 8—figure supplements 6*** and ***11***, ***Figure 8—video 6*** and ***11***) and thus innervate the contralateral DNa02 and DNa03 while MBON26 and MBON31 have bilateral output (***Figure 8—figure supplements 5*** and ***10***, ***Figure 8—video 5*** and ***10***). Thus the connectivity diagram shows the inputs onto the DNa02 and DNa03 of the right hemisphere (RHS), while many of the neurons providing input have their

*Figure 25 continued on next page*

*Figure 25 continued*

cell bodies in the left hemisphere (LHS). MBONs 26, 31, and 32 show similarity in their PN inputs and overall outputs (see *Figure 16—figure supplement 2*). Other inputs to DNa02 and DNa03 are summarized in *Figure 25—figure supplement 1*. Predicted neurotransmitters are indicated. In addition to the diagrammed connections, these MBONs connect to the DNa02 and DNa03 using paths that include interneurons (see *Figure 42*). DNa02 has been implicated in steering control (*Rayshubskiy et al., 2020*) and the outputs of these MBONs are likely to play a role in guiding approach and avoidance (see *Figure 42*).

The online version of this article includes the following figure supplement(s) for figure 25:

**Figure supplement 1.** Top inputs to DNa02 and DNa03.

**Figure supplement 2.** Central complex and LO inputs to DNa03.

to PPL101 (γ1pedc), PAM01 (γ5) and PAM05/6 (β′2) DANs identified a set of SEZONs. The hemibrain dataset provides access to the projections of all SEZONs (although characterizing their dendritic arbors required identifying the corresponding neurons in the FAFB volume) and confirms that they are a major source of DAN input (clusters 22 – 28) (*Figure 35* and *Figure 35—figure supplement 1*). Some single neuron SEZON clusters tend to connect to either exclusively positive valence (clusters 22, 23) or negative-valence (cluster 27) DANs, consistent with them relaying the valence of different tastants to the MB. Some clusters contain multiple SEZONs (clusters 25, 26 and 28); clusters 25 and 26 innervate only positive-valence DANs, while SEZONs in cluster 28 innervate either PPL101 or DANs of both positive and negative valence.

As with the PPL1 DANs, most neurons providing strong input to PAM DANs do not project from primary sensory areas in the brain. PAM DANs are therefore also likely to receive highly processed value-based information. However, as discussed above for PPL101 and previously noted for the PAM01 (γ5) and PAM02 (β′2a) DANs (*Otto et al., 2020*), we found that several PAM DANs receive significant input from SEZONs. Since blocking some of these neurons has been shown to impair sugar and/or bitter taste learning (*Otto et al., 2020*), SEZON input suggests that taste may provide fairly direct valence-related signals to the DANs. Various subtypes of PAM01, PAM02, PAM08, PAM09, PAM10 and PAM15 DANs all receive SEZON input (*Figure 35* and *Figure 35—figure supplement 1*), some of which is highly selective. The single SEZON in cluster 22 exclusively synapses onto the three subtypes of PAM11 (α1) DANs (*Figure 33B*), whereas another (cluster 27) synapses selectively with the PPL106 (α3) DAN (*Figure 35*). We speculate that the prevalence of SEZON innervation to the DANs indicates the ecological relevance of evaluating taste for the fly, and that these very selective pathways to PAM11 (α1) and PPL106 (α3) might represent stimuli that are of unique importance. PAM11 DANs have been previously implicated in the nutrient-dependent reinforcement of long-term memories. In addition to the recently reported involvement of SEZONs in sugar learning (*Otto et al., 2020*), an earlier study demonstrated the importance of a physical and functional connection between octopaminergic neurons and the PAM01 (γ5) and PAM02 (β′2a) DANs (*Burke et al., 2012*). Also within the top 50 inputs, we identified synapses from the OA-VPM3 neurons onto PAM01 (γ5), PAM08 (γ4) and to a lesser extent PAM02 (β′2a) DANs. It will be important to understand the relationship between the OA and SEZON inputs to the rewarding DANs.

Our work provides a comprehensive description of the substructure of DANs, uncovering a complexity that goes far beyond the previously defined 21 cell types (*Aso et al., 2014a*). Understanding the functional significance of this complexity will require more complete knowledge of the information about the outside world and internal brain state that is conveyed by the hundreds of neurons that provide input to DANs. Our analyses of the DAN connectivity, together with what we learned about the segregation of sensory modalities in different classes of KCs, the selective connectivity of MBONs to KCs, and the discovery of a new class of MBONs, opens a window onto a much broader and richer landscape of MB circuitry underlying what a fly might be able to learn, remember and use to guide its behavior.

## Discussion

The hemibrain dataset we have analyzed contains the most comprehensive survey of the cell types and connectivity of MB neurons available to date. This connectome allowed us to probe in fine detail the circuitry underlying canonically proposed functions of the MB, including the representation of olfactory information by KCs, computation of valence by MBONs, and reinforcement of associations

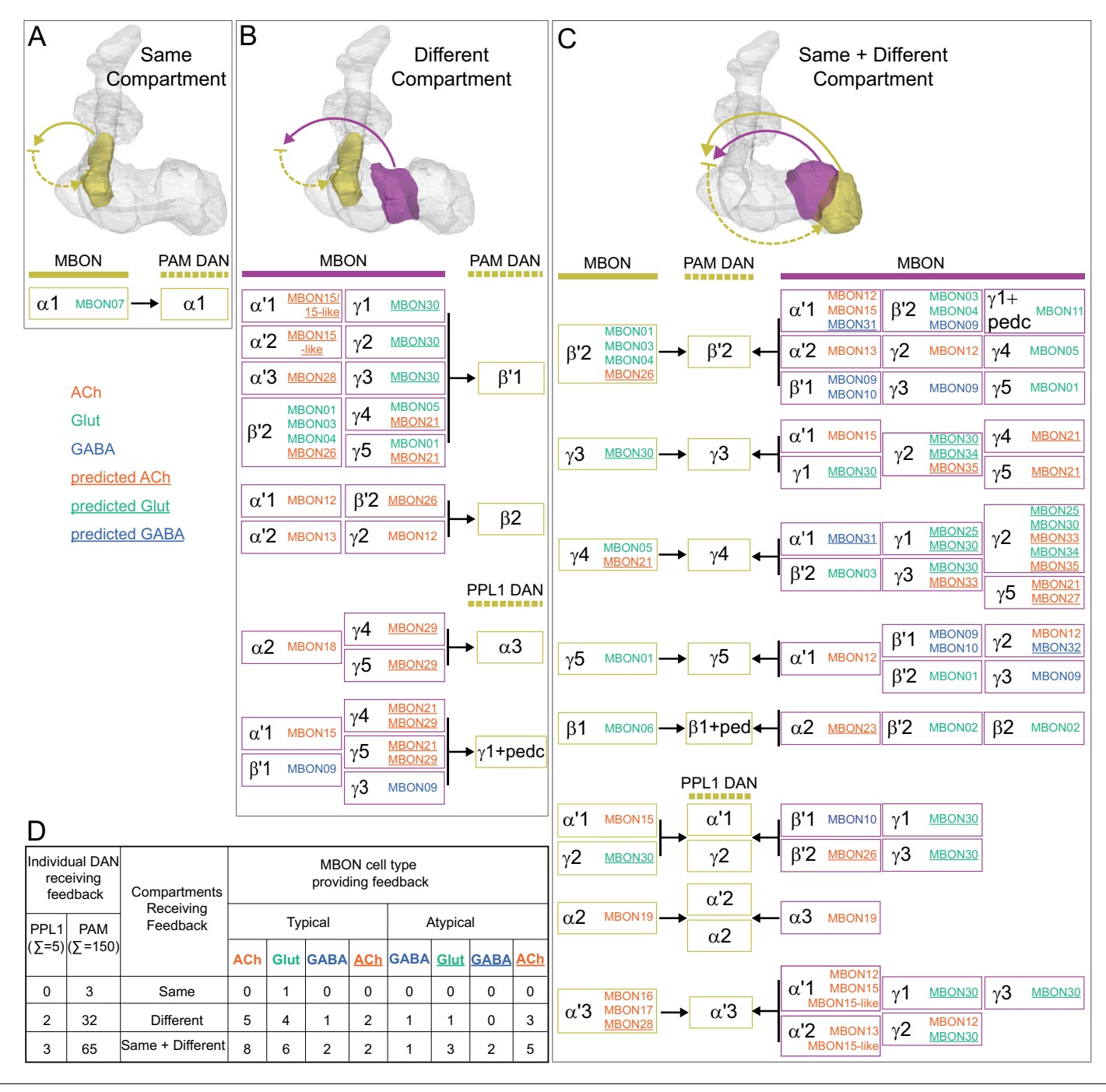

**Figure 26.** Feedback from MBONs to DANs. Input from MBONs onto DANs occurs in three distinct circuit motifs which are diagrammed. MBON connections to DANs from the same compartment are depicted as a yellow arrow, while MBON connections to DANs from a different compartment are depicted as a purple arrow. DANs downstream of these MBONs are depicted as dashed yellow arrows. (**A**) In the simplest case, the MBON and DAN form a reciprocal loop where the MBON innervates DAN(s) that send their axons back to the same compartment where the MBON's dendrites arborize. Below the diagram of this circuit motif, each of the instances we observed with this arrangement is presented; inputs onto PAM and PPL1 DANs are presented separately. The color of the MBON name indicates its neurotransmitter, as indicated. (**B**) In this case, MBON(s) synapse onto DAN(s) that send their axons to MB compartment(s) different from the compartment where the MBON's dendrites reside, providing a mechanism for cross-compartment communication. It is common for the DAN population of a compartment to receive input from several MBONs. (**C**) This motif combines the previous two motifs, with DANs receiving input from both MBON(s) of same and different compartments. (**D**) The table summarizes DAN populations that participate in the three different feedback motifs.

The online version of this article includes the following figure supplement(s) for figure 26:

*Figure 26 continued on next page*

*Figure 26 continued*

**Figure supplement 1.** Axo-axonal connections from MBON05, MBON06 and MBON11 onto DANs inside MB lobes.
**Figure supplement 2.** MBON-to-DAN feedback mediated by an interneuron.
**Figure supplement 3.** Distribution of DAN feedback mediated by an interneuron.

by DANs. We found patterns in the input to DANs, MBON-to-DAN and MBON-to-MBON connectivity that suggest how associative learning in the MB can affect both the acquisition of new information through learning and the expression of previously learned responses. The connectome also reveals circuitry that supports non-canonical MB functions, including selective structure in non-olfactory pathways, a network of atypical MBONs, extensive heterogeneity in DAN inputs, and connections to central brain areas involved in navigation and movement.

## Advantages, limitations and challenges

Our analysis of the hemibrain connectome relied heavily on an extensive catalog of previously identified and genetically isolated cell types and on decades of study illuminating the link between MB physiology and fly behavior. It is worth emphasizing the interdependency of anatomy, physiology and behavior as we enter the post-connectomic era in fly research. Some of the neurons we have described that appear similarly connected may turn out to have diverse functions due to different physiology (*Groschner et al., 2018*) and, conversely, neurons that are morphologically distinct may turn out to have similar functions. In addition, a set of inputs with high synapse counts might appear, at the connectome level, to represent a major pathway for activating a particular neuron, but this will not be true if these inputs rarely fire at the same time. Likewise, a set of highly correlated inputs can be effective even if their individual synapse counts are modest. Lack of knowledge about correlated activity is probably the most significant uncertainty when attempting to map synapse counts onto circuit function, likely larger than possible errors in synapse identification and the effects of imposing various thresholds (see below). Finally, the connectome does not reveal gap junction connections or identify more distant non-synaptic modulation. These caveats should be kept in mind when interpreting connectome data (*Bargmann and Marder, 2013*).

A number of our studies involved imposing a cutoff on the number of synapses required to include a particular connection in the analysis. The intent of these thresholds is to focus the analysis on what are likely to be the strongest inputs and outputs. Whether this cutoff should be based on a fixed number of synapses or on a percentage of total synapse counts is open to debate as, of course, is the actual value that the cutoff should take. Neurons differ widely in their numbers of inputs and outputs and these differences need to be taken into account when choosing thresholds, as illustrated in *Figure 6—figure supplement 3* for MBON outputs and DAN inputs. That is why we often present information about percentage of total inputs and outputs as well as synapse numbers. An approach to thresholding that takes such information into account has been used by *Hulse et al., 2020*; in the case of MBON to CX connections explored both here and in *Hulse et al., 2020*, differences in the connectivity map obtained with different thresholding methods were limited to the weakest connections.

Most neurons make a large number of weak connections to other neurons, often involving just a single synapse. Some cases of low synapse number may be the result of incomplete reconstruction of neuronal arbors. In the MB lobes, an extensive effort was made to fully reconstruct all neurites and >80% of all computationally predicted synapses were assigned to identified neurons. Likewise, the arbors of MBONs and DANs outside the MB were extensively proofread. However, in other brain regions, where MBON outputs and DAN inputs lie, reconstruction was often much less complete (see Table 2 in *Scheffer et al., 2020*). In such regions, it is difficult to estimate the extent to which the mapped synapses are representative of the full set of connections. *Hulse et al., 2020* report an analysis of connectivity determined at two stages of reconstruction in the CX and, while the number of assigned synapses increased with additional proofreading, there was little difference in the connectivity maps. Errors in either synapse prediction (*Scheffer et al., 2020*) or developmental wiring (see *Takemura et al., 2015*) are also likely to produce some false connections represented by only one or two synapses. Of course, real, but sparse, connections might still be impactful if they fire concurrently, but this is not something we can judge from information that is available to us. Therefore,

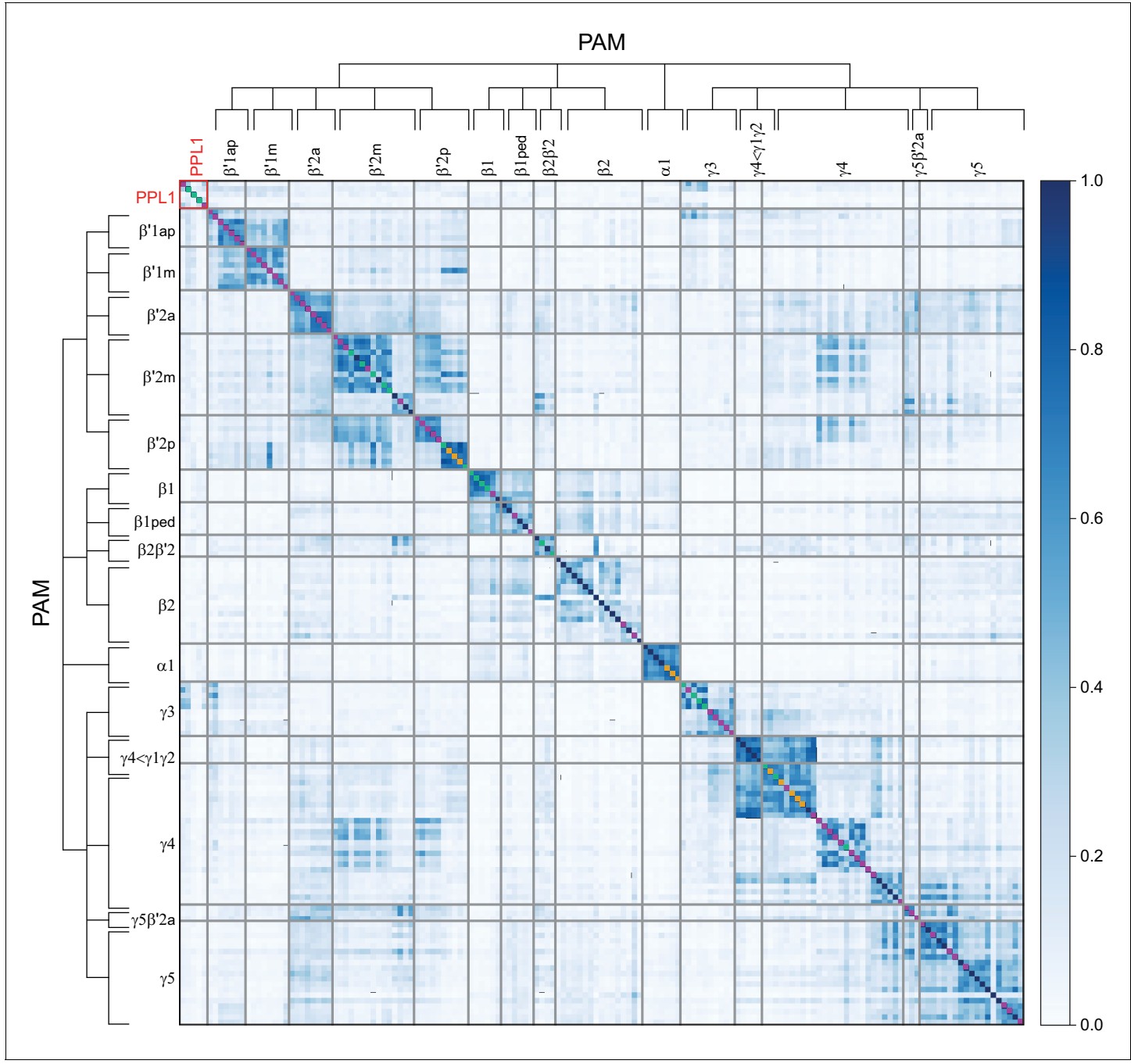

**Figure 27.** Similarity of input to individual DANs. Heatmap representing the similarity of inputs received by DANs. Each square represents the cosine similarity of inputs received by each PAM or PPL1 DAN. In order to focus on inputs received outside the lobes of the MB, inputs from KCs, other DANs, APL, and DPM neurons have been excluded. DANs are grouped by type, and the ordering within each type is determined by spectral clustering. Colors on the diagonal indicate whether the given DAN receives feedback from MBONs in the same compartment (yellow), MBONs from different compartments (purple), both (green), or neither (dark blue) as defined in *Figure 26*. *Figure 27—figure supplement 1* shows an expanded view of the PPL1 DAN portion of the heatmap. *Figure 27—figure supplement 2* shows the average input similarity between DAN cell types computed after pooling the data for all cells of a given DAN cell type.

The online version of this article includes the following figure supplement(s) for figure 27:

**Figure supplement 1.** Similarity of inputs to PPL1 DANs.

**Figure supplement 2.** Similarity of inputs to DAN cell types.

**Figure supplement 3.** DAN input distribution by brain region.

**Figure supplement 4.** Comparing DAN inputs and MBON outputs reveals credit assignment by DANs.

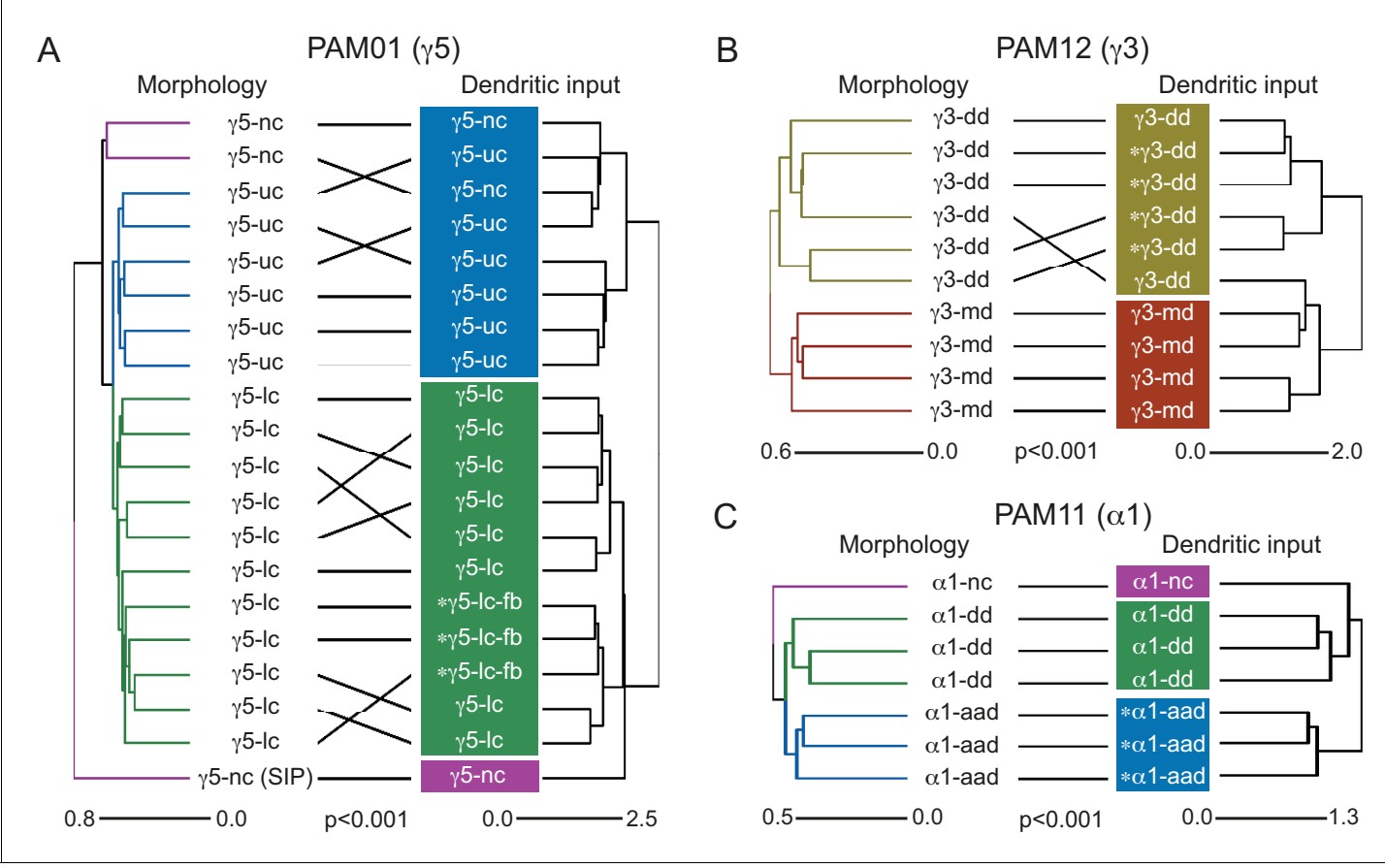

**Figure 28.** Morphologically defined DAN subtypes receive similar input. Tanglegrams show that DAN subtyping based on hierarchical morphological clustering matches that generated by hierarchical clustering by input connectivity (compare this figure to DAN input similarity shown in *Figure 27*). We show data here for three compartments; we found that clustering by morphology or input connectivity gave similar results in all PAM DAN compartments (not shown). A complete list of PAM subtypes is given in *Supplementary file 1*. (A–C) Tanglegrams for right hemisphere PAM01 (γ5), PAM12 (γ3), and PAM11 (α1) DANs. The left hand dendrograms represent hierarchical clustering by morphology and colors mark neurons clustered together (morphologies shown in *Figure 28—figure supplement 1* and *Figure 30*). Right hand dendrograms represent hierarchical clustering by dendritic input connectivity. Colored boxes mark DAN subtype clusters defined by input similarity (compare to *Figure 27*). Asterisks denote neurons that receive MBON feedback from the same compartment, as discussed in *Figures 26,27*. (A) PAM01 DANs cluster into subtypes. Two morphological types are categorized by their contralateral commissure tract: upper commissure (uc) and lower commissure (lc) (*Figure 28—figure supplement 1A and B*). Three neurons within the lower commissure subtype receive MBON01 (γ5β′2a) feedback (asterisk) and have distinct connectivity and axon morphology features (*Figure 28—figure supplement 1D*). Two neurons within the first morphological subtype are non-canonical (nc) with significantly different dendritic or axonal branching patterns (*Figure 28—figure supplement 1C*). Subtypes are consistent with those described in *Otto et al., 2020*. Connectivity clustering produces three subtypes by grouping the two non-canonical neurons with the uc subtype. Connectivity and morphology clustering are not significantly independent (Mantel test: r = 0.764, p < 10$^{-7}$, n = 10$^{7}$). (B) PAM12 DANs cluster into two subtypes mostly defined by their dendritic field anatomy: dorsal dendrite (dd), and medial dendrite (md) (*Figure 28—figure supplement 1E–F*). Clustering by input connectivity also produces two subtypes. Connectivity and morphology clustering are not statistically independent (Mantel test: r = 0.838, p = 2.76 × 10$^{-7}$, n = 3.6 × 10$^{6}$). (C) PAM11 DANs cluster into three morphological subtypes. Two are distinguishable by their dendritic anatomy: dorsal dendrite (dd) and those with an additional anterior dendrite (aad). The third subtype is a single non-canonical α1 DAN, which has a restricted axonal field (*Figure 28—figure supplement 1G–I*). Morphological clustering perfectly matches input clustering, and neither is statistically independent (Mantel test: r = 0.779, p = 3.97 × 10$^{-4}$, n = 5 × 10$^{4}$).

The online version of this article includes the following figure supplement(s) for figure 28:

**Figure supplement 1.** Anatomy of PAM01, PAM12, and PAM11 DAN subtypes.

it seemed reasonable to ignore weak connections in our analyses. Thresholds must obviously be chosen with care and the effects of any particular cutoff value on results and conclusions should be assessed, preferably in conjunction with experimental data.

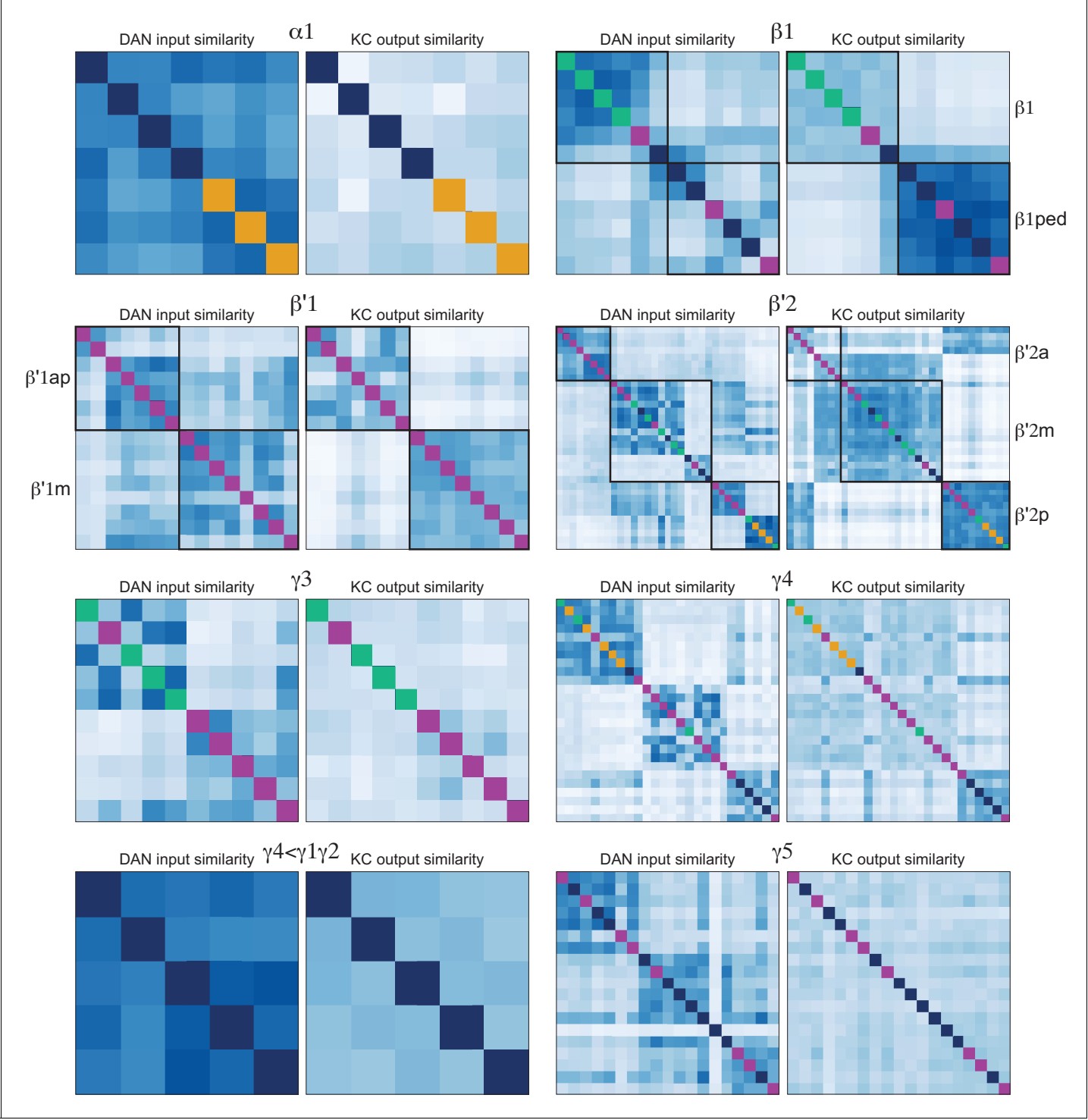

**Figure 29.** Relationship between DAN dendritic inputs and DAN axonal outputs to KCs. These plots explore whether DANs of the same cell type that have distinct inputs also connect to distinct populations of KCs within a compartment. Analysis of eight compartments is shown. Within each example, the left heatmap represents the similarity of input to DANs (identical to the corresponding block in *Figure 27*). The right heatmap represents the cosine similarity of the output synapses made by DANs onto KCs. The clustering observed in the right panels most likely results from individual PAM neurons only arborizing in a portion of their compartment (see neuronal morphologies *Figure 28—figure supplement 1* and *Figure 30*). Such clustering may permit specific reinforcement signals to be conveyed to subsets of KCs within a compartment. See detailed discussion of DAN-KC connectivity in *Figure 30* and *Figure 30—figure supplements 1* and *2*. The potential computational significance of this phenomenon is explored in

*Figure 29 continued on next page*

*Figure 29 continued*

the Discussion (see *Figure 38*). Additional anatomical information about DAN cell types PAM11 (a1), PAM09 (b1-ped), PAM12 (g3) and PAM07 (g4<g1g2) is presented in *Figure 29—video 1*, *Figure 29—video 2*, *Figure 29—video 3*, *Figure 29—video 4*, respectively.

The online version of this article includes the following video(s) for figure 29:

**Figure 29—video 1.** PAM11 (α1) dopaminergic neurons.
https://elifesciences.org/articles/62576#fig29video1
**Figure 29—video 2.** PAM09 (β1-ped) dopaminergic neurons.
https://elifesciences.org/articles/62576#fig29video2
**Figure 29—video 3.** PAM12 (γ3) dopaminergic neurons.
https://elifesciences.org/articles/62576#fig29video3
**Figure 29—video 4.** PAM07 (γ4<γ1γ2) dopaminergic neurons.
https://elifesciences.org/articles/62576#fig29video4

Fly research has been greatly facilitated by the development of numerous fly lines that provide cell-type specific genetic access. Our analysis has revealed, particularly in the case of DANs, subtypes within groups of neurons that would previously have been considered a single type. Thus, the higher resolution view of cell types that connectomics provides points out the need to develop driver lines or other experimental methods for more fine-tuned genetic access.

## Comparison with the larval MB

The cell type constituents and circuit motifs of the MB in the adult fly have many similarities with its precursor at the larval stage of development (*Aso et al., 2014a*; *Eichler et al., 2017*; *Takemura et al., 2017*; *Eschbach et al., 2020a*; *Eschbach et al., 2020b*). Both the larval and adult MBs support associative learning and, in both, PNs from the antennal lobe that convey olfactory information provide the majority of the sensory input, complemented by thermal, gustatory and visual sensory information that is segregated into distinct KC populations. However, the multi-layered organization of non-olfactory inputs in the main and accessory calyces (including integration of diverse input sources by LVINs) suggests that the KC representation in the adult is more highly enriched and specialized for non-olfactory sensory features. It is worth noting that the earliest born types of each of the three main adult KC classes (KCγs, KCγd, KCγt, KCα′β′ap1, KCαβp) appear to be specialized for non-olfactory sensory cues, and in most cases their dendrites lie in the accessory calyces.

In the first instar larval MB, the only larval stage for which a connectome is available (*Eichler et al., 2017*), there are roughly 70 mature KCs, which is increased nearly 30-fold in the adult. This enrichment likely increases odor discrimination and olfactory memory capacity. The larval MB has only eight compartments in its horizontal and vertical lobes. Although the increase in the adult to 15 compartments is only about a factor of two, the extent of their DAN modulation is greatly expanded. The larva has only seven confirmed DANs and five additional cells of unknown neurotransmitter thought to provide modulatory input, a factor of 15-fold less than the adult. Whereas each larval KC innervates all eight compartments, individual adult KCs innervate only five out of 15 compartments. Therefore the DANs in the larva are capable of modulating all KCs, whereas in the adult, DANs in different compartments modulate specific subtypes of KCs. The expansion of the number of DANs within many compartments in the adult MB, and subcompartmental targeting of individual DANs within a compartment, further increases the difference in granularity of DAN modulation between the larva and adult.

MBONs feedback onto DANs and converge onto common downstream targets in both the larva (*Eschbach et al., 2020a*; *Eschbach et al., 2020b*) and adult, implying some shared computational strategies. However, the greatly increased DAN complexity in the adult fly and the presence of subcompartmental organization of DAN axon targets not present in the larva suggest a substantial increase in the specificity of the learning signals involved in memory formation and raises the possibility of modality-specific learning signals to complement the multimodal KC representation.

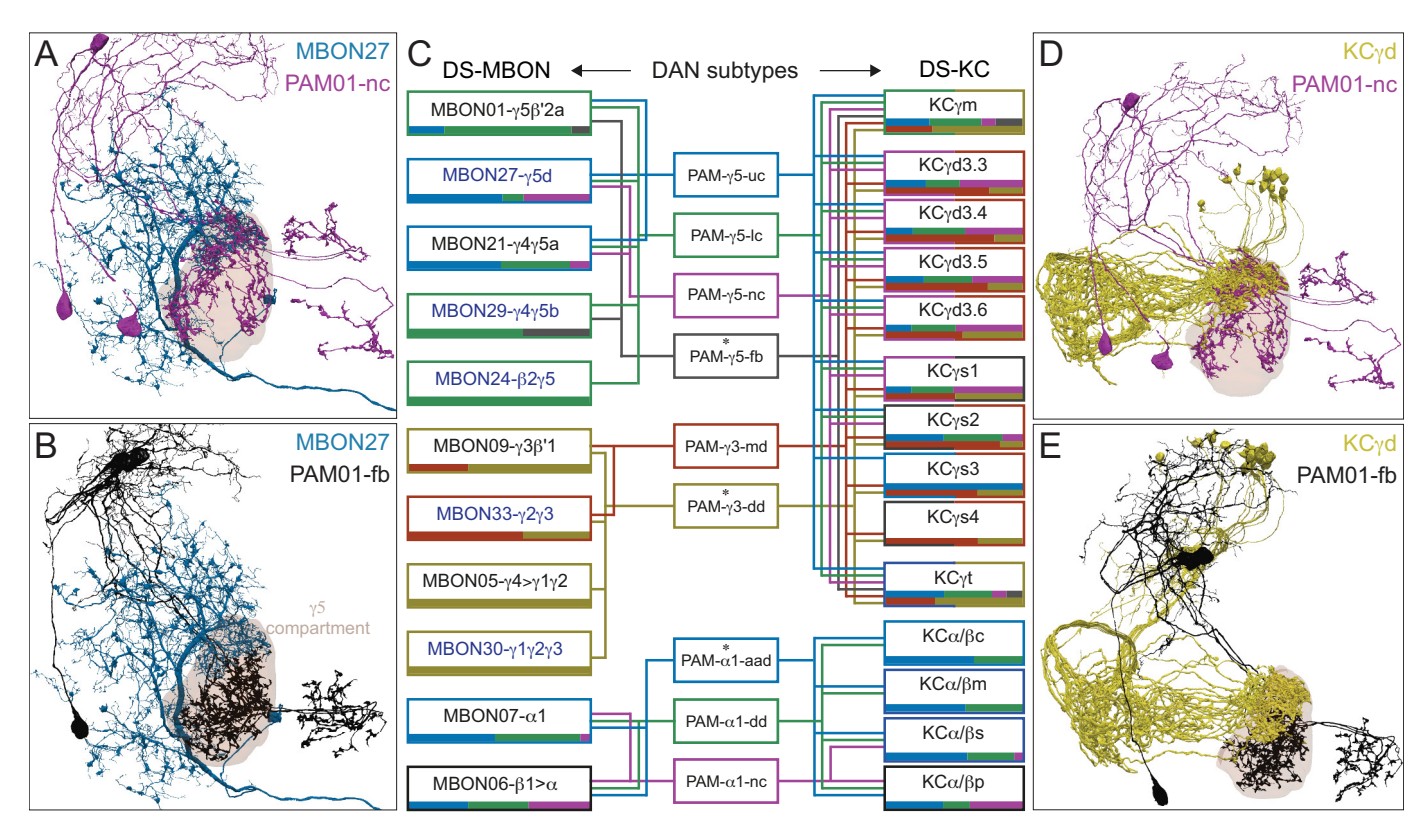

**Figure 30.** Different subtypes of the PAM01, PAM12, and PAM11 DANs synapse onto specific subpopulations of KCs and MBONs within their compartment. KCs are subtyped as shown in *Figure 4—figure supplements 1*, *2* and *4*. The axons from certain subtypes of DAN occupy different regions of, and together tile, a MB compartment. This permits for specific connectivity to KCs and MBONs within the compartment. A zonal subcompartment arrangement is a feature of many MB compartments and is illustrated here by the distinct morphologies of two subtypes of the γ5 compartment DANs (A, B, D, E). Implications for connectivity of this type of restricted DAN arborization are summarized for the γ5, γ3, and α1 compartments in panel C. *Figure 30—figure supplements 2* and *3* show connectivity matrices for additional compartments in the right hemisphere. (A) The axons from PAM01-nc (magenta) overlap with dendrites of MBON27 (γ5d; dark blue). Note that the axons of the two PAM01-nc DANs (magenta) are confined to the dorsal and lateral region of the γ5 compartment (see also *Figure 28—figure supplement 1C*). (B) PAM01-fb (black) axons do not overlap with the dendrites of MBON27 (dark blue). (C) Schematic of the innervation of subtypes of γ5, γ3, and α1 DANs within the compartment showing their connection to MBONs (left) and KCs (right). Boxes for DAN subtypes and connectivity edges are colored according to subtype identity. Boxes around MBON types are colored by their strongest DAN subtype input. The boxes around KC names are colored to reflect their strongest DAN subtype input in each compartment (the two halves for KCγ types reflect their projections through the γ3 and γ5 compartments). Horizontal bars along the lower edge of the MBON and KC boxes are histograms, with bar colors corresponding to DAN subtypes, representing the proportion of each DAN subtype providing input to the respective MBON or KC subtype. For the KCγ types, the upper bar represents inputs in γ5 and the lower bar in γ3. Typical MBON names in black and atypical MBONs are in blue. Asterisks denote DAN subtypes receiving MBON feedback (compare with *Figures 26–28*). Only cases where a DAN provides at least 0.1% of the MBON's synaptic input or at least 1% of a KC type's input from DANs are shown. KC cell types are biased in their representation of different sensory modalities (*Figure 15—figure supplement 2*). Note that the synapses onto the feedforward MBON06 (β1>α) are likely axo-axonal. (D) The axons from PAM01-nc (magenta) also overlap with the axonal processes of KCγd (yellow). (E) Axons from PAM01-fb (black) do not overlap with the axonal processes of KCγd (yellow).

The online version of this article includes the following figure supplement(s) for figure 30:

**Figure supplement 1.** DAN subtypes within a compartment can be biased toward KC types representing different sensory modalities.

**Figure supplement 2.** Within a compartment PAM DAN subtypes synapse onto specific KC subtypes as defined by the clustering shown in *Figure 4— figure supplements 1*, *2* and *4*.

**Figure supplement 3.** PAM DAN subtypes synapse selectively onto particular MBONs within a compartment.

## Sensory input to KCs

Previous theoretical work has emphasized the advantages of mixing sensory input to the KCs so that they provide a high-dimensional representation from which MBONs, guided by DAN modulation, can derive associations between sensory input and stimulus-valence (*Litwin-Kumar et al., 2017*).

The connectome data modifies this viewpoint in two ways. First, although olfactory input to the KCs is highly mixed, various structural features reduce the dimensionality of the KC odor representation. A recent analysis of EM data from an adult *Drosophila* MB identified groups of PNs thought to represent food odors that are preferentially sampled by certain KCs (*Zheng et al., 2020*). Consistent with this observation, our analysis revealed subtype-specific biases in PN sampling by KCs, including an overrepresentation of specific glomeruli by α/β and α′/β′ KCs (*Figure 13B*). These biases, as well as other structural features, appear to arise from the stereotyped arrangement of PN axons within the calyx and their local sampling by KCs. This may reflect a developmental strategy by which the KC representation is organized to preferentially represent particular PN combinations. Further analyses of KC connectomes across hemispheres and animals, as well as experimental studies, will help evaluate the impact of this structure.

The hemibrain connectome also revealed a second, more dramatic, structural feature of sensory input to the MB: non-olfactory input streams corresponding to visual and thermo-hygro sensation are strongly segregated. This organization may reflect the nature of stimulus-valence associations experienced by flies. In purely olfactory learning, when valence is associated with particular sets of olfactory receptors, MBONs need to be able to sample those combinations to successfully identify the stimulus. This requires that KCs mix input from multiple glomeruli within the olfactory stream. However, KCs that mix across streams corresponding to different sensory modalities may not be necessary if each modality can be used separately to identify valence. For example, either the visual appearance of an object or its odor may individually be sufficient to identify it as a food source (for a discussion of multi-sensory learning in flies, see *Guo and Guo, 2005*). We asked whether there is an advantage in having separate modality-specific sensory pathways, as seen in the connectome data, when valences can be decomposed in this way.

To address this question we considered a model in which KC input is divided into two groups, visual and olfactory (*Figure 38A*). In one version of the model ('shared KC population'), all KCs receive sparse random input from both types of PNs, corresponding to a high degree of mixing across modalities. In the other ('separate KC populations'), half of the KCs receive sparse random input exclusively from visual PNs and half only from olfactory PNs, corresponding to no cross-modal mixing. We define two tasks that differ in the way valences are defined. In the first (factorizable) version of the task, each olfactory stimulus and each visual stimulus is assigned a positive or neutral sub-valence, and the net valence of the combined stimulus is positive if either of these components is positive (olfactory OR visual). In the second (unfactorizable) version, a valence is randomly assigned independently to each olfactory+visual stimulus pair. We evaluated the ability of a model MBON, acting as a linear readout, to determine valence from these two different KC configurations. Separate modality-specific KC populations are indeed beneficial when the valence can be identified from either one modality or the other (the 'factorizable' case in *Figure 38B*). Dedicating different KC subtypes to distinct sensory modalities allows the predictive value of each modality to be learned separately. This result suggests that the divisions into KCs specialized for visual, thermo/hygro and olfactory signals may reflect how natural stimuli of different modalities are predictive of valence.

## Sub-compartmental modulation by DANs

On the basis of light-level studies, DAN modulation of KC-to-MBON synapses has been considered to operate at the resolution of MB compartments. However, taken with other recent studies (*Lee et al., 2020*; *Otto et al., 2020*), the morphology and connectivity data indicate that functionally distinct PAM-DAN subtypes operate within a MB compartment. DAN subtypes receive different inputs and likely modulate different KC-MBON synapses within a compartment.

A prior analysis showed that PAM01-fb DANs were required to reinforce the absence of expected shock during aversive memory extinction, whereas a different set of γ5 DANs were needed for learning with sugar reinforcement (*Otto et al., 2020*). It is conceivable that another γ5 DAN subtype is required for male flies to learn courtship rejection (*Keleman et al., 2012*). The connectome data revealed DAN subtypes in every compartment that is innervated by PAM DANs.

The γ3 compartment provides an interesting example of subcompartmental targeting of modulation by DANs and of KC input onto MBONs (*Figure 39*). There are two subtypes of PAM DANs and three types of MBONs in the γ3 compartment. MBON09 and MBON30 primarily receive olfactory information from γm KCs whereas MBON33 primarily receives visual information from γd KCs. PAM12-dd and PAM12-md DANs appear to modulate KC inputs to MBON09/MBON30 or

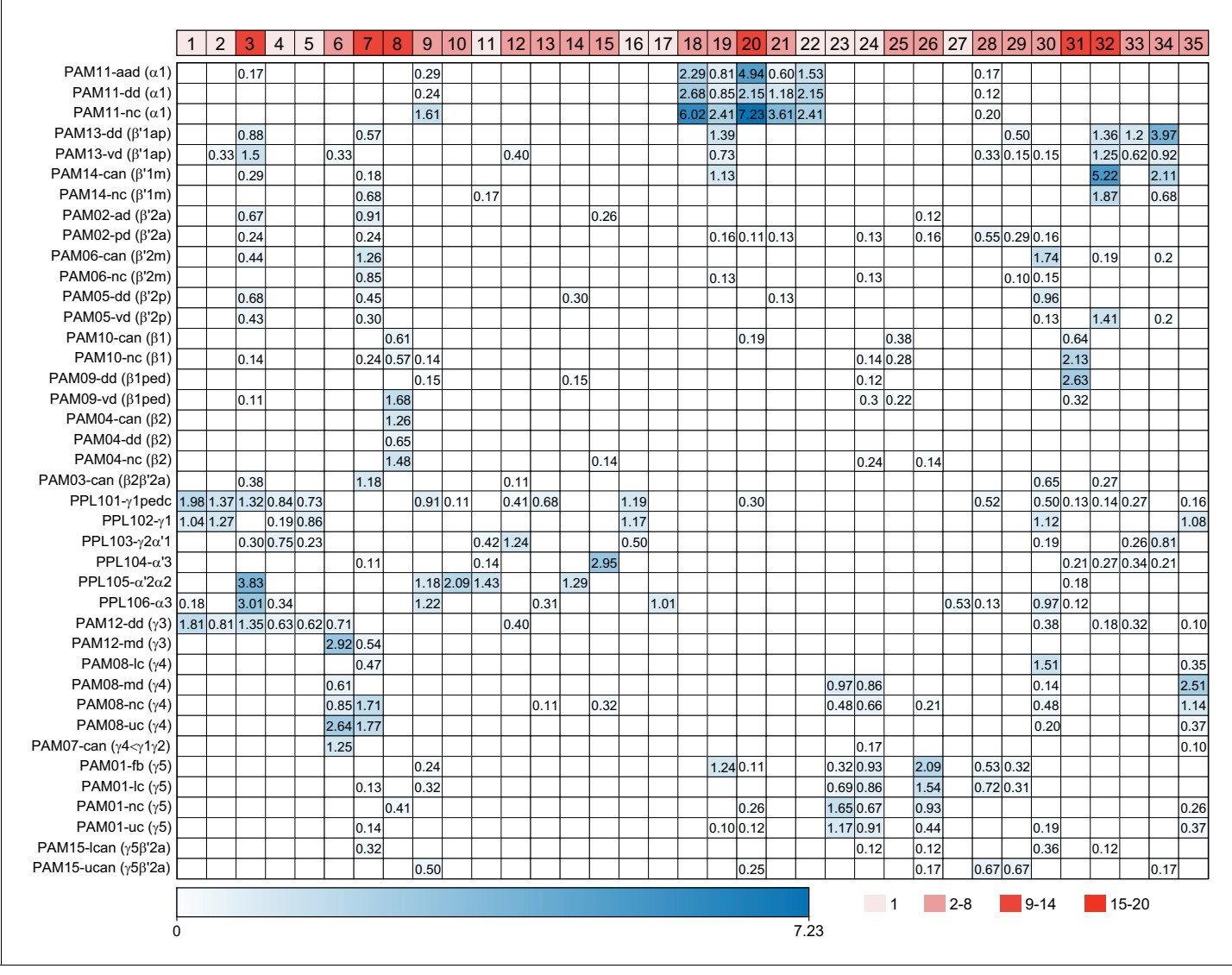

**Figure 31.** Shared input neurons reveal expected and unexpected across-compartment groupings of DAN subtypes. Matrix representing the connectivity to DAN subtypes provided by the top 50 strongest input neurons to each DAN subtype in the right hemisphere (n = 901 neurons, excluding MBONs and MB intrinsic neurons). These inputs are either unique individual neurons (clusters of one) or clusters of multiple neurons grouped according to their morphology. A selection of 35 input clusters are shown, numbered in boxes whose shading reflects the number of neurons that contribute to that cluster (input clusters are listed in *Supplementary file 2*). Each cell represents the proportion of the total inputs to the dendrites of the DANs of the respective subtype (y-axis) provided by the input neuron cluster (x-axis). Selected input clusters and innervation patterns are further explored in *Figures 32–37*. Some DAN subtypes receive inputs from neurons projecting from heavily studied neuropils like the SEZ, the FB, and the LH; for example, clusters 10, 18, 19, 20, 21, 22, 23, 24, 25, 26, 27, 28, 31 (*Figures 33–35*). Many clusters selectively connect to groups of DAN subtypes of the PAM or PPL1 type specifically; for example, clusters 7, 8, 11, 12, 13, 14, 15, 29, 32 (*Figure 36*) and clusters 22, 23, 24, 26, 27 (*Figure 35*). Some input neuron clusters connect DAN subtypes with opposite/mixed valence; for example, clusters 3, 9, 30, 33, 35 (*Figure 37*) and cluster 28 (*Figure 35*; also see *Otto et al., 2020*). Input neuron clusters can also stand out by the DAN subtypes that they avoid; for example, clusters 3 and 30 (*Figure 37*). The matrix was thresholded to show inputs over 0.1% of total input to DAN dendrites and the threshold for strong connectivity was defined as 0.5%. The neuropil of the SEZ is not included in the hemibrain database. Therefore, more complete anatomy of DAN input neurons from this region were retrieved from FAFB (*Zheng et al., 2018* ; see *Figure 35*).

MBON33, respectively. Although existing driver lines do not separate the dd and md subtypes, PAM-γ3 DAN activity is suppressed by sugar (*Yamagata et al., 2016*) and activated by electric shock (*Jacob and Waddell, 2020*). We found that PAM12-dd DANs share input with PPL1 DANs conveying punishment signals, while PAM12-md DANs are co-wired with PAM08 (γ4) DANs conveying reward

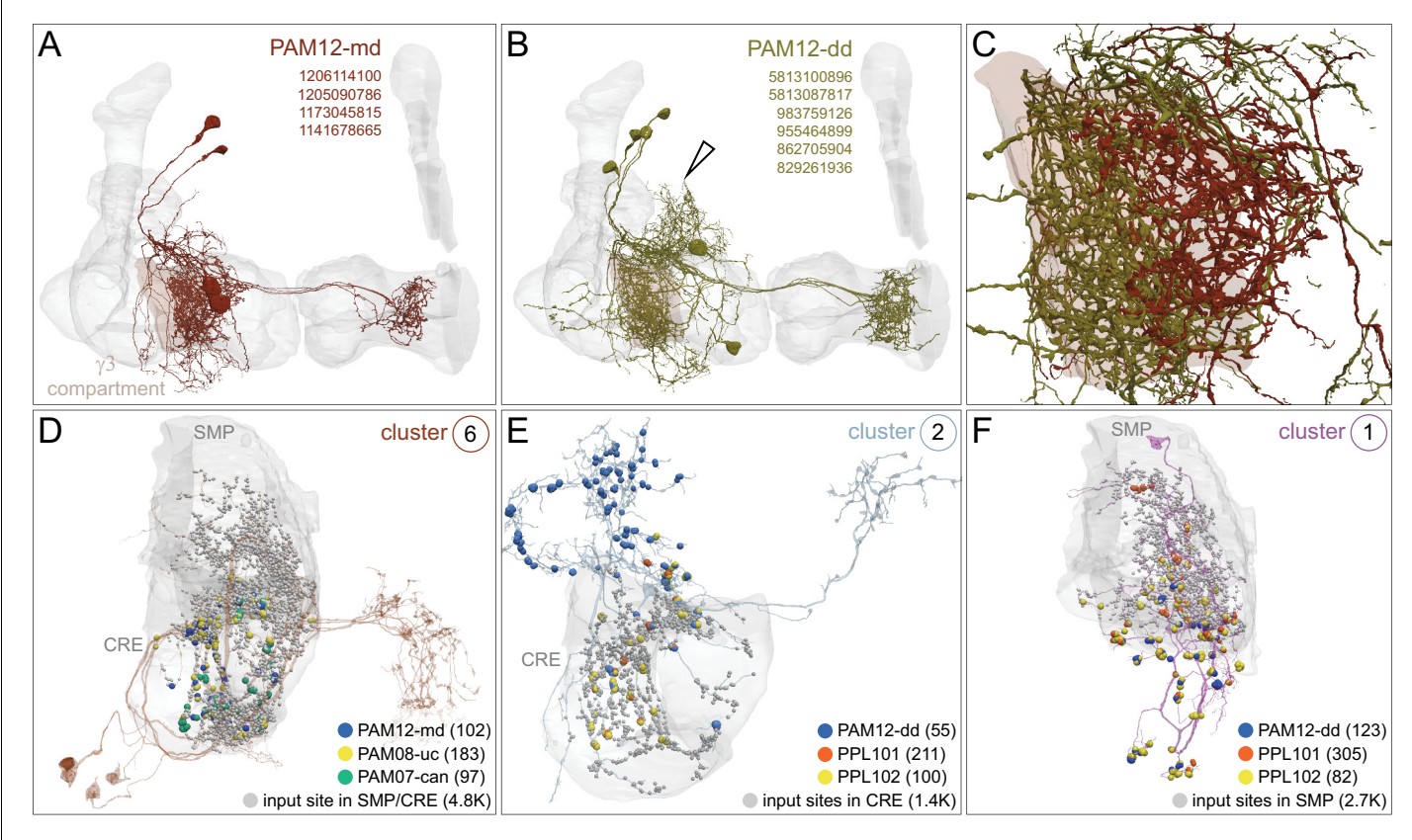

**Figure 32.** Morphologically distinct subtypes of PAM12-DANs share inputs with either PPL1 or PAM DANs. (**A**) The four PAM12-md DANs (maroon) innervate the γ3 compartment (brown) of the MB (gray ). (**B**) The PAM12-dd subtype (green) has additional dorsal dendritic branches (hollow arrowhead) not found in PAM12-md. A seventh PAM12-dd that lacked its ipsilateral axon was excluded from the analysis. (**C**) The axonal fields of PAM12-md and PAM12-dd DAN subtypes tile the γ3 compartment (brown), suggesting they modulate different downstream neuronal connections (see ***Figure 30*** and ***Figure 39*** for further analyses and discussion). (**D–F**) Strong inputs (see ***Figure 31***) to PAM12-md also specifically synapse onto positive-valence PAM DANs, whereas PAM12-dd shares inputs with negative-valence PPL1 DANs; synapse numbers are given in parentheses. Circled number in the upper right of each panel refers to the input cluster number in ***Figure 31***. (**D**) Cluster 6 collectively provides the strongest dendritic input to PAM12-md DANs (blue dots) and is also the strongest input to two DAN subtypes innervating γ4, PAM08-uc DANs (yellow dots) and canonical (can) PAM07-can (green dots). In addition, cluster 6 connects to other PAM08-DANs, but excludes the lower commissure (lc) subtype. (**E**) A single, morphologically distinct neuron (cluster 2) connects to the positive valence PAM12-dd (blue dots) and the negative-valence PPL101 (γ1pedc) (orange dots) and PPL102 (γ1; yellow dots). (**F**) This single (cluster 1) neuron provides the strongest dendritic input to positive-valence PAM12-dd (blue dots) and the negative-valence PPL101 (γ1pedc; orange dots) and also synapses onto PPL102 (γ1; yellow dots) DANs. Connections contributing less than 1% of a neuron's total dendritic input have been excluded. See ***Figure 32—figure supplement 1*** for further morphological details.

The online version of this article includes the following figure supplement(s) for figure 32:

**Figure supplement 1.** Strong inputs to PAM12-dd and PPL101 localize to different dendritic arbors.

signals. Thus, synaptic transmission from two sets of modality-specific KCs to different MBONs can be independently modulated by DANs signaling different valences, all within a single compartment.

To explore the implications of DAN modulation that is specific to sensory modality, we extended the model presented above (***Figure 38A***) by including two DANs, one conveying the visual component of valence and the other the olfactory component (***Figure 38C***). KCs were divided into visual and olfactory modalities, and we considered two configurations for DAN modulation, one, 'shared reward signal', that is compartment-wide and non-specific, and the other, 'separate reward signals', in which each DAN only induces plasticity onto KC synapses matching its own modality. This latter case models a set of DANs that affect synapses from visual KCs onto the MBON and another set that affect olfactory synapses (alternatively, it could model two MBONs in different compartments that are modulated independently and converge onto a common target). We find that, when KCs

are divided into separate populations and separate modalities can be used to identify valence, learning is more efficient if the pathways are modulated individually (*Figure 38D*).

## Functional implications of DAN input heterogeneity and MBON feedback to DANs

Our analysis revealed that DANs receive very heterogeneous inputs but, nonetheless, some DANs both within and across compartments often share common input. This combination of heterogeneity and commonality provides many ways of functionally combining different DAN subtypes. For example, we expect that it allows DANs to encode many different combinations of stimuli, actions and events in a state-dependent manner and to transmit this information to specific loci within the MB network.

In addition to heterogeneous inputs from a variety of brain regions, the DAN network receives a complex arrangement of within and across-compartment monosynaptic input from a variety of MBONs, using both excitatory and inhibitory neurotransmitters. We found that nearly all MB compartments contain at least one direct within-compartment MBON-DAN feedback connection. MBON feedback onto these DAN subtypes allows previously learned associations that modify MBON activity to affect future learning. MBONs that feedback onto the same DANs that modulate them could, if the result of learning is reduced DAN activity, prevent excess plasticity for an already learned association. In cases where MBON activity excites the DAN, the self-feedback motif could assure that learning does not stop until the MBON has been completely silenced. More generally, MBON inputs to DANs imply that dopaminergic signals themselves reflect learned knowledge and the actions it generates. This could, in turn, allow MBON modulation of DAN activity to support a number of learning paradigms beyond pure classical conditioning, including extinction, second-order conditioning, operant conditioning and reinforcement learning.

Flies can perform second-order conditioning, in which a stimulus that comes to be associated with reinforcement may itself act as a pseudo-reinforcement when associated with other stimuli (*Tabone and de Belle, 2011*). This computational motif of learning the value of sensory states and using the inferred value as a surrogate reinforcement to guide behavioral learning is the core principle behind a class of machine learning techniques known as actor-critic algorithms. These algorithms consist of two modules: the 'actor', whose job is to map sensory inputs to behavioral outputs and the 'critic', whose job is to map sensory inputs to their inferred values and provide these values as a learning signal for the actor. In the mammalian basal ganglia, the dorsal and ventral striatum, the latter of which strongly influences the activity of dopamine neurons in areas including VTA, have been proposed to represent actor and critic modules, respectively. In the MB , this perspective suggests a possible additional 'critic' function for some MBONs beyond their known 'actor' role in directly driving behaviors. Consistent with this view, activation of individual MBONs can excite DANs in other compartments (*Cohn et al., 2015*; *Felsenberg et al., 2017*). We found strong direct monosynaptic connections between some MBONs and DANs in other compartments. Functional studies will be needed to determine which MBONs, if any, participate in an actor-critic arrangement, and which circuit mechanisms—for example, release from inhibition, or reduction of excitation—are at work.

Another potential role of cross-compartment MBON-DAN feedback is to gate the learning of certain associations so that the learning is contingent on other associations having already been formed. Such a mechanism could support forms of memory consolidation in which long-term memories are only stored after repeated exposure to a stimulus and an associated reward or punishment. Prior studies have linked plasticity in the γ1 and γ2 compartments to short-term aversive memory (*Aso et al., 2012*; *Hige et al., 2015*; *Perisse et al., 2016*) and plasticity in the α2 and α3 compartments to long-term aversive memory (*Aso and Rubin, 2016*; *Awata et al., 2019*; *Jacob and Waddell, 2020*; *Pai et al., 2013*; *Séjourné et al., 2011*). The cross-compartmental MBON-to-DAN connections we observed suggest an underlying cicuit mechanism for this 'transfer' of short to long term memory. Aversion drives PPL101 (γ1pedc) and depresses the conditioned odor-drive to the GABAergic MBON11 (γ1pedc>α/β). MBON11 is strongly connected with PPL1-γ1pedc and is more weakly connected with PPL105 and PPL106. Depression of MBON11 will therefore also release the PPL105 and PPL106 DANs from MBON11-mediated inhibition, increasing their activity in response to the conditioned odor and making them more responsive during subsequent trials. The net result is that short-term aversive learning by MBON11 (γ1pedc>α/β) promotes long-term learning in MBON18 (α2sc) and MBON14 (α3) by releasing the inhibition on the dopamine neurons that

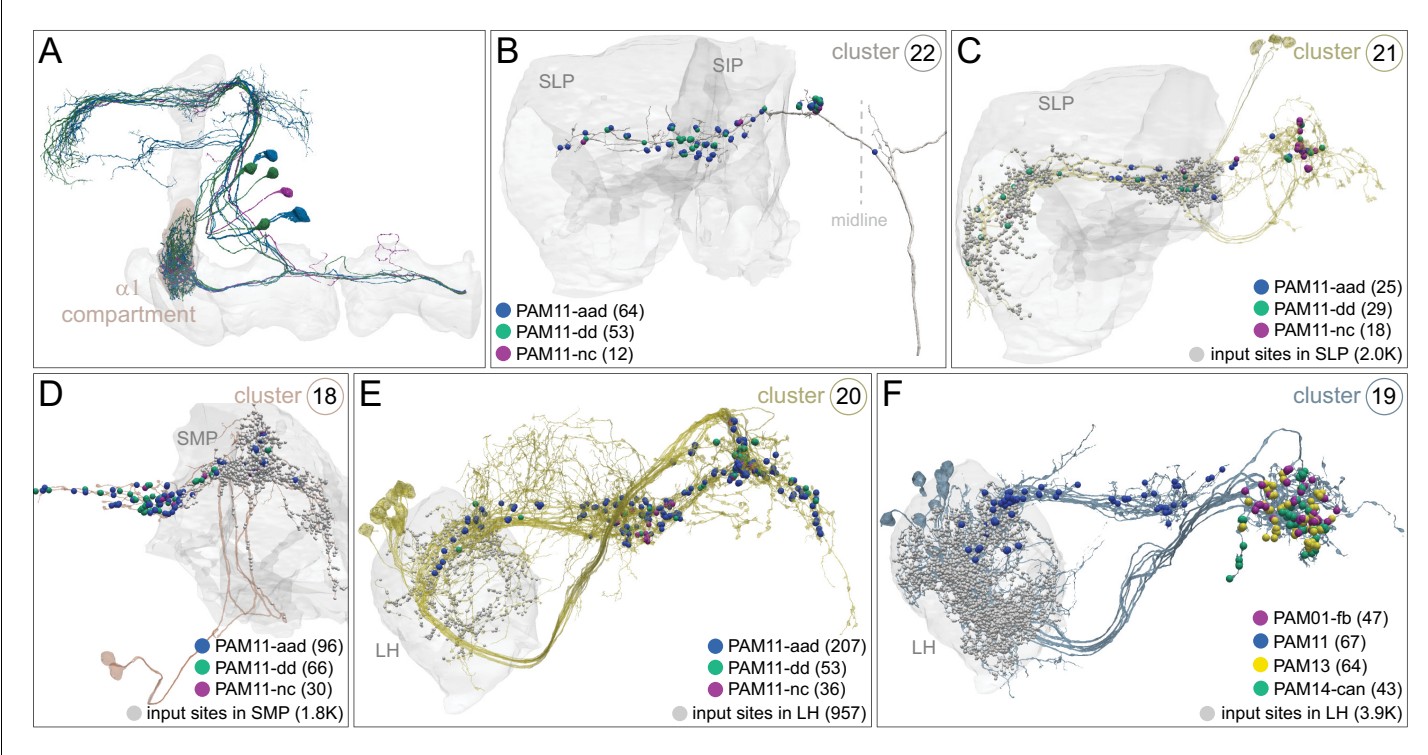

**Figure 33.** Examples of local neurons, SEZONs, and LHONs that target PAM11 DANs. (**A**) All PAM11 DAN subtypes are shown, color-coded as in *Figure 28* and *Figure 28—figure supplement 1*; the MB lobes are shown in gray and the α1 compartment is shaded brown. (**B–F**) Examples of neurons from clusters in *Figure 30* that provide strong input to PAM11 DANs or to PAM11 and other positive-valence PAM DANs are shown; cluster identity is indicated by the circled number in the upper right of each panel. Neuropils where DAN input clusters receive their inputs are shown in gray (except for SEZONs whose dendrites are not in the hemibrain volume) and position of output synapses to PAM11 subtypes are shown color-coded; synapse numbers are given in parentheses. *Figure 33—figure supplement 1* shows more information about the neurons constituting these clusters. (**B**) The single neuron in cluster 22, a SEZON, makes synapses to all three subtypes of PAM11 DANs; PAM11 DANs are the only DAN targets of cluster 22. The same applies to its contralateral partner (see *Figure 35—figure supplement 1A*). (**C**) The local interneurons that make up cluster 21 receive inputs in the SLP and connect to all three PAM11 subtypes, but not to other DANs. (**D**) The local interneurons that make up cluster 18 (299626157, 361700223) receive inputs in the SMP and connect to PAM11 DANs, but not to other DANs. (**E**) LHON cluster 20 connects to all three PAM11 subtypes and weakly to PAM01 and other positive-valence PAM DANs. (**F**) The LHONs cluster 19 connect to all three PAM11 subtypes and also to PAM13 (β′1ap), PAM14 (β′1m) and PAM01-fb (γ5) DANs (compare to *Frechter et al., 2019*, *Dolan et al., 2019* , and *Otto et al., 2020*). The connections to the PAM11 and the other PAM DANs are located on different primary branches of the cluster 19 neurons. Connections contributing less than 0.5% of a neuron's total dendritic input have been excluded.

The online version of this article includes the following figure supplement(s) for figure 33:

**Figure supplement 1.** More information on the cell types shown in *Figure 33*.

innervate the α2 and α3 compartments (*Séjourné et al., 2011*; *Jacob and Waddell, 2020*). Indeed pairing inactivation of MBON11 with odor presentation can form an aversive memory that requires output from PPL1-DANs (*Ueoka et al., 2017*) and optogenetic stimulation of MBON11 during later trials of odor-shock conditioning impairs long-term memory formation (*Awata et al., 2019*).

Cross-compartment MBON-DAN feedback may also enable context-dependent valence associations, such as the temporary association of positive valence with neutral stimuli when a fly is repeatedly exposed to aversive conditions. Multiple, spaced aversive conditioning trials were recently shown to form, in addition to an aversive memory for the shock-paired odor, a slowly emerging attraction, a 'safety' memory, for a second odor that was presented over the same training period without shock (*Jacob and Waddell, 2020*). The GABAergic MBON09 (γ3β′1) appears to play a critical role in the formation of this safety memory. The synapses from KCs conveying the shock-paired odor are depressed in that portion of MBON09's dendrite that arborizes in γ3, while the synapses from KCs conveying the safety odor are depressed in its β′1 arbor. These combined modulations should gradually release downstream neurons from MBON09 feedforward inhibition, consistent with

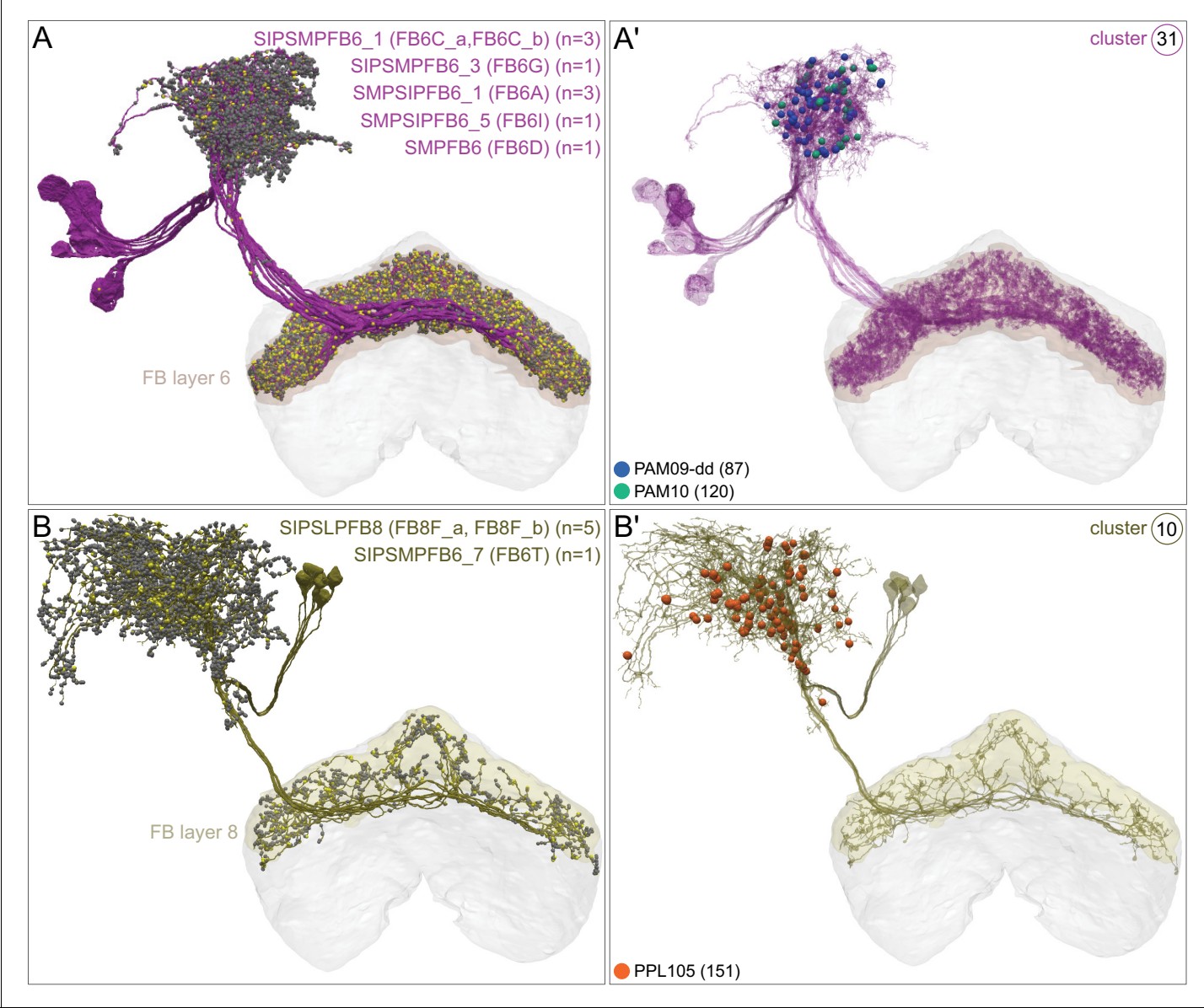

**Figure 34.** Distinct FB neurons provide input to β lobe PAM or PPL105 DANs. Two morphological clusters (31 and 10, *Figure 31*) of neurons from FB are among the strongest DAN inputs. These FB neurons have arbors of mixed polarity both in the SMP and in the FB, as can be seen in panels A and B where their presynaptic and postsynaptic sites are shown in yellow and gray , respectively. (A-A') Cluster 31 neurons are shown in magenta and consists of nine FB layer 6 neurons of five cell types. These neurons synapse onto the dendrites of three potentially positive-valence PAM DAN subtypes: PAM09-dd (β1ped; blue dots), and two β1 DAN subtypes, PAM10-can and PAM10-nc(dd) (combined in figure; green dots); synapse numbers are given in parentheses. Remarkably, seven of the nine neurons of cluster 31 are connected, via an interneuron, to MBON19 (α2p3p; see *Figure 20—figure supplement 1C* and *Figure 20—video 4*). (B-B') Cluster 10 neurons are shown in yellow and consist of five layer 8 (cell type: SIPSLPFB8) and one layer 6 FB neurons and make synapses in the SMP onto PPL105 (α′2α2; orange dots; synapse number is given in parentheses); the negative-valence PPL105 is the only DAN they innervate. Connections contributing less than 0.5% of a neuron's total dendritic input have been excluded.

the proposed mechanism for PAM13/14 (β′1) and PAM05/06 (β′2m and β′2p) DANs becoming more responsive to the safe odor (*Jacob and Waddell, 2020*). The connectome data revealed that MBON09 is directly connected to the PAM13/14 (β′1ap) and PAM05 (β′2p) DANs, consistent with release from inhibition underlying the delayed encoding of safety.

DANs also make direct connections with MBONs in each MB compartment, and optogenetic activation of PAM11 DANs can directly excite MBON07 with slow dynamics (*Takemura et al., 2017*). This might provide a mechanism to temporarily suppress memory expression without impairing the

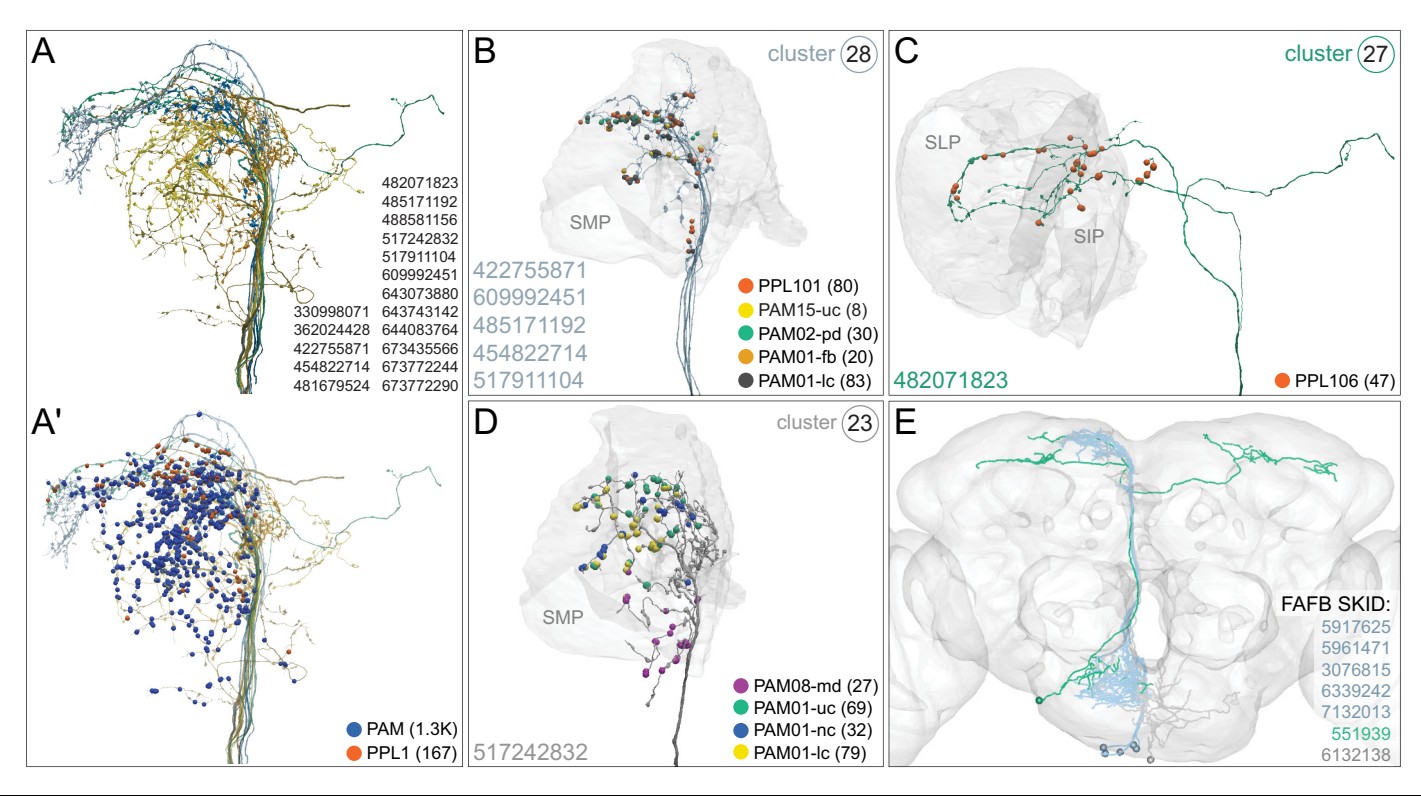

**Figure 35.** Output neurons from different areas of the SEZ tile the SMP where they synapse onto select DAN subtypes. (A) Axons from multiple SEZON cell types in clusters 22, 23, 24, 25, 26, 27, and 28 (see *Figure 31*) that provide strong inputs to DAN subtypes are shown together. (A') The same SEZON axons shown with synapses connecting to PAM DANs (blue, from five clusters) and to PPL1 DANs (orange, from two clusters); synapse numbers are given in parentheses. (B) The five neurons of SEZON cluster 28 (light blue; these neurons correspond to the SEZON01 cluster 5.4.1 of *Otto et al., 2020*) branch in the SMP and synapse onto the PPL101 (orange), PAM01-lc (black), PAM01-fb (dark yellow), PAM02-pd (green), and PAM15-uc (yellow) subtypes; synapse numbers are given in parentheses. (C) The single neuron in SEZON cluster 27 (green) branches in the SMP and SIP and synapses only onto PPL106; synapse number is given in parentheses. (D) The single neuron in SEZON cluster 23 (gray) branches in the SMP and connects to positive-valence DANs: PAM01-lc (yellow), PAM01-uc (green), PAM01-nc (blue), and PAM08-md (magenta); synapse numbers are given in parentheses. (E) The axons of SEZON neurons can be matched to those in the FAFB EM volume (*Zheng et al., 2018*) to permit the retrieval of their dendritic field, which resides in tissue that is missing from the hemibrain dataset; skeleton ID numbers from FAFB are shown. FAFB matches to hemibrain SEZONs are shown and colored to match their hemibrain counterparts (colors correspond to those used in panels B-D); the SEZON 551939 was traced in the left hemisphere in FAFB and a mirror image is shown here to facilitate comparison. Their dendritic fields differ, suggesting these SEZON clusters are likely to convey distinct information. Connections contributing less than 0.5% of a neuron's total dendritic input have been excluded.

The online version of this article includes the following figure supplement(s) for figure 35:

**Figure supplement 1.** Most strongly connected SEZON clusters exclusively synapse onto positive-valence PAM DANs.

underlying memory, which is stored as depressed KC-to-MBON synapses (*Krashes et al., 2009*; *Schleyer et al., 2020*; *Senapati et al., 2019*; *Takemura et al., 2017*). For example, DANs are known to control hunger and thirst-dependent memory expression (*Krashes et al., 2009*; *Senapati et al., 2019*), and their excitation of MBONs could provide a possible mechanism.

## MBON-MBON interactions

Feedforward MBON-MBON connections were postulated, based on behavioral and light micro-scopic anatomical observations, to propagate local plasticity between compartments (*Aso et al., 2014a*; *Aso et al., 2014b*). The GABAergic MBON11 is poised to play a key role in such propaga-tion as it is connected with 17 other MBONs (at a threshold of 10 synapses). Thus local depression of KC-MBON11 synapses by the shock responsive PPL1-γ1pedc DAN would be expected to result in disinhibition of MBONs in other compartments. Indeed enhanced CS+ responses of MBON01 and MBON03 after odor-shock conditioning have been ascribed to release from MBON11 inhibition (*Owald et al., 2015*; *Perisse et al., 2016*; *Felsenberg et al., 2018*). The consequence of releasing

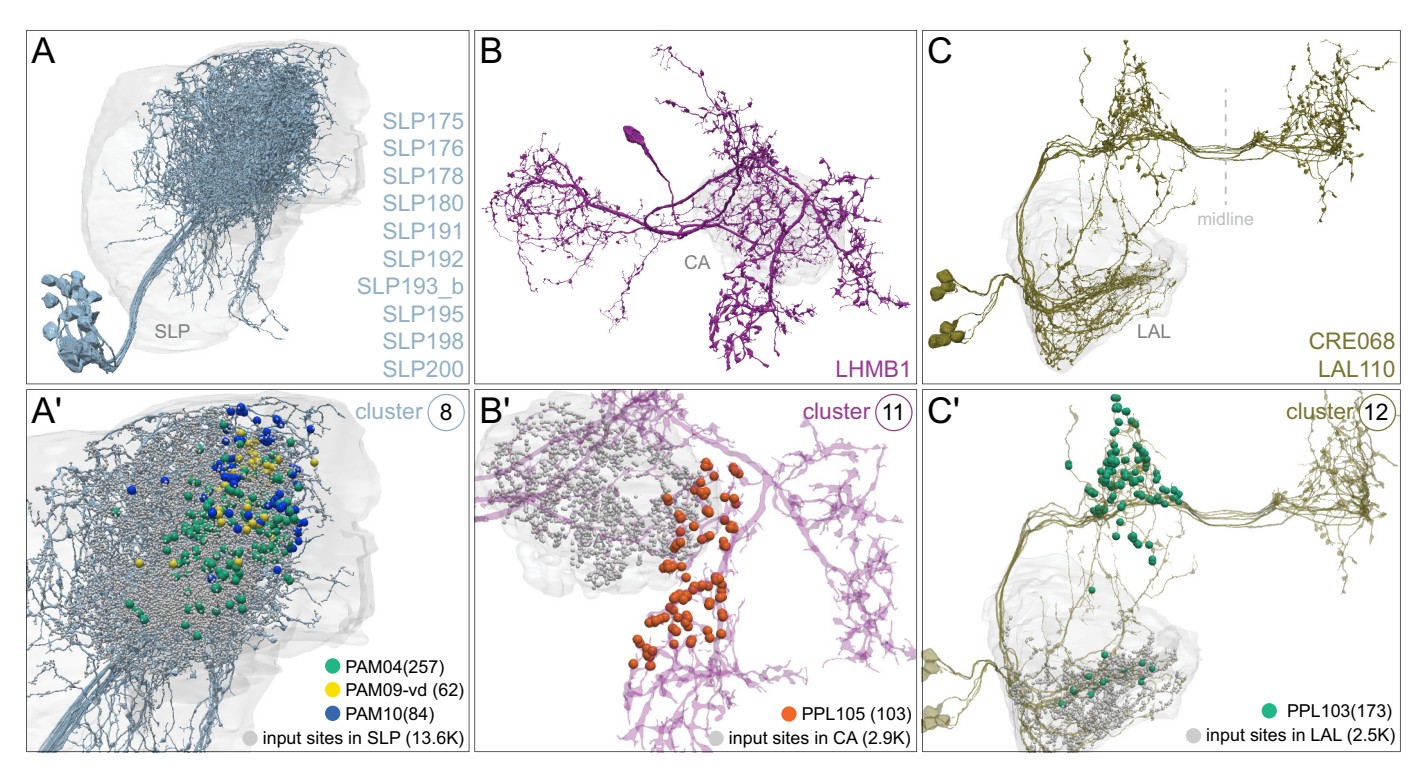

**Figure 36.** A number of input neurons specifically target DAN groups of the PAM or PPL1 clusters. Neurons providing strong input to PAM or PPL1 DANs are shown. Neuropil areas of inputs to these neurons are shown in gray in (**A–C**) and synapses to DANs are shown color-coded in (**A'-C'**); synapse numbers are given in parentheses. The threshold for connectivity is 0.5% of the DANs total inputs (*Figure 31*). (**A-A'**) Cluster 8 is comprised of 20 neurons of 10 similar cell types which provide input to subtypes of β-lobe PAM DANs in the SLP: PAM10 (β1; blue); PAM09-vd (β1ped; yellow) and all PAM04 subtypes (β2; green). (**B-B'**) The single neuron in cluster 11 receives inputs in the CA and connects to PPL105 (α'2α2; orange). (**C-C'**) The seven neurons in cluster 12 (LAL110, CRE068) receive inputs in the LAL and provide the strongest input to PPL103 (γ2α'1; green). Connections contributing less than 0.5% of a neuron's total dendritic input have been excluded.

The online version of this article includes the following figure supplement(s) for figure 36:

**Figure supplement 1.** Inputs specific to PAM DANs.
**Figure supplement 2.** Inputs specific to PPL1 DANs.

other strongly connected MBONs (MBON07, MBON14 and MBON29) from MBON11 inhibition awaits future study. As discussed above, MBON11 is also connected to DANs innervating its cognate compartment and to DANs innervating other compartments. MBON11 may therefore coordinate MBON network activity via both direct and indirect mechanisms. We also found analogous feedforward inhibitory connections from MBON09 (γ3β'1) to MBON01 and MBON03. Aversive learning therefore reduces both MBON11- and MBON09-mediated inhibition of these MBONs, which further skews the MBON network toward directing avoidance of the previously punished odor (*Figure 40*).

Disinhibition likely also plays an important role in appetitive memory (*Figure 41*). The connectivity of MBON06 (β1>α) revealed here indicates that local plasticity in the β1 compartment can propagate to other MBONs. Similar to MBON11, MBON06 is directly connected with nine other MBONs with a threshold of 10 synapses. MBON06 gradually increases its response to repeated odor exposure (*Hattori et al., 2017*) and odor-evoked responses of β1 DANs vary with metabolic state (*Siju et al., 2020*). More compellingly, artificially triggering PAM10 (β1) DANs can assign appetitive valence to odors (*Perisse et al., 2013*; *Huetteroth et al., 2015*; *Aso and Rubin, 2016*). However, the role of MBON06 in appetitive memory and how the PAM10 (β1) DANs modulate KC synapses to MBON06 has not been investigated. The glutamatergic MBON06 (β1>α) makes a large number of reciprocal axoaxonic connections with the GABAergic MBON11 (*Takemura et al., 2017*), whose activation favors approach (*Aso et al., 2014b*; *Perisse et al., 2016*). This reciprocal network motif

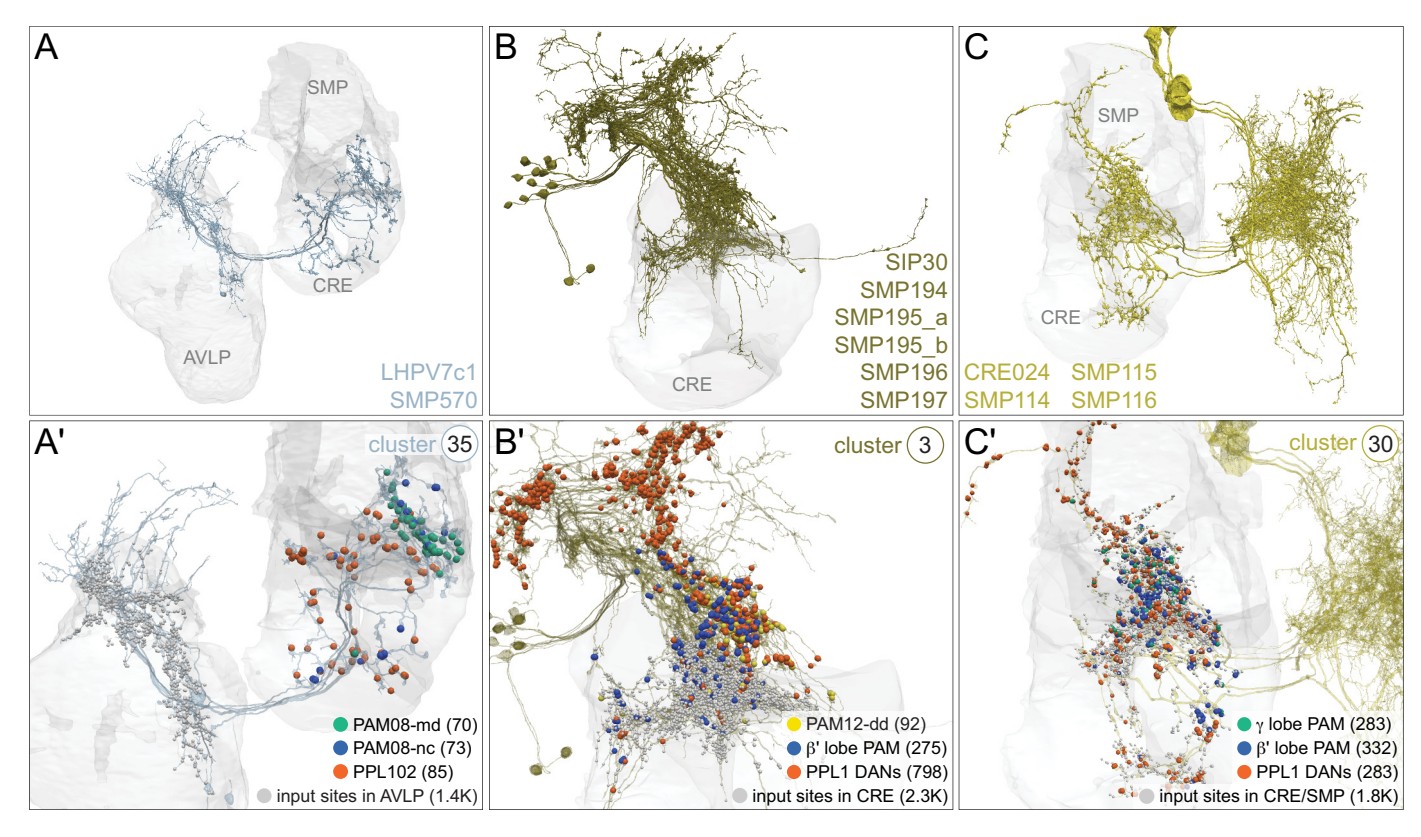

**Figure 37.** Neurons that provide strong input to both PAM and PPL1 DANs. Neuropil areas indicated in gray and synapses to DANs are color-coded ; synapse numbers are given in parentheses. (**A-A'**) The four neurons of cluster 35 receive inputs in the AVLP and provide strong input to the PAM08-md (γ4; green) and PAM08-nc (blue synapses) subtypes as well as the PPL102 (γ1) DAN (orange synapses). (**B-B'**) The 13 neurons of cluster 3 receive some inputs in the CRE; they constitute the strongest input to PPL105 (α'2α2) and PPL106 (α3) and also connect to PPL101 (γ1pedc), PPL102 (γ1), and PPL103 (γ2α'1) (all PPL1 DAN synapses combined, red), all of the β' lobe PAM DAN subtypes (blue synapses) and the PAM12-dd (γ3) subtype (yellow synapses). (**C-C'**) Cluster 30 has four neurons that receive inputs in the SMP and CRE and are upstream of negative-valence PPL1 neurons (PPL101, PPL102, PPL105, and PPL106; orange synapses) and PAM DANs innervating the β' (blue synapses) and γ (green synapses) lobe (PAM08, PAM01, PAM03, PAM05, PAM06). While both cluster 3 and 30 connect to both PPL1 and PAM DANs, cluster 30 connects to γ lobe PAM DANs, in contrast to cluster 3 favoring β' lobe PAM DANs.

The online version of this article includes the following figure supplement(s) for figure 37:

**Figure supplement 1.** Additional examples of neurons providing common input to both PAM and PPL1 DANs.

and the positive sign of behavior resulting from β1 DAN-driven memory suggests that MBON06 released glutamate is likely to be inhibitory to MBON11 and its other downstream targets. MBON06 also makes twice as many connections onto the glutamatergic MBON07 and the cholinergic MBON14 as does MBON11. MBON14 (α3) and MBON07 (α1) have established roles in appetitive memory (*Plaçais et al., 2013*; *Huetteroth et al., 2015*; *Yamagata et al., 2015*; *Ichinose et al., 2015*; *Widmer et al., 2018*). Assuming that PAM10 (β1) DANs encode appetitive memory by depressing synapses from odor-specific KCs onto MBON06 (β1>α), MBON06 suppression will release the feedforward inhibition of MBON06 onto MBON07 (α1), freeing it to participate in driving the PAM11-aad (α1) DANs (*Ichinose et al., 2015*). Release of MBON06 inhibition should also simultaneously potentiate the responses of MBON14 (α3) to the conditioned odor (*Plaçais et al., 2013*). Lastly, releasing the strong inhibition from MBON06 frees MBON11 to provide weaker inhibition that would further favor odor-driven approach (*Perisse et al., 2016*; *Sayin et al., 2019*).

Aversive learning will also alter the function of the MBON06:MBON11 network motif (*Figure 40*). Aversive reinforcement through the PPL101 (γ1pedc) DAN depresses KC-MBON11 connections. This depression releases MBON06 from MBON11-mediated suppression and allows MBON06 to then suppress output through MBON07 and MBON11, further favoring odor-driven avoidance.

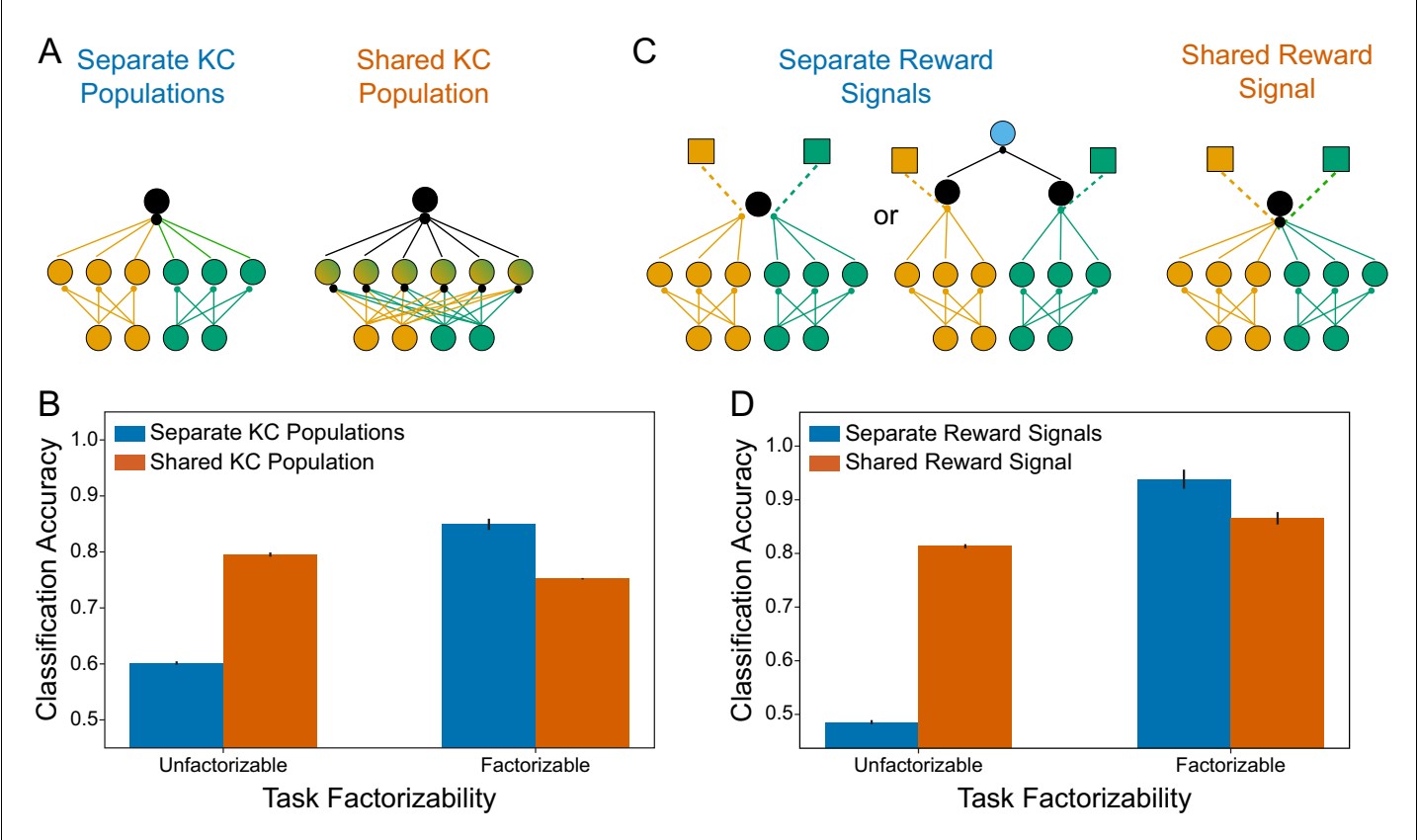

**Figure 38.** Effects of segregated or mixed-modality signals. (A) Schematic of two possible architectures for an MB-like circuit, in which KCs are either specialized for one sensory modality (left) or receive mixed input (right). Bottom layer circles represent PN inputs, middle layer circles represent KCs, and black output circles represent MBONs. Green and yellow shades indicate different sensory modalities. (B) Task performance in a stimulus discrimination model for the architectures shown in A. Stimuli consist of random binary patterns in two sensory modalities. In the 'factorizable' task, the stimulus from each modality has an assigned valence (positive or negative) and the overall valence is positive only if both constituent valences are positive. In the 'unfactorizable' task, each stimulus is randomly assigned its own unique valence, irrespective of its constituent modalities. A linear readout, modeling an MBON, is fit to discriminate the valences of the stimuli based on the activations of the KC population. The linear fit procedure corresponds to the experienced-based plasticity of KC-MBON synapses. Task performance is defined as the binary classification accuracy—that is, the frequency with which the thresholded response of the MBON corresponds correctly to the valence of the input—of this MBON readout. 25 stimuli are presented in the unfactorizable task and 200 in the factorizable task, to make overall task difficulty comparable. Error bars represent standard error over 20 simulations. (C) Schematic of possible architectures for DAN-dependent plasticity in an MB-like circuit. In the leftmost model, DANs (colored squares) modulate KC-MBON synapses for KCs of a particular sensory modality. The second-to-left model depicts two MBONs, each integrating input from KCs of a particular sensory modality, and each with its own corresponding DAN, that additively converge onto a common output (blue). These two models are functionally equivalent, and hence are grouped together under 'separate reward signals.' In the rightmost model, there is a single MBON, and DANs modulate all the KC-MBON synapses (right). (D) Task performance in a stimulus discrimination model (same as B) for the architectures shown in C. For each task the optimal KC structure from B is used ('separate' for the factorizable task, 'shared' for the unfactorizable task).

## The influence of internal state in the network

Several studies have described the influence of internal states such as hunger and thirst on the function and physiology of the MB. In essence, states appear to modulate the DAN-MBON network so that the fly preferentially engages in the pursuit of its greatest need (*Senapati et al., 2019*). Since our current knowledge suggests these deprivation states employ volume release of modulatory peptides or monoamines to control specific DANs and their downstream MBONs (*Krashes et al., 2009*; *Lin et al., 2014b*; *Lewis et al., 2015*; *Tsao et al., 2018*; *Sayin et al., 2019*; *Siju et al., 2020*), the connectome we have analyzed does not provide a complete description of this circuit. Nevertheless, direct connectivity does provide some interesting new insight concerning hunger (*Figure 41*) and thirst-dependent control.

The MBON06 (β1>α):MBON11 (γ1pedc>α/β) cross-inhibitory network motif is likely to be relevant for the dependence of learning and memory expression on hunger state (*Figure 41*). PPL101 (γ1pedc) DANs and MBON11 (γ1pedc>α/β) are sensitive to nutrient/satiety status, with MBON11 being more responsive in hungry flies (*Krashes et al., 2009*; *Plaçais et al., 2013*; *Perisse et al., 2016*; *Pavlowsky et al., 2018*). Therefore, the hungry state favors the activity of the MBONs that are normally repressed by MBON06.

Thirst-dependent seeking of water vapor requires the activity of DANs innervating the β′2 compartment (*Lin et al., 2014b*). Our current work shows that these DANs are likely to directly modulate thermo/hygrosensory KCs. In addition, a recent study showed that thirst-dependent expression of water memory required peptidergic suppression of the activity of both the PPL103(γ2α′1) and PAM02 (β′2a) DANs (*Senapati et al., 2019*). Interestingly, blocking the PAM02 (β′2a) DANs released memory expression in water-sated flies whereas blocking the PPL103 (γ2α′1) DANs had no effect. However, if the PPL103 (γ2α′1) DANs were blocked together with PAM02 (β′2a) they further facilitated water memory expression. The connectome suggests circuit mechanisms that could reconcile these observations: MBON12 (γ2α′1) provides strong cholinergic input to the PAM02 (β′2a) DANs, suggesting that PPL103 (γ2α′1) DANs might facilitate water memory expression by suppressing MBON12's excitatory input onto PAM02 (β′2a) DANs.

## A possible role for MB output in the control of movement

A role for the MB in guiding locomotion and navigation in ants and other insects has been proposed (*Ardin et al., 2016*; *Collett and Collett, 2018*; *Kamhi et al., 2020*; *Kim et al., 2019*; *Mizunami et al., 1998*; *Le Möel and Wystrach, 2020*; *Paulk and Gronenberg, 2008*; *Sun et al., 2020*). The strong and direct connections we observed from the majority of MBONs to the CX, the fly's navigation and locomotion control center, provide one circuit path for the MB to exert influence on motor behaviors. Discovering how this input is utilized by the CX will require additional experimental work.

Optogenetic activation of *Drosophila* MBONs can promote attraction or avoidance by influencing turning at the border between regions with and without stimulating light (*Aso et al., 2014b*). The effect of MBON activation is additive: coactivation of positive-valence MBONs produced stronger attraction, whereas coactivation of positive and negative-valence MBONs cancelled each other out. Because the fly needs to balance the outputs of different compartments, we expect that those downstream neurons that integrate inputs from multiple MBONs will have a privileged role in motor control.

The activity of some DANs has been shown to correlate with motor activity (*Berry et al., 2015*; *Cohn et al., 2015*), and the optogenetic activation of PAM-β′2 or PPL1-α3 DANs can attract flies, indicating that DAN activity can itself in some circumstances drive motor behavior. The circuit mechanisms generating the correlation between DAN activity and motor behavior remain to be discovered. Downstream targets of MBONs provide extensive input to DANs, and we found that neurons downstream of multiple MBONs are twice as likely as other MBON targets to provide such direct input to DANs.

We also discovered a direct pathway mediated by atypical MBONs that connects to the descending neurons (DNs) that control turning, an observation that provides additional support for the importance of the MB in the control of movement. The connections of these MBONs appear to be structured so as to promote directional movement, often involving a push-pull arrangement of MBONs signaling approach and avoidance. In addition to direct connections to DNs, there is a network of connections mediated by both local LAL interneurons and interneurons that connect the right and left hemisphere LALs. Examples of these circuit motifs are diagrammed in *Figure 42*. The atypical MBONs that connect directly to the descending steering system, MBON26, MBON27, MBON31 and MBON32, appear to be among the most integrative neurons in the MB system in the sense that they combine direct KC input from the MB compartments with both input from many other MBONs and non-MB input. At the level of the descending neurons, the highly processed signals from these MBONs are combined with inputs from many other sources, including the central complex, to affect a decision to turn. This high degree of integration presumably reflects the complexity and importance of this decision, with many factors involved that might act individually or in combination.

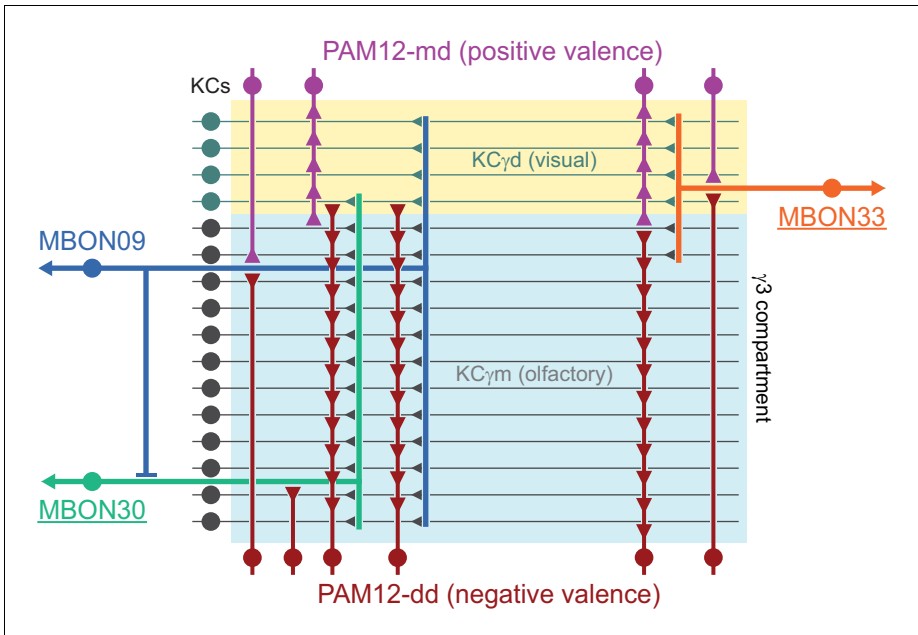

**Figure 39.** Schematic of subcompartment DAN and MBON innervation within γ3. The two PAM12 (γ3) DAN subtypes likely represent opposite valence based on their shared input with either aversively reinforcing PPL1-DANs or appetitively reinforcing PAM08 (γ4) DANs. Whereas PAM12-md DANs selectively innervate the visual γd KCs, the PAM12-dd DANs innervate both γd and γm (olfactory) KCs. The dendrite of atypical MBON30 is restricted to the olfactory γm (olfactory) KCs and its DAN input is exclusively from the PAM12-dd subtype. In contrast, the dendrite of atypical MBON33 preferentially collects γd KC input and receives input from both PAM12-md and PAM12-dd DANs. The GABAergic MBON09 (γ3β'1) pools γm and γd KC input, receives DAN input from PAM12-md and selectively synapses onto MBON30.

Visual input to the MB is over-represented in the output to the descending neurons, predominantly through MBON27. Short- and long-term learning based on features in a visual scene has been reported to involve the CX (*Liu et al., 2006*; *Neuser et al., 2008*). Plasticity in the CX enables visual feature input from the sky and surrounding scenery to be mapped flexibly onto the fly's internal compass (*Fisher et al., 2019*; *Kim et al., 2019*). The visual input conveyed to the MB and, presumably, the learning at the synapses between visual KCs and MBON27, may be of lower resolution, encoding broader features such as color and contrast (*Guo and Guo, 2005*; *Tang and Guo, 2001*; *Zhang et al., 2007*). An early study demonstrated that the MB is dispensable for flying *Drosophila* to learn shapes but that it is required for them to generalize their learning if the visual context changes between training and testing (*Liu et al., 1999*). Memory of visual features and the ability to generalize context could allow visual landmarks to help guide navigation either through the CX or by directly influencing descending neurons. The thermo/hygrosensory features conveyed by MBON26 could play a similar role, as could the large amount of odor-related information present in this MBON output pathway.

## Concluding remarks

The MB has an evolutionarily-conserved circuit architecture and uses evolutionarily-conserved molecular mechanisms of synaptic plasticity. The dense connectome analysed in this report has uncovered many unanticipated circuit motifs and suggested potential circuit mechanisms that now need to be explored experimentally. In *Drosophila*, we currently have access to many of the required tools such as cell-type-specific driver lines, genetically encoded sensors and microscopy methods to observe whole-brain neuronal activity and fine ultrastructure. These features make the fly an excellent system in which to study many general issues in neuroscience, including: the functional diversity of dopaminergic neurons that carry distributed reinforcement signals, the interactions between parallel memory systems, and memory-guided action selection, as well as the mechanisms underlying cell-type-specific plasticity rules, memory consolidation, and the influence of internal state. We expect studies

of the MB to provide insight into general principles used for cognitive functions across the animal kingdom.

As mentioned in the introduction, the MB shares many features with the vertebrate cerebellum, and our results should be informative for studies of the cerebellum proper as well as other cerebellum-like structures such as the dorsal cochlear nucleus and the electrosensory lobe of electric fish. A distinctive feature of these systems, and of the MB, is that learning is driven by a particular mechanism; for example DAN modulation in the MB or complex spiking driven by climbing fiber input in the cerebellum. Studies of learning in cortical circuits have traditionally focused on Hebbian forms of learning driven by the ongoing input and output activity of a neuron. However, recent results from both hippocampal (*Bittner et al., 2017*) and cortical (*Gambino et al., 2014*; *Lacefield et al., 2019*; *Larkum, 2013*) circuits have stressed the importance of plasticity that is driven by dendritic plateau potentials or bursts that resemble the distinct learning events seen in cerebellar and MB circuits. Thus, the form of plasticity seen in the MB and its control by output and modulatory circuits may inform studies of learning in the cerebral cortex as well.

## Materials and methods

### Connectivity and morphological data

The analyses reported in this paper were all based on the hemibrain:v1.1 dataset (*Scheffer et al., 2020*). Analyses were done by querying this dataset using the neuPrint user interface or, for more complex queries, by directly querying the neuPrint backend Neo4J graph database (*Clements et al., 2020*). In some analyses and diagrams, a threshold was applied so as to only consider connections representing more than a certain number or percentage of synapses. Any such thresholds are stated in the figure legends; otherwise, all connections were included in the analysis. Each individual neuron has a unique numerical identifier (generally 8 to 10 digits) that refers to that neuron as seen in this dataset. Neurons were also grouped into putative cell types and given names as described in *Scheffer et al., 2020* and, for olfactory neurons, Schlegel et al., (2020); we use those v1.1 names here. Even though most of the cell types in the MB were already known, we still found a few new cell types, which we named using established naming schemes. We further refined morphological groupings with relevant information on connectivity (see below).

Each neuron's unique identifier is permanent, while we expect cell type classifications and neuron names to be continuously updated in response to new biological information. For that reason, we present the unique identifiers for all the neurons we discuss, either by stating them in the main text, figures or figure legends, or by providing links in the figure legends to the appropriate data in neuPrint. The version of the dataset that we used for the analyses reported here will be archived under the name 'hemibrain:v1.1' and will remain available in neuPrint, even after additional data releases are made. As described in *Scheffer et al., 2020*, in addition to the 'named neurons' (whose status is indicated as 'Traced' in neuPrint) there are also small fragments that, while assigned a unique identifier, are not connected to any named neuron. These small fragments were excluded from our analysis.

### Neurotransmitter predictions

Computational neurotransmitter predictions were carried out as described in *Eckstein et al., 2020*. Synapse locations (minimum of 100) within the neurons in the FAFB dataset corresponding to each of the atypical and typical MBONs were identified based on their characteristic T-bar morphology and used for predictions.

### Morphological clustering

Our overall method of cell type classification is described in *Scheffer et al., 2020* and was applied to generate *Figures 3* and *6*.

Kenyon Cell Morphological Clustering. Analysis of Kenyon cell morphologies (*Figures 4* and *5*) was carried out in R using the natverse toolkit (http://natverse.org; *Bates et al., 2020a*). Briefly, skeletons were downloaded from neuprint.janelia.org using the neuprint_read_neurons function from the neuprintr package (https://github.com/natverse/neuprintr; *Bates and Jefferis, 2020*) or the hemibrainr package (https://github.com/flyconnectome/hemibrainr; *Clements et al., 2020*),

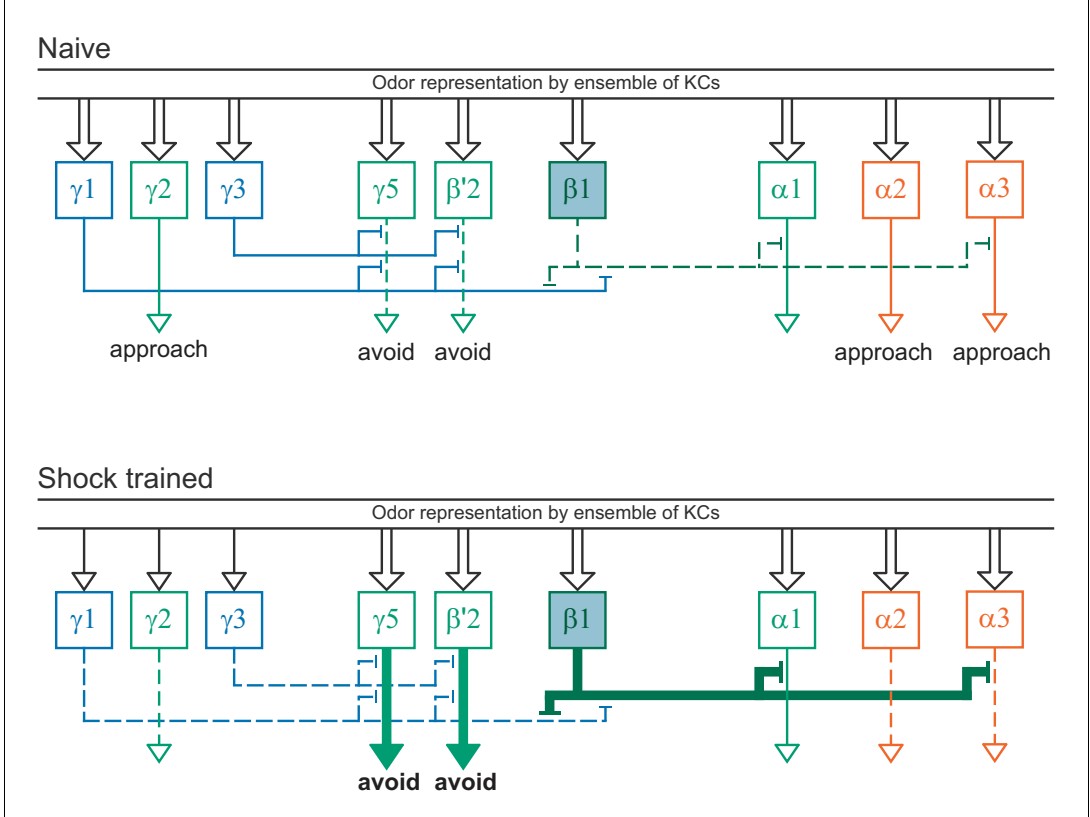

**Figure 40.** MBON network representation of aversive memory. Shock responsive DANs innervating γ1, γ2 and γ3 depress odor-specific KC synapses onto the respective MBONs. Depression of MBON11 (γ1pedc>α/β) and MBON09 (γ3β′1) releases their feedforward inhibition onto the avoidance-directing γ5 and β′2 MBONs. In addition, MBON11 release increases the inhibitory effect of the MBON06 (β1>α) onto the approach-directing α1 and α3 compartments. Repetitive aversive training trials, and a release from inhibition also allows the DANs in α2 and α3 to depress odor-specific KC synapses onto the respective approach- directing MBONs. These changes together leave the network in a configuration to direct strong odor-specific avoidance behavior.

which healed any gaps in the skeleton and rerooted on the soma. When required, neurons were simplified to one major branch using nat::simplify_neuron (https://github.com/natverse/nat; *Schlegel, 2018*). Prior to NBLAST, neurons were scaled to units of microns, resampled to 1 μm step size and converted to dotprops format with k = 5 neighbors. In order to reduce the weight given to the many fine branches present in EM reconstructions, nat::prune_twigs was applied with twig_length = 2. Standard NBLAST clustering was then carried out using the nat.nblast package as previously described (*Costa et al., 2016*). For KC cell typing, we carried out a stepwise manual NBLAST clustering followed by manual review, which resulted in 17 reassignments. These manual reassignments were almost exclusively cases in which a KC more closely resembled one KCα/β subtype in the vertical lobe and either the subsequent or previously generated subtype in the medial lobe, possibly because it was born during the developmental transition from generation of KCα/βs to KCα/βm or from KCα/βm to KCα/βc.

Morphological Clustering of DANs. DAN cell types reported in the hemibrain:v1.1 were further divided by morphological clustering into subtypes using NBLAST and manual inspection as described in *Otto et al., 2020*. To generate the results presented in *Figures 28*, *30*, *31*, the same steps as described above, but no simplification was carried out, and nat::prune_twigs was applied with twig_length = 5 a combination that was manually reviewed to preserve subtle differences in dendrites and axons. For *Figure 28* and *Figure 28—figure supplement 1*, hierarchical clustering was performed on DANs of each type/compartment using a wrapper function for base R clustering functions with nat.nblast (nat.blast::nhclust), taking Euclidean distance matrices of similarity scores, with average linkage clustering criterion. The number of clusters and the content was manually reviewed.

Clustering by morphology of the top dendritic inputs to each DAN subtype (*Figures 31–37*) was performed as follows: The 50 input neurons with the most input synapses to each DAN subtype were clustered using hierarchical clustering with average linkage criterion on all-by-all NBLAST similarity scores, which were obtained as described above. In addition, a multi-step approach was needed because of the morphological diversity of the upstream population (see *Otto et al., 2020*). Only designated right hemisphere dendritic input neurons were considered; MB intrinsic neurons (MBONs, KCs, DPM, and APL) and unnamed neuron fragments were excluded. Before clustering, neurons were simplified by nat::prune_twigs with twig length = 5. Input neurons were initially split into 25 coarse clusters, largely representative of neuropil of origin. These primary clusters were then individually sub-clustered to yield 235 clusters using an iterative manual review process, taking into account within cluster differences in connectivity. Thirty-five representative clusters were selected and used for the analyses shown in *Figures 31–37*.

## Connectivity analysis

Connectivity-based clustering: Neurons with practically indistinguishable shapes but with different connectivity patterns can often be split into connectivity subtypes within a morphology type. To generate the cell type assignments reported in *Scheffer et al., 2020* we made extensive use of a tool for cell type clustering based on neuron connectivity, called CBLAST. In an iterative process, using neuron morphology as a template, neurons were regrouped after more careful examination of neuron projection patterns and their connections. This is especially useful in the case of a dataset like ours, in which noise and missing data make it difficult to rely solely on connectivity to find a good partitioning automatically. CBLAST clusters neurons together using a similarity feature score defined by how the neuron distributes inputs and outputs to different neuron types. In some cases, this readily exposes incompleteness (for example, due to the finite size of the volume) in some neurons. Based on these interactions, we made decisions and refined the clusters manually, iterating until further changes are not observed. CBLAST usually generates clusters that are consistent with the morphological groupings of the neurons, with CBLAST often suggesting new sub-groupings as intended.

In a number of instances we obtained high-level groupings of neurons based on their input or output connectivity and without regard to morphology: *Figure 10—figure supplement 3*; *Figure 11—figure supplements 2,4,5*; *Figure 13—figure supplements 1,2*; *Figure 16*; *Figure 16—figure supplements 1,3*; *Figure 17*; *Figure 17—figure supplement 1*; *Figure 27*; *Figure 27—figure supplements 1,2*. This was achieved by performing spectral clustering (*Shi and Malik, 2000*) on the input (or output) connectivity of a set of neurons using cosine similarity as the clustering metric. Spectral clustering is particularly appropriate in cases where there is no clear hierarchical structure to the data but there are clearly defined groupings. Spectral clustering requires specifying the number of clusters in advance – in some cases, we specified this value manually, and in others, we automatically determined it by choosing the value that yielded the optimal silhouette score, a measure of clustering quality. All other parameters of the implementation were taken from the default values used by the SpectralClustering method in Python's scikit-learn package.

Connectivity analyses shown in *Figure 16—figure supplements 3,4*; *Figure 17—figure supplement 1* and *Figure 21* were carried out in Python using neuprint-python (https://github.com/connectome-neuprint/neuprint-python; *Jari Oksanen, 2019*) wrapped by navis (https://github.com/schlegelp/navis) to fetch data from neuPrint. Data were processed and plotted using pandas (https://pandas.pydata.org/) and seaborn (https://seaborn.pydata.org/).

For the hierarchical clustering by input connectivity of PAM DANs shown in *Figure 28*, KC-DAN connectivity was excluded to prevent bias. Before clustering the number of synapses between input neurons and each DAN was normalized by the remaining input to that DAN (minus KC input). Hierarchical clustering with the r-base function hclust was performed for DANs of each type/compartment using the Manhattan distance between upstream connectivity profiles of DANs with Ward's clustering criterion.

For determination of the DAN to KC and DAN to MBON connectivity shown in *Figure 30* and *Figure 30—figure supplements 2* and *3*, synapse counts between subtypes of DANs and subclasses of KCs were normalized by the total synapse count between DANs and that KC type, and then thresholded at 0.5%. Synapse counts between DAN subtypes of the right hemisphere and MBONs

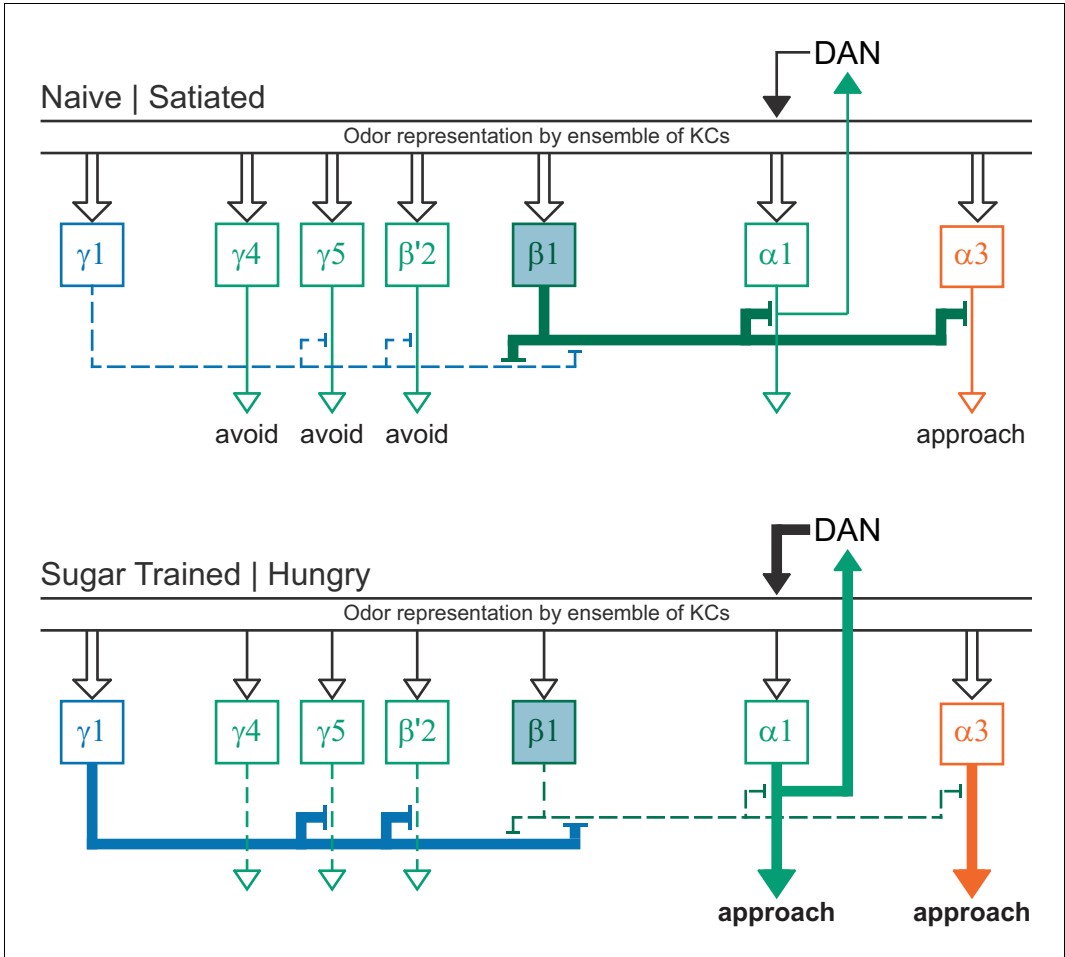

**Figure 41.** MBON network representation of appetitive memory. Sugar-responsive DANs innervating γ4, γ5, β'2, and β1 depress odor-specific KC synapses onto the respective MBONs. Depression of the odor-specific response of MBON06 (β1>α) reduces odor-specific feedforward inhibition onto the MBON07 (α1) and MBON14 (α3). This frees α3 to direct odor-driven approach behavior and increases activity in a recurrent α1 MBON07-PAM11aad DAN loop to consolidate long-term appetitive memory. MBON11 (γ1pedc>α/β) is sensitive to satiety state and is more responsive when the fly is hungry. Hunger therefore further promotes appetitive memory formation and expression by increasing inhibition onto MBON01 (γ5β'2a) and β'2 MBONs and by further reducing the feedforward inhibitory effects of MBON06 (β1>α) MBON within α1 and α3. These changes together leave the network in a hunger state-dependent configuration that directs strong odor-specific approach behavior.

with dendrites in the right hemisphere were normalized by the total number of synapses connections made to that MBON type, and then thresholded at 0.1%.

For producing the connectivity data shown in *Figures 31–37*, connectivity information was retrieved from neuPrint with the neuprintr function neuprint_connection_table (natverse) for each morphological cluster of upstream neurons to each DAN subtype innervating the right MB. Only synapses in the right hemisphere were used due to incomplete connectivity in the left hemisphere and to prevent bias between PPL and PAM DANs. Connectivity to DAN subtypes was thresholded as indicated in the figure legends.

Comparison of clustering based on morphology and connectivity (*Figure 28*): Tanglegrams were generated to facilitate visual comparison of dendrograms of morphology- and connectivity-based clustering using the tanglegram function of the dendextend package (see *Otto et al., 2020*). Dendrogram layouts were determined to minimize edge crossing using dendextend::untangle with method='step2side' (*Galili, 2015*). The Mantel test (*Legendre and Legendre, 2012*) implemented in vegan::mantel (https://github.com/vegandevs/vegan) was used to evaluate the similarity of morphology- and connectivity- based clustering. Pearson's correlation between the distance matrices of these two observed datasets was calculated, then one of the matrices was shuffled all possible ways

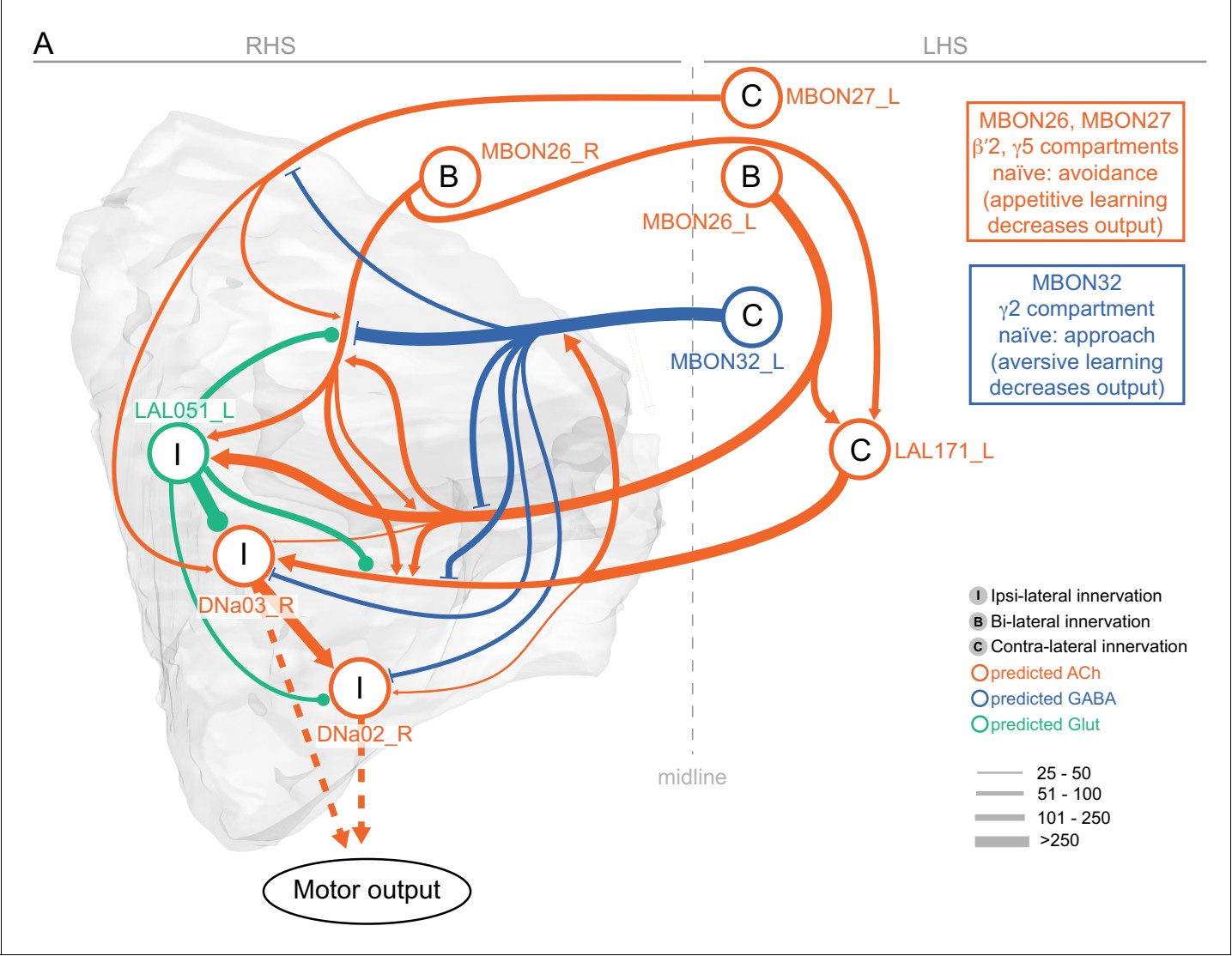

**Figure 42.** Multiple paths from atypical MBONs to DNs. A simplified circuit diagram illustrating some of the prominent motifs used to connect atypical MBONs with the dendrites of DNs in the right hemisphere LAL. While several atypical MBONs synapse directly onto DNs, there are additional paths connecting atypical MBONs to DNs that are mediated by interneurons. Some interneurons, such as LAL051, are local, and others, such as LAL171, connect the right (RHS) and left (LHS) hemisphere LALs. Axo-axonal connections between MBONs and interneurons are common. These connections seem highly selective for two DNs cell types, DNa02 and DNa03, that are involved in steering the fly's direction of movement: DNa03_R is MBON27's third most highly connected output, MBON26's 17[th], LAL051_R's second and LAL171_L's third. While only about 15% of the DNs have so far been identified in the hemibrain v1.1 dataset, among those ~50 DN types only DNa03 and DNa02 receive significant input from the MBONs shown here. The atypical MBONs represent different types of sensory information, have different predicted behavioral valences and transmitters, and push-pull arrangements on shared targets are common. The GABAergic MBON32 has its dendrites in the γ2 compartment where it receives inputs from KCs representing both olfactory and visual sensory information (*Figure 15* and *Figure 25—figure supplement 1*). The cholinergic MBON26 has its dendrites in the β'2 compartment where it receives inputs from KCs representing both olfactory and hygro/thermo sensory information (*Figure 15* and *Figure 25—figure supplement 1*) and the cholinergic MBON27 has its inputs in the γ5 compartment where it receives inputs from KCs representing predominantly visual input (*Figure 15* and *Figure 25—figure supplement 1*). MBONs 27 and 32 connect to the contralateral LAL and therefore only MBON 27_L and MBON32_L are shown, whereas MBON26 makes bilateral connections and both MBON26_L and MBON26_R are shown. LAL171_L receives about 20% of its total input from these two MBON types and then provides strong input to the contralateral DNa03. LAL051 gets over 15% of its input from left and right hemisphere MBON26s and also provides strong input to DNa03. DNa03 is strongly connected to DNa02, which promotes turning to the stimulated side (*Rayshubskiy et al., 2020*). Thus activity in MBON27 and MBON26 might promote avoidance by activating DNa03 in the contralateral LAL, causing steering away from a stimulus. Learning-related reductions in MBON26/27 activity would then promote approach. In general, the complexity of this circuitry, along with that of the atypical output network (as discussed in the text), further illustrates the highly integrative nature of the decision to turn. The synaptic inputs and outputs to LAL171_L were estimated using a combination of information from LAL171_L and LAL171_R to

*Figure 42 continued on next page*

*Figure 42 continued*

compensate for the incompleteness of the left hemisphere LAL in the hemibrain volume. See *Figure 42—figure supplement 1* for the morphologies of LAL051 and LAL171.

The online version of this article includes the following figure supplement(s) for figure 42:

**Figure supplement 1.** Morphologies of LAL051 and LAL171.

or at least $10^7$ times and each event tested for correlation with the observed data. The number of events where the correlation was higher than between the two original datasets was divided by the amount of comparisons to create a p-value. When p-values were lower than the significance level, the null model of independence between the two feature spaces was rejected.

## Calculation of multi-step effective connectivity

In several analyses, we computed the 'effective' connectivity through multi-synaptic pathways between a set of source and target neurons: *Figure 10—figure supplement 2*; *Figure 11—figure supplement 1*; *Figure 26—figure supplements 2,3*; *Figure 27—figure supplement 3*. Although our procedure generalizes to pathways of any length, we only performed it for two-step (or 'one-hop') pathways. To do so, we determined the set of interneurons either postsynaptic to the source population or presynaptic to the target population. Starting with the matrices of source-interneuron connectivity and interneuron-target connectivity, we normalized each so that the sum of inputs to each postsynaptic cell summed to 1. Then we multiplied the two matrices to yield an estimate of effective source-target connectivity. This procedure reflects the assumption that an output synapse from an interneuron conveys information about its inputs to varying degrees, which are proportional to the number of input synapses coming from each input.

## Data presentation

The 3D renderings of neurons presented in the Figures were generated using the visualization tools of NeuTu (*Zhao et al., 2018a*); gray -scale images of EM data were taken from NeuTu. Annotations were added using Adobe Illustrator. Color depth MIP masks of MCFO or FAFB skeletons were generated using the ColorMIP_Mask_Search plugin for Fiji (https://github.com/JaneliaSciComp/Color-MIP_Mask_Search; *Otsuna et al., 2018*). Cytoscape (cytoscape.org) was used to produce the node layout of connectivity diagrams of connections between neurons, which were then edited in Adobe Illustrator. Videos were produced using Blender (blender.org) and Python scripts (*Hubbard, 2020*). Narration was recorded using Camtasia (techsmith.com) and text and narration were added to videos using Adobe Premiere Pro.

## Acknowledgements

This work was supported by the Howard Hughes Medical Institute, by the MRC (MC-U105188491 to GSXEJ), by the Wellcome Trust (203261/Z/16/Z to GSXEJ, GMR, and SW), by the Simons Foundation (Simons Collaboration on the Global Brain to JL, YA, LFA, AL-K, and GMR), by Burroughs Wellcome Foundation (1017109 to AL-K), by NIH (R01EB029858 to AL-K), by the NSF (NeuroNex Award to LFA, AL-K and JL) and by a Department of Energy Computational Science Graduate Fellowship (DE-SC0020347) to JL. We thank Tanya Wolff for her help in the analysis of MBON to CX connections. We thank Tanya Wolff and Vivek Jayaramn for detailed comments on the manuscript, Kazunori Shinomiya for help generating *Figure 2—figure supplement 1*, Philip Hubbard for providing advice on video production and for generating segments of *Figure 9—video 1*, *Figure 10—video 1*, *Figure 20—video 1*. Emily Joyce narrated the Videos. Konrad Heinz and Joseph Hsu provided proofreading assistance and the The Janelia FlyLight project team provided the images used in LM-EM comparisons (*Figure 10—figure supplement 1*).

# Additional information

## Funding

| Funder | Grant reference number | Author |
| --- | --- | --- |
| Howard Hughes Medical Institute | Internal funding | Marisa Dreher<br>Aljoscha Nern<br>Shin-ya Takemura<br>Nils Eckstein<br>Audrey Francis<br>Ruchi Parekh<br>Louis K Scheffer<br>Yoshinori Aso<br>Gerald M Rubin |
| Wellcome Trust | 203261/Z/16/Z | Feng Li<br>Elizabeth C Marin<br>Nils Otto<br>Georgia Dempsey<br>Ildiko Stark<br>Philipp Schlegel<br>Tansy Yang<br>Amalia Braun<br>Marta Costa<br>Gregory SXE Jefferis<br>Scott Waddell<br>Gerald M Rubin |
| Wellcome Trust | 200846/Z/16/Z | Scott Waddell |
| National Science Foundation | NeuroNex DBI-1707398 | Jack W Lindsey<br>Larry F Abbott<br>Ashok Litwin-Kumar |
| Department of Energy | DE-SC0020347 | Jack W Lindsey |
| Medical Research Council | MC-U105188491 | Alexander S Bates<br>Markus William Pleijzier<br>Philipp Schlegel<br>Gregory SXE Jefferis |
| Simons Foundation | SCGB | Jack W Lindsey<br>Yoshi Aso<br>Larry F Abbott<br>Ashok Litwin-Kumar<br>Gerald M Rubin |
| Burroughs Wellcome Fund | 1017109 | Ashok Litwin-Kumar |
| National Institutes of Health | R01EB029858 | Ashok Litwin-Kumar |

The funders had no role in study design, data collection and interpretation, or the decision to submit the work for publication.

## Author contributions

Feng Li, Conceptualization, Data curation, Formal analysis, Supervision, Validation, Investigation, Visualization, Methodology, Writing - original draft, Project administration, Writing - review and editing, Primary responsibility for Figure 2-figure supplement 1, Figure 3, Figure 3-figure supplement 1, Figure 6, Figure 6-figure supplement 1, Figure 7, Figure 8, Figure 8-figure supplement 1 to 14, Figure 9, Figure 9-figure supplement 1 to 3, Figure 10, Figure 10-figure supplement 4 and 5, Figure 11, Figure 11-figure supplement 3, Figure 12, Figure 19, Figure 19-figure supplement 1, Figure 20, Figure 20-figure supplement 1, Figure 21-figure supplement 2, Figure 22-figure supplement 1, Figure 23, Figure 23-figure supplement 1 to 3, Figure 24-figure supplement 1, Figure 25, Figure 25-figure supplement 1 and 2, Figure 26 and Figure 26-figure supplement 1. Contributed to Figure 10-figure supplement 1 panel D,G,K,N,O, Figure 10-figure supplement 2 panel B, Figure 10-figure supplement 3 panel B, Figure 11-figure supplement 1 panel B, Figure 11-figure supplement 2 panel B, Figure 12-figure supplement 2 panel A, Figure 18 panel B, Figure 18-figure supplement 1, Figure 18-figure supplement 3, Figure 32, Figure 32-figure supplement 1, Figure 33, Figure 33-figure supplement 1,

Figure 34, Figure 35, Figure 35-figure supplement 1, Figure 36, Figure 36-figure supplement 1, Figure 37, and Figure 37-figure supplement 1 and Figure 42; Jack W Lindsey, Formal analysis, Validation, Investigation, Visualization, Methodology, Writing - original draft, Primary responsibility for Figure 6-figure supplement 3, Figure 8-figure supplement 15, Figure 11-figure supplements 4 and 5, Figure 13-figure supplements 1 and 2, Figure 16, Figure 16-figure supplement 1 and 2, Figure 17, Figure 18-figure supplement 2, Figure 26-figure supplements 1 and 2, Figure 27, Figure 27-figure supplements 1 to 4, Figure 29 and Figure 38. Contributed to Figure 10-figure supplement 2 panel A, Figure 10-figure supplement 3 panel A and C, Figure 11-figure supplement 1 panel A, Figure 11-figure supplement 2 panel A and C, Figure 17-figure supplement 1, Figure 18 panel A, Figure 18-figure supplement 1, Figure 18-figure supplement 3 and Figure 30-figure supplement 1; Elizabeth C Marin, Data curation, Formal analysis, Supervision, Validation, Investigation, Visualization, Methodology, Writing - original draft, Primary responsibility for Figure 4, Figure 4-figure supplements 1, 2 and 4, and Figure 12-figure supplement 1. Contributed to Figure 9-figure supplement 3, Figure 10-figure supplement 1, Figure 12, Figure 12-figure supplement 2 panels B, C, and D and Figure 15- figure supplement 1 panel A; Nils Otto, Data curation, Formal analysis, Supervision, Validation, Investigation, Visualization, Methodology, Writing - original draft, Primary responsibility for Figure 28, Figure 28-figure supplement 1, Figure 30, Figure 30-figure supplements 2 to 3, Figure 31, Figure 32, Figure 32-figure supplement 1, Figure 33, Figure 33-figure supplement 1, Figure 34, Figure 35, Figure 35-figure supplement 1, Figure 36, Figure 36-figure supplement 1, Figure 37, and Figure 37-figure supplement 1. Contributed to Figure 6-figure supplement 2 and Figure 30-figure supplement 1; Marisa Dreher, Formal analysis, Validation, Visualization, Primary responsibility for Figure 24. Contributed to graphic design on many figures and generated nearly all of the videos; Georgia Dempsey, Data curation, Formal analysis, Validation, Investigation, Visualization, Contributed to Figure 6-figure supplement 2, Figure 28, Figure 28-figure supplement 1, Figure 30, Figure 30-figure supplements 2 and 3, Figure 31, Figure 32, Figure 32-figure supplement 1, Figure 33, Figure 33-figure supplement 1, Figure 34, Figure 35, and Figure 35-figure supplement 1, Figure 36, Figure 36-figure supplement 1, Figure 37, and Figure 37-figure supplement 1; Ildiko Stark, Data curation, Formal analysis, Validation, Contributed to Figure 28, Figure 30, Figure 30-figure supplements 2 and 3, Figure 31, Figure 35, and Figure 35-figure supplement 1; Alexander S Bates, Resources, Data curation, Software, Formal analysis, Validation, Investigation, Primary responsibility for Figure 21-figure supplement 1. Cross-identified new MBONs in the FAFB dataset to allow computational neurotransmitter prediction; Markus William Pleijzier, Data curation, Formal analysis, Validation, Investigation, Primary responsibility for Figure 16-figure supplement 3 and 4, and Figure 17-figure supplement 1; Philipp Schlegel, Software, Formal analysis, Validation, Investigation, Responsible for Figure 21; Aljoscha Nern, Formal analysis, Validation, Investigation, Writing - original draft, Responsible for the classification of VPNs and contributed to Figure 10-figure supplement 1; Shin-ya Takemura, Tansy Yang, Data curation, Formal analysis, Validation; Nils Eckstein, Resources, Formal analysis, Investigation, Responsible for computational neurotransmitter prediction and Figure 6-figure supplement 2; Audrey Francis, Amalia Braun, Data curation, Validated and extended neuronal morphologies in the FAFB and hemibrain datasets; Ruchi Parekh, Data curation, Supervision; Marta Costa, Data curation, Supervision, Validation, Responsible for classification of olfactory PNs; Louis K Scheffer, Formal analysis, Validation, Investigation, Visualization, Primary responsibility for Figure 5-figure supplement 2 and Figure 22; Yoshinori Aso, Funding acquisition, Validation, Investigation, Visualization, Methodology, Writing - original draft, Primary responsibility for Figure 2 and Figure 4-Figure supplement 3; Gregory SXE Jefferis, Resources, Software, Formal analysis, Supervision, Funding acquisition, Validation, Investigation, Visualization, Methodology, Writing - original draft, Responsible for computational classification of KCs and Figure 5. Contributed to Figure 4, Figure 4-figure supplement 1, 2 and 4, Figure 5-figure supplement 1, and Figure 16-figure supplement 3 and 4 and Figure 42; Larry F Abbott, Conceptualization, Formal analysis, Supervision, Funding acquisition, Validation, Investigation, Visualization, Methodology, Writing - original draft, Writing - review and editing, Primary responsibility for Figure 15 and Figure 15-figure supplements 1 to 3. Contributed to Figure 42; Ashok Litwin-Kumar, Formal analysis, Supervision, Funding acquisition, Validation, Investigation, Visualization, Methodology, Writing - original draft, Responsible for Figure 13 and Figure 14; Scott Waddell, Formal analysis, Supervision, Funding acquisition, Validation, Investigation, Visualization, Methodology, Writing - original draft, Responsible for Figure 40 and Figure 41 and contributed to Figure 39; Gerald M Rubin, Conceptualization, Formal analysis, Supervision, Funding acquisition, Validation, Investigation,

Visualization, Methodology, Writing - original draft, Project administration, Writing - review and editing, Responsible for Figure 1 and contributed to Figure 39 and Figure 42

### Author ORCIDs

Feng Li  https://orcid.org/0000-0002-6658-9175
Jack W Lindsey  https://orcid.org/0000-0003-0930-7327
Elizabeth C Marin  http://orcid.org/0000-0001-6333-0072
Nils Otto  https://orcid.org/0000-0001-9713-4088
Marisa Dreher  http://orcid.org/0000-0002-0041-9229
Georgia Dempsey  http://orcid.org/0000-0002-1854-8336
Alexander S Bates  http://orcid.org/0000-0002-1195-0445
Markus William Pleijzier  http://orcid.org/0000-0002-7297-4547
Philipp Schlegel  http://orcid.org/0000-0002-5633-1314
Aljoscha Nern  http://orcid.org/0000-0002-3822-489X
Shin-ya Takemura  http://orcid.org/0000-0003-2400-6426
Tansy Yang  http://orcid.org/0000-0003-1131-0410
Audrey Francis  http://orcid.org/0000-0003-1974-7174
Ruchi Parekh  http://orcid.org/0000-0002-8060-2807
Marta Costa  http://orcid.org/0000-0001-5948-3092
Louis K Scheffer  http://orcid.org/0000-0002-3289-6564
Yoshinori Aso  https://orcid.org/0000-0002-2939-1688
Gregory SXE Jefferis  http://orcid.org/0000-0002-0587-9355
Ashok Litwin-Kumar  http://orcid.org/0000-0003-2422-6576
Scott Waddell  http://orcid.org/0000-0003-4503-6229
Gerald M Rubin  https://orcid.org/0000-0001-8762-8703

### Decision letter and Author response

Decision letter https://doi.org/10.7554/eLife.62576.sa1
Author response https://doi.org/10.7554/eLife.62576.sa2

## Additional files

### Supplementary files

- Supplementary file 1. Constituents of each of the 402 PAM subtype clusters.
- Supplementary file 2. Constituents of each of the DAN input clusters shown in *Figure 32*.
- Transparent reporting form

### Data availability

There is no institutional resource for hosting connectome data. All the primary data used in this study are freely available through a publicly accessible web site, neuprint.janelia.org. All the underlying data behind that server are open source (CC-BY). We commit to keeping this available for at least 10 years, and provide procedures that allow users to copy any or all of it to their own computer. Login is via any Google account; users who wish to remain anonymous can create a separate account for access purposes only.

The following previously published dataset was used:

| Author(s) | Year | Dataset title | Dataset URL | Database and Identifier |
|---|---|---|---|---|
| Scheffer LK, Xu CS, Januszewski M, Luo Z, Takemura Sy, Hayworth K | 2020 | Resource Collection for a Connectome and Analysis of the Adult Drosophila Central Brain | https://doi.org/10.25378/janelia.12818645.v1 | Figshare, 10.25378/janelia.12818645.v1 |

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
