## [Decision Letter]

**Acceptance summary:**

The *Drosophila* hemibrain connectome is a resource that will be important for pushing forward our understanding of circuits in general. This paper is a data-rich contribution which shows how thoughtful questions can be put to this type of complicated data set. The results, which include detailed analyses of known cell types and identification of previously unknown cell types, are quite exciting and will generate a lot of new ideas/hypotheses in the community.

**Decision letter after peer review:**

Thank you for submitting your article "The connectome of the adult *Drosophila* mushroom body: implications for function" for consideration by *eLife*. Your article has been reviewed by three peer reviewers, including Leslie C Griffith as the Reviewing Editor and Reviewer #1, and the evaluation has been overseen by Eve Marder as the Senior Editor. The following individuals involved in review of your submission have agreed to reveal their identity: Jason Pipkin (Reviewer #2); Chris Q Doe (Reviewer #3).

The reviewers have discussed the reviews with one another and the Reviewing Editor has drafted this decision to help you prepare a revised submission.

Summary:

The *Drosophila* hemi brain EM reconstruction is an important resource for the fly community. This paper demonstrates how thoughtful, but broad, questions can be put to such a data set. Some of the analyses presented put details onto general concepts that were already out there in the field, but there is also a wealth of real discovery here that is going to catalyze new ways of looking at mushroom body function in the context of the rest of the brain. This is the most comprehensive mushroom body paper ever written and fully realizes the utility of connectomics.

Essential revisions:

The only concern the reviewers had was around the use of synapse number thresholds. It would be useful for the reader to have a bit more insight into why particular thresholds were chosen and what the impact of better resolution (i.e. more synapses) will have on conclusions since the Janelia team notes that the true count of synapses they've identified is likely to be an underestimate.

---

## [Author Response]

Essential revisions:The only concern the reviewers had was around the use of synapse number thresholds. It would be useful for the reader to have a bit more insight into why particular thresholds were chosen and what the impact of better resolution (i.e. more synapses) will have on conclusions since the Janelia team notes that the true count of synapses they've identified is likely to be an underestimate.

Thresholds were chosen to make the graphs and figures show the strongest connections.

We added text to provide additional information about thresholds in two places in the manuscript:

1) In the first place we mention threshold (Introduction) we describe what a threshold is and why we use them.

2) In the Discussion we added a paragraph to describe the utility and limitations of thresholds and the assumptions we make when using them.

We also added a new figure supplement (Figure 6—figure supplement 3) that illustrates the effect of choosing different thresholds on the fraction of synapses retained for MBON outputs and DAN inputs.